# Non-Convex Tensor Recovery from Tube-Wise Sensing

**Tongle Wu**
The Pennsylvania State University
`tfw5381@psu.edu`

**Ying Sun**$^*$
The Pennsylvania State University
`ybs5190@psu.edu`

## Abstract

In this paper, we propose a novel tube-wise local tensor compressed sensing (CS) model under the tensor product framework, where sensing operators are independently applied to each tube of a third-order tensor. To recover the low-rank ground truth tensor, we minimize a non-convex objective via Burer–Monteiro factorization and solve it using gradient descent (GD) with spectral initialization. We prove that this approach achieves exact recovery with a linear convergence rate. Notably, our method attains provably lower sample complexity than existing TCS methods if the low tubal rank ground truth tensor satisfies the defined incoherence condition. Our proof leverages the leave-one-out technique to show that gradient descent generates iterates implicitly biased towards solutions with bounded incoherence, which ensures contraction of optimization error in consecutive iterates. Empirical results validate the effectiveness of GD in solving the proposed local TCS model.

## 1 Introduction

Tensor compressed sensing (TCS) that recovers a low-rank tensor from limited measurements is a fundamental problem in signal processing and machine learning. Specifically, the classical TCS problem aims to recover a ground truth $\boldsymbol{\mathcal{X}}^\star \in \mathbb{R}^{n_1 \times n_2 \times n_3}$ tensor from a few linear measurements generated as

$$\boldsymbol{y} := [y_1, \cdots, y_m]^\top = \boldsymbol{\mathcal{A}}(\boldsymbol{\mathcal{X}}^\star) := [\langle \boldsymbol{\mathcal{A}}_1, \boldsymbol{\mathcal{X}}^\star \rangle, \cdots, \langle \boldsymbol{\mathcal{A}}_m, \boldsymbol{\mathcal{X}}^\star \rangle]^\top, \tag{1}$$

with $\boldsymbol{\mathcal{A}} : \mathbb{R}^{n_1 \times n_2 \times n_3} \to \mathbb{R}^m$ being a linear sensing map. Without any structure assumption on $\boldsymbol{\mathcal{X}}^\star$, the problem is ill-posed and it is impossible to recover it from measurements less than its dimension. In this paper, we consider $\boldsymbol{\mathcal{X}}^\star$ exhibiting a structure that can be well-approximated by a low-rank model. However, unlike a matrix that has a unique notion of rank, a tensor has multiple definitions of rank [19] induced by different tensor decompositions such as Canonical Polyadic (CP) [16], Tucker [54], tensor singular value decomposition (t-SVD) [17], and others [39, 65, 70]. Algorithms and their theoretical guarantees to recover $\boldsymbol{\mathcal{X}}^\star$ under different notions of low-rankness have been studied in a series of existing works [3, 11, 18, 26, 28, 30, 36, 43, 44].

Despite the long-standing literature, works mentioned above share a common limitation: the TCS model (1) is a global CS model where the sensing operator $\boldsymbol{\mathcal{A}}$ is applied to the entire tensor $\boldsymbol{\mathcal{X}}^\star$. In certain applications, however, fulfilling such a requirement is impossible or inefficient. For example, when dealing with massive tensor datasets that exceed available memory, one common strategy is to split the data into smaller sub-tensors and allocate them across a distributed network of storage nodes [49, 55]. In this case, it is more practical to take measurements of the sub-tensors locally at each node out of implementation efficiency and privacy considerations. Another example is real-time applications such as video streaming, where the data is modeled as a tensor whose dimension is increasing with time. For these applications, algorithms that can process incoming measurements

---

$^*$Corresponding author.

39th Conference on Neural Information Processing Systems (NeurIPS 2025).

incrementally and generate the tensor on the fly are favored over those that repeatedly measure and recover the full tensor whenever new data arrives [37, 64].

To address the above limitations in the global TCS model, recent work [64] firstly proposed a local TCS model where measurements are taken with respect to each lateral slice matrix of $\mathcal{X}^\star$. Methods proposed therein provably recover the ground truth tensor $\mathcal{X}^\star$ of low tubal rank with a lower sample complexity than the global TCS model. Inspired by this result, this work takes a further step by proposing a more flexible local TCS model that senses each tube of $\mathcal{X}^\star$. Our proposed tube-wise TCS model only requires access to individual tubes (mode-3 fiber vectors) of the third-order tensor, whereas slice-wise [64] and global sensing methods [20, 26, 28, 60] must measure the entire slice (full matrix) and the whole tensor, respectively. We refer to our framework as the tube-wise TCS model and define it precisely below:

**Definition 1** (Tube-Wise TCS). *Given third-order tensor $\mathcal{X}^\star \in \mathbb{R}^{n_1 \times n_2 \times n_3}$, tube-wise TCS measures $\mathcal{X}^\star$ by the sensing map $\mathcal{Z} : \mathbb{R}^{n_1 \times n_2 \times n_3} \to \mathbb{R}^{n_1 \times n_2 \times m}$ that consists of $n_1 \times n_2$ sensing matrices $\mathbf{Z}_{ij} \in \mathbb{R}^{m \times n_3}$, $i = 1, \ldots, n_1$, $j = 1, \ldots, n_2$ and generates the measurement*

$$\mathcal{Y} = \mathcal{Z}(\mathcal{X}^\star), \quad \text{where} \quad \mathcal{Y}(i, j, :) := \mathbf{Z}_{ij}\mathcal{X}^\star(i, j, :), \ \forall (i, j) \in [n_1] \times [n_2]. \tag{2}$$

Here, $n_1$, $n_2$, and $n_3$ are respectively the height, width, and depth of $\mathcal{X}^\star$; and $n_1$ and $n_2$ are collectively referred to as the spatial dimension. Each $\mathcal{X}^\star(i, j, :)$ denotes $(i, j)$-th tube vector of $\mathcal{X}^\star$, which is measured by $\mathbf{Z}_{ij}$ with an output vector $\mathcal{Y}(i, j, :) \in \mathbb{R}^m$, where $m$ denotes the tube-wise sample size. We can observe that given samples and measurements obtained via the tube-wise TCS model, we can turn it into an equivalent slice-wise or global TCS model, but not the other way around. This reflects the flexibility of the proposed tube-wise TCS model.

This work considers recovering $\mathcal{X}^\star$ with low tubal rank (cf. Definition 8) from $\mathcal{Y}$ under the tube-wise TCS model. The low tubal rank assumption is well-motivated in applications where multi-dimensional data exhibit strong correlations along the third mode [24, 68], such as time or spectrum. Under the t-SVD framework, this assumption enables a compact representation of the tensor and has shown empirical success in video, hyperspectral, and multichannel signal recovery tasks [22, 41, 57, 58, 62]. The TCS model under the t-SVD framework has also been widely studied in existing works [15, 25, 26, 28, 63, 66, 67]. Although we consider the Fourier transform-based t-SVD framework, our results in this paper can be extended to general t-SVD framework whose transform matrix satisfies $\mathbf{L}^\top \mathbf{L} = \mathbf{L}\mathbf{L}^\top = n_3\mathbf{I}$, which is standard in the transformed t-SVD framework [27, 29] that includes the DFT matrix as a special case.

Given the proposed tube-wise TCS model, a central question is how to design a method that can exactly recover the ground truth tensor $\mathcal{X}^\star$ that is both sample and computationally efficient. This paper addresses this question and our main contributions are summarized as follows:

- **Computational complexity.** We formulate the tensor recovery problem by minimizing a quadratic loss and encode the low rank structure utilizing tensor Burer-Monteiro (BM) factorization [5]. By reparameterizing the tensor to be learned as the product of two smaller factors based on low tubal rank assumption, the BM factorization enables more computationally and memory-efficient numerical solutions than existing t-SVD-based TCS methods [20, 25, 28, 60, 67]. Specifically, we prove that despite being nonconvex, with a sufficient amount of measurements under the defined tensor incoherence condition (cf. Assumption 2), applying the GD with proper spectral initialization finds the ground truth $\mathcal{X}^\star$ at a linear rate.
- **Tube-wise sample complexity.** We provide tube-wise sample complexity to ensure the linear rate of GD and exactly recover the ground truth, given by (12). The expression decreases linearly with the spatial dimension $n$, implying that for fixed tubal rank and tube size, increasing spatial dimension can reduce the sample complexity significantly. Such a property is unique to our local TCS model and has not appeared in any previous results, revealing that the local TCS model is more sample-efficient than existing methods, as summarized in Table 1. Simulation result shown in Fig. 2 corroborates our findings. Furthermore, our sample complexity is essentially tight since the total sample complexity of all tubes nearly matches the information-theoretical lower bound.
- **Technical analysis.** To prove the linear rate of GD under the provided tube-wise sample complexity, our analysis hinges on two key ingredients. One is showing the existence of a benign landscape near the optimum of non-convex loss (6), and the other is that GD iterates stay in this region. To this end, we leverage the leave-one-out (LOO) technique and judiciously construct two auxiliary sequences to show that GD with spectral initialization is implicitly regularized with all iterates having bounded

Table 1: Comparison of existing works under different TCS models.

| Method | TCS Model | Convexity | Convergence Rate | Normalized sample complexity |
|---|---|---|---|---|
| TNN [28] | Global | Convex | ✗ | $O(r(2n-r)n_3)$ |
| IR-tTNN [60] | Global | Non-convex | Sub-linear | ✗ |
| Schatten-$p$ TNN [20] | Global | Non-convex | ✗ | ✗ |
| FGD [26] | Global | Non-convex | Linear | $\tilde{\mathcal{O}}(r^2 n n_3)$ |
| Alt-PGD-Min [64] | Slice-wise | Non-convex | Linear | $\tilde{\mathcal{O}}(r^2 n_3)$ |
| **Ours** | Tube-wise | Non-convex | Linear | $\tilde{\mathcal{O}}\left(\max\left\{\frac{r^5 n_3}{n}, 1\right\}\right)$ |

$^\dagger$ Here we assume for simplicity and without loss the generality that $n_1 = n_2 = n$. $r$ denotes the low tubal rank of the ground truth tensor. Normalized sample complexity is defined as the total number of entries in the sensing map divided by the tensor size $n_1 \times n_2 \times n_3$.

incoherence. Based on this implicit regularization phenomenon, new concentration results are established showing that the empirical Hessian of non-convex loss is close to its population counterpart under reduced sample complexity than state-of-the-art results.

**Related works.** Based on our proposed model and technique, we organize relevant works as follows.

- **T-SVD based global TCS.** Recovering low-tubal-rank tensors under the t-SVD framework has been considered in [28], wherein a convex tensor nuclear norm (TNN) minimization formulation is proposed with order-optimal sample complexity guarantees. The results were extended to the robust recovery settings in [67]. Under the TNN minimization framework, nonconvex surrogates for tubal rank have been explored in [20] and [58] to improve performance, yet no sample complexity has been provided. In addition to low-rankness, the local smoothness structure has been incorporated in [25] for visual data to improve sample complexity. Algorithmically, all these methods require performing multiple SVDs per iteration, which is computationally intense. More recently, a nonconvex formulation based on the BM factorization for symmetric tensors was proposed in [26], wherein the factorized gradient descent (FGD) algorithm is developed to solve the problem with reduced computational costs and storage requirements. All these works focus on the global TCS model rather than the local sensing setting considered in this paper.
- **Local CS model.** Compared to the global CS model, methods for local CS have remained relatively underexplored. Recent works [21, 38, 48, 49, 55] have studied local CS for low-rank matrix recovery, yet much less work has been devoted to local CS for tensors. A lateral column-wise sketching method under the Tucker decomposition was proposed in [4], and gradient-based algorithms were introduced, but without theoretical analysis. Recent work [64] has studied slice-wise local TCS under the t-SVD framework, wherein both sample complexity and algorithmic convergence are provided. In contrast, this work considers a more fine-grained tube-wise TCS model where the sensing map is applied to each tube independently. Our model achieves improved sample complexity compared to [64], as shown in Table 1.
- **Implicit regularization in tensor learning.** Under the BM factorization, one line of research has studied overparameterized tensor models and has shown that some optimization algorithms implicitly promote low-rank structures in CP [10, 12, 45], Tucker [13, 31, 46], and t-SVD decompositions [15]. Another line of work has studied the exact parameterization setting and has proved that GD or its variants generate balanced factors [51, 52] or enforce incoherence conditions [2, 6, 7, 59]. There are also works that have investigated other types of implicit regularization in tensor learning, such as [33, 34]. Our work reveals the implicit regularization effect of gradient descent in the proposed tube-wise local TCS model, which allows the tube-wise sample complexity required for exact recovery to be reduced linearly as the spatial dimension increases, as shown in (12). This leads to a significant improvement in sample efficiency.

## 2 Preliminaries

**Notations.** Scalars, vectors, matrices, and tensors are denoted by $x$, $\boldsymbol{x}$, $\boldsymbol{X}$, and $\boldsymbol{\mathcal{X}}$, respectively. For third-order tensor $\boldsymbol{\mathcal{X}} \in \mathbb{R}^{n_1 \times n_2 \times n_3}$, we denote its $(i, j, k)$-th entry as $\boldsymbol{\mathcal{X}}_{ijk}$. The $(i, j)$-th tube, denoted as $\boldsymbol{\mathcal{X}}(i, j, :)$ or $\boldsymbol{\mathcal{X}}_{ij:}$, is considered as a tensor in $\mathbb{R}^{1 \times 1 \times n_3}$ or a vector in $\mathbb{R}^{n_3}$, depending

on the context. The $i$-th horizontal, lateral, and frontal slice of $\mathcal{X}$ are denoted as $\mathcal{X}(i,:,:)$ or $\mathcal{X}_{i::} \in \mathbb{R}^{1 \times n_2 \times n_3}$, $\mathcal{X}(:,i,:) \in \mathbb{R}^{n_1 \times 1 \times n_3}$, and $\mathcal{X}(:,:,i) \in \mathbb{R}^{n_1 \times n_2 \times 1}$ respectively. We also refer to the frontal slice $\mathcal{X}(:,:,i)$ as $\boldsymbol{X}^{(i)} \in \mathbb{R}^{n_1 \times n_2}$. $\langle \mathcal{X}, \mathcal{Y} \rangle = \sum_{ijk} \mathcal{X}_{ijk} \mathcal{Y}_{ijk}$ is the tensor inner product. $\overline{\mathcal{X}}$ denotes the result of performing DFT on all the tubes of $\mathcal{X}$, i.e., $\overline{\mathcal{X}} := \text{fft}(\mathcal{X}, [\,], 3)$; and the inverse DFT is $\mathcal{X} := \text{ifft}(\overline{\mathcal{X}}, [\,], 3)$. $\sigma_{\max}(\boldsymbol{X})$ ($\sigma_{\min}(\boldsymbol{X})$) is the largest (smallest) nonzero singular value of $\boldsymbol{X}$.

**Basics of t-SVD algebra.** The three basic operations, unfold, fold and block circulant, for $\mathcal{X} \in \mathbb{R}^{n_1 \times n_2 \times n_3}$ are defined as

$$\text{Unfold}(\mathcal{X}) := \left[ \boldsymbol{X}^{(1)}; \cdots ; \boldsymbol{X}^{(n_3)} \right]; \qquad \text{bcirc}(\mathcal{X}) := \begin{bmatrix} \boldsymbol{X}^{(1)} & \boldsymbol{X}^{(n_3)} & \cdots & \boldsymbol{X}^{(2)} \\ \boldsymbol{X}^{(2)} & \boldsymbol{X}^{(1)} & \cdots & \boldsymbol{X}^{(3)} \\ \vdots & \vdots & \ddots & \vdots \\ \boldsymbol{X}^{(n_3)} & \boldsymbol{X}^{(n_3-1)} & \cdots & \boldsymbol{X}^{(1)} \end{bmatrix}. \quad (3)$$

$$\text{Fold}(\text{Unfold}(\mathcal{X})) := \mathcal{X};$$

With the above operations, we introduce the following tensor algebraic basics [17, 28].

**Definition 2.** *The tensor product of $\mathcal{X} \in \mathbb{R}^{n_1 \times n_2 \times n_3}$ and $\mathcal{Y} \in \mathbb{R}^{n_2 \times n_4 \times n_3}$ is defined as:*

$$\mathcal{X} * \mathcal{Y} = \text{Fold}(\text{bicrc}(\mathcal{X}) \text{Unfold}(\mathcal{Y})) \in \mathbb{R}^{n_1 \times n_4 \times n_3}. \quad (4)$$

**Definition 3.** *The conjugate transpose of a tensor $\mathcal{X} \in \mathbb{R}^{n_1 \times n_2 \times n_3}$ is defined as the tensor $\mathcal{X}^\top \in \mathbb{R}^{n_2 \times n_1 \times n_3}$, obtained by taking the conjugate transpose of each frontal slice of $\mathcal{X}$, then reversing the order of transposed frontal slices 2 through $n_3$.*

**Definition 4.** *If $\mathcal{I}(:,:,1) = \boldsymbol{I}_n$ and $\mathcal{I}(:,:,i) = \boldsymbol{0}_n$ for $2 \leq i \leq n_3$, then $\mathcal{I} \in \mathbb{R}^{n \times n \times n_3}$ is defined as the identity tensor.*

**Definition 5.** *$\mathcal{Q}^{n \times n \times n_3}$ is orthogonal if and only if it satisfies $\mathcal{Q}^\top * \mathcal{Q} = \mathcal{Q} * \mathcal{Q}^\top = \mathcal{I}$.*

**Definition 6.** *A tensor is $f$-diagonal if each of its frontal slices is diagonal .*

**Definition 7.** *$\mathcal{X} \in \mathbb{R}^{n \times n \times n_3}$ has an inverse $\mathcal{Y}$ if $\mathcal{X} * \mathcal{Y} = \mathcal{I}$ and $\mathcal{Y} * \mathcal{X} = \mathcal{I}$. If $\mathcal{X}$ is invertible, we use $\mathcal{X}^{-1}$ to denote its inverse.*

**Definition 8.** *The tubal and average ranks are respectively defined as $\text{rank}_t(\mathcal{X}) := \max_{i \in [n_3]} \text{rank}\big(\overline{\boldsymbol{X}}^{(i)}\big)$, and $\text{rank}_a(\mathcal{X}) := (1/n_3) \cdot \sum_{i=1}^{n_3} \text{rank}\big(\overline{\boldsymbol{X}}^{(i)}\big)$.*

**Theorem 1** (t-SVD [17]). *$\mathcal{X} \in \mathbb{R}^{n_1 \times n_2 \times n_3}$ can be decomposed as $\mathcal{X} = \mathcal{U} * \mathcal{S} * \mathcal{V}^\top$ with $\mathcal{U} \in \mathbb{R}^{n_1 \times n_1 \times n_3}, \mathcal{V} \in \mathbb{R}^{n_2 \times n_2 \times n_3}$ being orthogonal and $\mathcal{S} \in \mathbb{R}^{n_1 \times n_2 \times n_3}$ $f$-diagonal.*

**Definition 9.** *The spectral norm of $\mathcal{X} \in \mathbb{R}^{n_1 \times n_2 \times n_3}$ is defined as $\|\mathcal{X}\| := \sigma_{\max}(\text{bcirc}(\mathcal{X}))$.*

Given the sensing map $\mathcal{Z}$ in (2), its adjoint map is defined as follows:

**Definition 10.** *The adjoint map of $\mathcal{Z} : \mathbb{R}^{n_1 \times n_2 \times n_3} \to \mathbb{R}^{n_1 \times n_2 \times m}$, denoted as $\mathcal{Z}^c : \mathbb{R}^{n_1 \times n_2 \times m} \to \mathbb{R}^{n_1 \times n_2 \times n_3}$, is defined such that for $\forall \mathcal{Y} \in \mathbb{R}^{n_1 \times n_2 \times m}$, it outputs $\mathcal{X} = \mathcal{Z}^c(\mathcal{Y}) \in \mathbb{R}^{n_1 \times n_2 \times n_3}$ with*

$$\mathcal{X}(i,j,:) = \boldsymbol{Z}_{ij}^\top \mathcal{Y}(i,j,:), \quad \forall (i,j) \in [n_1] \times [n_2]. \quad (5)$$

## 3 Local TCS via Burer-Monteiro factorization

We employ the Burer-Monteiro factorization [5] to parameterize the low rank tensor to be learned as $\mathcal{X} = \mathcal{A} * \mathcal{B}$, where $\mathcal{A} \in \mathbb{R}^{n_1 \times r \times n_3}, \mathcal{B} \in \mathbb{R}^{n_2 \times r \times n_3}$. Recovering $\mathcal{X}$ under the tube-wise local TCS model (2) can then be formulated as minimizing the following non-convex loss function:

$$\mathcal{L}(\mathcal{A}, \mathcal{B}) := \sum_{i=1}^{n_1} \sum_{j=1}^{n_2} \left\| \boldsymbol{Z}_{ij} \left( \mathcal{A}(i,:,:) * (\mathcal{B}(j,:,:))^\top \right) - \boldsymbol{y}_{ij} \right\|^2 + \frac{m}{8} \left\| \mathcal{A}^\top * \mathcal{A} - \mathcal{B}^\top * \mathcal{B} \right\|_F^2, \quad (6)$$

where the first term fits the model to the measurements and $\boldsymbol{y}_{ij} \in \mathbb{R}^m$ is the measurement vector for $(i,j)$-th tube. Since the BM factorization is not unique, factors $\mathcal{A}$ and $\mathcal{B}$ cannot be identified. To address this ambiguity, we generalize a technique widely used in asymmetric matrix recovery problems [8, 53, 73] to tensors and introduce in (6) the second term balances the size of $\mathcal{A}$ and $\mathcal{B}$. If there is no such term, the loss has an identifiability issue since for any invertible $\mathcal{R} \in \mathbb{R}^{r \times r \times n_3}$,

factors $(\mathcal{A}, \mathcal{B})$ and $(\mathcal{A} * \mathcal{R}^{-1}, \mathcal{B} * \mathcal{R}^{\top})$ has the same loss value. Such ambiguity may result in the norms of the two factors being highly unbalanced, leading to even divergence of the algorithm.

To minimize $\mathcal{L}$, we substitute the expression of measurement $\mathcal{Y}$ generated in Eq. (2) into the above loss function and then apply the GD method with step size $\eta$ taking the form:

$$\mathcal{A}^{t+1} = \mathcal{A}^t - \eta \mathcal{Z}^c \mathcal{Z} \left( \mathcal{A}^t * \left( \mathcal{B}^t \right)^{\top} - \mathcal{X}^{\star} \right) * \mathcal{B}^t - \frac{m\eta}{2} \mathcal{A}^t * \left( \left( \mathcal{A}^t \right)^{\top} * \mathcal{A}^t - \left( \mathcal{B}^t \right)^{\top} * \mathcal{B}^t \right) \quad (7)$$

$$\mathcal{B}^{t+1} = \mathcal{B}^t - \eta \left( \mathcal{Z}^c \mathcal{Z} \left( \mathcal{A}^t * \left( \mathcal{B}^t \right)^{\top} - \mathcal{X}^{\star} \right) \right)^{\top} * \mathcal{A}^t - \frac{m\eta}{2} \mathcal{B}^t * \left( \left( \mathcal{B}^t \right)^{\top} * \mathcal{B}^t - \left( \mathcal{A}^t \right)^{\top} * \mathcal{A}^t \right), \quad (8)$$

where the adjoint map $\mathcal{Z}^c$ is introduced in Definition 10. The estimated recovery result at each iteration $t$ is therefore given by $\mathcal{X}^t := \mathcal{A}^t * \left( \mathcal{B}^t \right)^{\top}$.

To overcome the challenge of getting stuck at local solutions due to the nonconvexity of (6) and enabling exact recovery of $\mathcal{X}^{\star}$, we further provide a spectral initialization scheme that starts GD at a point sufficiently close to the ground truth. The complete algorithm is given by Algorithm 1 and 2.

---

**Algorithm 1** Gradient descent with spectral initialization for minimizing (6).

---

**Input:** Operator $\mathcal{Z}$ and measurements $\mathcal{Y} \in \mathbb{R}^{n_1 \times n_2 \times m}$, rank $r$, step size $\eta$, total iteration $T_{\max}$.
    ▷ **Spectral Initialization:**
1: $\mathcal{X}^0 := \frac{\mathcal{Z}^c(\mathcal{Y})}{m}$
2: $[\mathcal{A}^0, \mathcal{B}^0] = \texttt{T-SVD-Spec}(\mathcal{X}^0, r)$
    ▷ **Gradient Descent:**
3: **for** $t = 0, \ldots, T_{\max} - 1$ **do**
4:     **Update** $\mathcal{A}^{t+1} = \mathcal{A}^t - \eta \nabla_{\mathcal{A}} \mathcal{L}(\mathcal{A}^t, \mathcal{B}^t), \mathcal{B}^{t+1} = \mathcal{B}^t - \eta \nabla_{\mathcal{B}} \mathcal{L}(\mathcal{A}^t, \mathcal{B}^t)$ based on (7), (8).
5: **end for**
**Output:** Recover tensor $\mathcal{X}^{T_{\max}} = \mathcal{A}^{T_{\max}} * (\mathcal{B}^{T_{\max}})^{\top}$.

---

**Algorithm 2** Truncated t-SVD based spectral initialization: $[\mathcal{A}, \mathcal{B}] = \texttt{T-SVD-Spec}(\mathcal{X}, r)$

---

**Input:** $\mathcal{X} \in \mathbb{R}^{n_1 \times n_2 \times n_3}, r$.
1: $\overline{\mathcal{X}} = \text{fft}(\mathcal{X}, [\,], 3)$
2: **for** $i = 1, \ldots, n_3$ **do**
3:     $[\overline{U}^{(i)}, \overline{S}^{(i)}, \overline{V}^{(i)}] = \text{SVD}(\overline{X}^{(i)})$
4:     $\overline{A}^{(i)} = \overline{U}^{(i)}(:, 1:r)(\overline{S}^{(i)}(1:r, 1:r))^{\frac{1}{2}}; \overline{B}^{(i)} = \overline{V}^{(i)}(:, 1:r)(\overline{S}^{(i)}(1:r, 1:r))^{\frac{1}{2}}$
5:     $\mathcal{A} = \text{ifft}(\overline{\mathcal{A}}, [\,], 3), \mathcal{B} = \text{ifft}(\overline{\mathcal{B}}, [\,], 3)$
6: **end for**
**Output:** Tensor factors $\mathcal{A}, \mathcal{B}$.

---

To establish the theoretical results, we first impose the following low-rank structure on $\mathcal{X}^{\star}$.

**Assumption 1.** *Each front slice of $\overline{\mathcal{X}^{\star}}$ has rank $r \ll \min\{n_1, n_2, n_3\}$.*

Assumption 1 is the same as the low multi-rank assumption in [69] and implies $\text{rank}_t(\mathcal{X}^{\star}) = r$. Thus, Theorem 1 implies that $\mathcal{X}^{\star}$ has exact skinny t-SVD [68] given by

$$\mathcal{X}^{\star} = \mathcal{U}^{\star} * \mathcal{S}^{\star} * (\mathcal{V}^{\star})^{\top}, \quad (9)$$

where $\mathcal{U}^{\star} \in \mathbb{R}^{n_1 \times r \times n_3}, \mathcal{S}^{\star} \in \mathbb{R}^{r \times r \times n_3}$ and $\mathcal{V}^{\star} \in \mathbb{R}^{n_2 \times r \times n_3}$. Based on (9), we introduce the two balanced factors $\mathcal{A}^{\star} \in \mathbb{R}^{n_1 \times r \times n_3}, \mathcal{B}^{\star} \in \mathbb{R}^{n_2 \times r \times n_3}$ defined as

$$\overline{A}^{\star(i)} := \overline{U}^{\star(i)} \left( \overline{S}^{\star(i)} \right)^{\frac{1}{2}}, \overline{B}^{\star(i)} := \overline{V}^{\star(i)} \left( \overline{S}^{\star(i)} \right)^{\frac{1}{2}}; \mathcal{A}^{\star} := \text{ifft}(\overline{\mathcal{A}^{\star}}, [\,], 3), \mathcal{B}^{\star} := \text{ifft}(\overline{\mathcal{B}^{\star}}, [\,], 3).$$

In addition to Assumption 1, we assume $\mathcal{X}^{\star}$ satisfies the following tensor incoherence condition, which ensures information is spread out across all tubes of the tensor, and no small subset of tubes carries dominant information.

**Assumption 2** (Tensor incoherence condition). *Let $\mathcal{X}^{\star}$ satisfy Assumption 1 with skinny t-SVD given by (9). There exists constant $\mu$ that*

$$\max_{i \in [n_1]} \|\mathcal{U}^{\star}(i, :, :)\| \leq \sqrt{\frac{\mu r}{n_1 n_3}} \quad \text{and} \quad \max_{j \in [n_2]} \|\mathcal{V}^{\star}(j, :, :)\| \leq \sqrt{\frac{\mu r}{n_2 n_3}}, \quad (10)$$

*where $\|\cdot\|$ is the tensor spectral norm (cf. Definition 9). Each horizontal slice matrix in the above is actually a degenerated third-order tensor.*

**Remark 1.** *Assumption 2 imposes uniform bounds on the tensor spectral norm of horizontal slices of tensor factors. This assumption is equivalent to the matrix weak incoherence condition [68, Eq. (31)], which is assumed in the tensor completion problem under random tubal sampling. This condition is slightly stronger than the existing tensor incoherence condition that uses the Frobenius norm [24, 28, 68], due to the fact that our measurements of $\mathcal{X}^\star$ are taken tube-wise rather than globally.*

For simplicity and without loss of generality, we assume $n_1 = n_2 = n$ for the spatial dimension of $\mathcal{X}^\star$. We denote $\sigma_{\max} := \sigma_{\max}(\mathcal{X}^\star)$, $\sigma_{\min} := \sigma_{\min}(\mathcal{X}^\star)$ and the tensor condition number $\kappa := \frac{\sigma_{\max}}{\sigma_{\min}}$. Since the factors are identifiable up to orthogonal transformations, we introduce the following alignment tensor $\mathcal{R}^t$ to remove the ambiguity:

$$\mathcal{R}^t := \arg\min_{\mathcal{R} \in \mathcal{O}_r} \left\| \begin{bmatrix} \mathcal{A}^t \\ \mathcal{B}^t \end{bmatrix} * \mathcal{R} - \begin{bmatrix} \mathcal{A}^\star \\ \mathcal{B}^\star \end{bmatrix} \right\|_F, \tag{11}$$

where the set $\mathcal{O}_r$ includes all orthogonal tensors with size $r \times r \times n_3$. Recall that $m$ is the number of measurements taken for each tube (cf. Definition 1). With Assumption 1 and 2, we are now ready to state the main result.

**Theorem 2.** *Let $\mathcal{X}^\star$ satisfy Assumption 1 and 2, and the entries of local sensing matrices $\{Z_{ij}\}$ be generated from $i.i.d$ Gaussian distribution $\mathcal{N}(0, 1)$. If the tube-wise measurements satisfy*

$$m \geq \frac{(5000C)^2 (\mu r)^5 \kappa^9 n_3 \log^2(n \vee n_3)}{n} \vee 1. \tag{12}$$

*Then under the step size condition $\eta \leq \frac{\sigma_{\min}}{100 m \sigma_{\max}^2}$, with probability at least $1 - \frac{1}{(n \vee n_3)^{10}}$ the iterates generated by Algorithm 1 satisfy*

$$\left\| \begin{bmatrix} \mathcal{A}^t \\ \mathcal{B}^t \end{bmatrix} * \mathcal{R}^t - \begin{bmatrix} \mathcal{A}^\star \\ \mathcal{B}^\star \end{bmatrix} \right\| \leq (1 - 0.1 m \eta \sigma_{\min})^t \cdot 98C \sqrt{\frac{(\mu r)^4 \kappa^5 n_3 \log^2(n \vee n_3) \sigma_{\max}}{mn}}, \tag{13}$$

$$\max_{l \in [2n]} \left\| \left( \begin{bmatrix} \mathcal{A}^t \\ \mathcal{B}^t \end{bmatrix} * \mathcal{R}^t - \begin{bmatrix} \mathcal{A}^\star \\ \mathcal{B}^\star \end{bmatrix} \right)_{l::} \right\| \leq (1 - 0.1 m \eta \sigma_{\min})^t \cdot 2964C \sqrt{\frac{(\mu r)^5 \kappa^9 n_3 \log^2(n \vee n_3) \sigma_{\max}}{mn^2}}, \tag{14}$$

*where $C$ is a universal constant.*

Theorem 2 further implies the following fine-grained uniform bound on the recovery accuracy for all tubes of $\mathcal{X}^\star$; a new result in the t-SVD based tensor recovery literature.

**Corollary 1** (Tube-wise bound). *Under the same setting as Theorem 2, $\forall \epsilon > 0$, the iterates generated by GD achieve*

$$\max_{ij} \left\| \left( \mathcal{A}^t * (\mathcal{B}^t)^\top - \mathcal{X}^\star \right)_{ij:} \right\|_F \leq \epsilon \tag{15}$$

*in at most $\mathcal{O}(\kappa^2 \log \frac{1}{\epsilon})$ iterations with high probability.*

Next, we discuss the implications of Theorem 2 and Corollary 1 in detail.

*Computational complexity.* The above results show that even for the non-convex loss (6), GD with spectral initialization converges to $\mathcal{X}^\star$ at a linear rate with iteration complexity $\mathcal{O}(\kappa^2 \log \frac{1}{\epsilon})$. The complexity scales quadratically with $\kappa$, showing that GD converges faster if the ground truth tensor is well-conditioned. The computation cost per iteration mainly lies in performing FFT, SVD, inverse FFT, and matrix multiplication. The complexity of initialization is $\mathcal{O}\left(mn_3n^2 + n^2n_3\log n_3 + n^3n_3\right)$ and that of per GD iteration is $\mathcal{O}\left(mn_3n^2 + n^2n_3\log n_3 + rnn_3\log n_3 + n^2r + nr^2\right)$. Thus, the total computational cost of Algorithm 1 is $\tilde{\mathcal{O}}\left(\kappa^2\left(mn_3n^2 + n^2n_3\log n_3 + n^2r\right) + n^3n_3\right)$.

*Sample complexity.* Based on the local TCS model (1), a naive approach to reconstruct $\mathcal{X}^\star$ is to recover each tube independently, which requires measurements $m$ on the order of $\mathcal{O}(n_3)$. In contrast, Eq. (12) shows that collectively recovering all the tubes exploiting the low-rankness structure can

significantly reduce the sample complexity. Specifically, since the right hand side of (12) decreases linearly with $n$, we have that for a fixed tubal rank $r$ and tube length $n_3$, a larger spatial dimension $n$ leads to fewer samples needed per tube to achieve exact recovery. A direct consequence is that the sample complexity of our model outperforms state-of-the-art results as illustrated by Table 1. The total number of samples taken over all tubes is $\tilde{\mathcal{O}}\left(r^5 n_3 n\right)$, which is nearly order-optimal when compared to the degrees of freedom $\mathcal{O}(r n n_3)$ for tensors with tubal rank $r$ [28].

**Remark 2.** *The dependence of the sample complexity on the tubal rank of $\mathcal{X}^\star$ is $r^5$ due to analysis technicalities. Similar high-order polynomial dependencies on the rank can also be found in the literature of non-convex matrix and tensor recovery [8, 32, 47, 50, 59].*

*The implicit regularization of GD.* Eq. (14) reveals that even without explicit regularization, the iterates generated by GD remain incoherent throughout the optimization process. This implicit bias is crucial since the non-convex loss (6) is strongly convex and smooth when restricted to the incoherent directions, thereby enabling the linear convergence of GD. Eq. (14) also leads to the new tube-wise convergence rate (15), showing that the recovery error is spread out across all tubes. The main technique to prove this property is the LOO analysis, which will be discussed in detail in Sec. 4.

## 4 Theoretical analysis

This section outlines the analysis by providing intermediate results paving the way towards Theorem 2, whose complete proof is provided in Appendix A. The starting point is proving the sensing map $\mathcal{Z}$ in (2) satisfies the tensor Restricted Isometry Property (T-RIP).

**Proposition 1.** *If the sensing map $\mathcal{Z}$ and ground truth $\mathcal{X}^\star$ satisfy the conditions in Theorem 2, and the tube-wise sample size has*

$$m \gtrsim \frac{r^2(n-r)n_3 \kappa^2 \mu^2}{n^2 \delta_r^2}, \tag{16}$$

*then tensor RIP holds with parameter $\delta_r \in (0,1)$ given as*

$$m(1-\delta_r)\|\mathcal{X}^\star\|_F^2 \le \|\mathcal{Z}(\mathcal{X}^\star)\|_F^2 \le m(1+\delta_r)\|\mathcal{X}^\star\|_F^2 \tag{17}$$

*with probability at least $1 - \exp\{-cr(n-r)n_3\}$.*

For the global TCS model, existing proofs used a Bernstein inequality that is uniformly applicable to all rank-$r$ tensors to prove the RIP condition. If we use the same inequality, the sample size required to establish the RIP condition given in Proposition 1 would be much larger than our current result due to ignoring the structures of the tube-wise sensing model.

Thus, compared to RIP result in Lemma 5 of [26], Proposition 1 shows that under the local TCS model, a smaller sample size $m$ suffices to guarantee $\|\mathcal{Z}(\mathcal{X}^\star)\|_F^2$ concentrates on its expected value $m\|\mathcal{X}^\star\|_F^2$. The key property that leads to this result is that $\mathcal{X}^\star$ satisfies the tensor incoherent condition (10). Motivated by this observation, we prove next that the GD iterates also satisfy the incoherence condition, which will allow us to use similar arguments to show the empirical Hessian of the non-convex loss (6) concentrates on the population Hessian under the same order of sample complexity. The linear rate of GD then follows from the fact that the population Hessian is restricted strongly convex and smooth.

To prove that GD iterates satisfy the incoherence condition implicitly, we leverage the LOO technique and define the LOO map for the tube-wise sensing map $\mathcal{Z}$, which is motivated by the matrix completion setting [8, 32].

**Definition 11.** *Given a tube-wise TCS map $\mathcal{Z}$, the LOO map $\mathcal{Z}_{-i,\cdot} : \mathbb{R}^{n \times n \times n_3} \to \mathbb{R}^{n \times n \times m}$ is identical to $\mathcal{Z}$ but excludes the measurements of all tubes belonging to the $i$-th horizontal slice of $\mathcal{X}^\star$. Correspondingly, the projection map $\mathcal{P}_{i,\cdot} : \mathbb{R}^{n \times n \times n_3} \to \mathbb{R}^{n \times n \times m}$ for $\mathcal{X}^\star$ sets its $i$-th horizontal slice to zero. That is,*

$$[\mathcal{Z}_{-i,\cdot}(\mathcal{X})]_{kj:} := \begin{cases} \mathbf{Z}_{kj} \mathcal{X}_{kj:}, & k \ne i, \\ \mathbf{0}_m, & k = i. \end{cases}; \quad [\mathcal{P}_{i,\cdot}(\mathcal{X})]_{kj:} := \begin{cases} \mathcal{X}_{kj:}, & k = i, \\ \mathbf{0}_{n_3}, & k \ne i. \end{cases} \tag{18}$$

In analogy to Definition 11, we additionally introduce the map $\mathcal{Z}_{\cdot,-j}$ that leaves out measuring the $j$-th horizontal slice and projection map $\mathcal{P}_{\cdot,j}$.

Next, we construct the auxiliary sequences as the GD iterates generated by minimizing loss functions defined based on the LOO maps. Specifically, for $1 \leq l \leq n$, we define the loss function corresponding to leaving out measuring the $i$-th horizontal slice as

$$\mathcal{L}^{(l)}\left(\mathcal{A}, \mathcal{B}\right) := \left\|\mathcal{Z}_{-l, \cdot}\left(\mathcal{A} * \mathcal{B}^{\top} - \mathcal{X}^{\star}\right)\right\|_F^2 + \left\|\mathcal{P}_{l, \cdot}\left(\mathcal{A} * \mathcal{B}^{\top} - \mathcal{X}^{\star}\right)\right\|_F^2 + \frac{m}{8}\left\|\mathcal{A}^{\top} * \mathcal{A} - \mathcal{B}^{\top} * \mathcal{B}\right\|_F^2 \tag{19}$$

with variables $\mathcal{A} \in \mathbb{R}^{n_1 \times r \times n_3}, \mathcal{B} \in \mathbb{R}^{n_2 \times r \times n_3}$. The LOO sequence is given by

$$\mathcal{A}^{t+1,l} = \mathcal{A}^{t,l} - \nabla_{\mathcal{A}}\mathcal{L}^{(l)}(\mathcal{A}^{t,l}, \mathcal{B}^{t,l}); \quad \mathcal{B}^{t+1,l} = \mathcal{B}^{t,l} - \nabla_{\mathcal{B}}\mathcal{L}^{(l)}(\mathcal{A}^{t,l}, \mathcal{B}^{t,l}). \tag{20}$$

Similarly, for $n + 1 \leq l \leq 2n$, we can also define the loss based on leaving out $(l - n)$-th lateral slice map $\mathcal{Z}_{\cdot, -(l-n)}$ and corresponding generated GD iterates as (19) and (20), respectively.

Furthermore, we define the following two orthogonal alignment tensors

$$\mathcal{R}^{t,l} := \underset{\mathcal{R} \in \mathcal{O}_r}{\arg\min} \left\| \begin{bmatrix} \mathcal{A}^{t,l} \\ \mathcal{B}^{t,l} \end{bmatrix} * \mathcal{R} - \begin{bmatrix} \mathcal{A}^{\star} \\ \mathcal{B}^{\star} \end{bmatrix} \right\|_F ; \mathcal{T}^{t,l} := \underset{\mathcal{T} \in \mathcal{O}_r}{\arg\min} \left\| \begin{bmatrix} \mathcal{A}^{t,l} \\ \mathcal{B}^{t,l} \end{bmatrix} * \mathcal{T} - \begin{bmatrix} \mathcal{A}^t \\ \mathcal{B}^t \end{bmatrix} * \mathcal{R}^t \right\|. \tag{21}$$

**Theorem 3.** *In the same setting as Theorem 2, the LOO sequences satisfy*

$$\max_{l \in [2n]} \left\| \left( \begin{bmatrix} \mathcal{A}^{t,l} \\ \mathcal{B}^{t,l} \end{bmatrix} * \mathcal{R}^{t,l} - \begin{bmatrix} \mathcal{A}^{\star} \\ \mathcal{B}^{\star} \end{bmatrix} \right)_{l::} \right\| \leq (1 - 0.1m\eta\sigma_{\min})^t \cdot 2496C \sqrt{\frac{(\mu r)^5 \kappa^9 n_3 \log^2(n \vee n_3) \sigma_{\max}}{mn^2}} \tag{22}$$

$$\max_{l \in [2n]} \left\| \begin{bmatrix} \mathcal{A}^{t,l} \\ \mathcal{B}^{t,l} \end{bmatrix} * \mathcal{T}^{t,l} - \begin{bmatrix} \mathcal{A}^t \\ \mathcal{B}^t \end{bmatrix} * \mathcal{R}^t \right\| \leq (1 - 0.1m\eta\sigma_{\min})^t \cdot 52C \sqrt{\frac{(\mu r)^4 \kappa^2 n_3 \log^2(n \vee n_3) \sigma_{\max}}{mn^2}} \tag{23}$$

*with probability at least $1 - \frac{1}{(n \vee n_3)^{10}}$.*

Assumption 2 and (22) together show that the LOO sequences satisfy the bounded incoherence condition, while (23) shows that the GD iterates are sufficiently close to the LOO sequences. Eq. (22) and (23) thereby ensure that the incoherence condition (14) holds along the GD iterates.

The rest of the proof for Theorem 2 is devoted to establishing that the non-convex loss (6) under sample complexity (12) is strongly convex and smooth along the incoherent directions satisfying (14) in a local region sufficiently close to $\mathcal{X}^{\star}$ (cf. Lemma 7 in Appendix A).

Our last result shows that Algorithm 1 generates initial points sufficiently close to $\mathcal{X}^{\star}$ and also satisfies the tensor incoherence condition. Similar to the GD stage, $2n$ auxiliary LOO variables are introduced as follows. For $1 \leq l \leq n$, we define the $l$-th LOO variables $\mathcal{A}^{0,l}, \mathcal{B}^{0,l}$ as

$$\mathcal{X}^{0,l} := \frac{1}{m} \cdot \left( \mathcal{Z}_{-l, \cdot}^c(\mathcal{Y}) + \mathcal{P}_{l, \cdot}(\mathcal{X}^{\star}) \right) \quad [\mathcal{A}^{0,l}, \mathcal{B}^{0,l}] = \texttt{T-SVD-Spec}(\mathcal{X}^{0,l}, r). \tag{24}$$

For $n + 1 \leq l \leq 2n$, we generate $(l - n)$-th LOO pair of variables in analogous to (24) by leaving $l$-th lateral slice out.

**Theorem 4.** *In the same setting as Theorem 2, if the tube-wise measurements satisfy*

$$m \geq \frac{(100C)^2 (\mu r)^4 \kappa^7 n_3 \log^2(n \vee n_3)}{n} \vee 1, \tag{25}$$

*then the initialization generated by Algorithm 1 and (24) satisfy*

$$\left\| \begin{bmatrix} \mathcal{A}^0 \\ \mathcal{B}^0 \end{bmatrix} * \mathcal{R}^0 - \begin{bmatrix} \mathcal{A}^{\star} \\ \mathcal{B}^{\star} \end{bmatrix} \right\| \leq 98C \sqrt{\frac{(\mu r)^4 \kappa^5 n_3 \log^2(n \vee n_3) \sigma_{\max}}{mn}} \tag{26}$$

$$\max_{l \in [2n]} \left\| \left( \begin{bmatrix} \mathcal{A}^0 \\ \mathcal{B}^0 \end{bmatrix} * \mathcal{R}^0 - \begin{bmatrix} \mathcal{A}^{\star} \\ \mathcal{B}^{\star} \end{bmatrix} \right)_{l::} \right\| \leq 594C \sqrt{\frac{(\mu r)^5 \kappa^5 n_3 \log^2(n \vee n_3) \sigma_{\max}}{mn^2}} \tag{27}$$

$$\max_{l \in [2n]} \left\| \left( \begin{bmatrix} \mathcal{A}^{0,l} \\ \mathcal{B}^{0,l} \end{bmatrix} * \mathcal{R}^{0,l} - \begin{bmatrix} \mathcal{A}^{\star} \\ \mathcal{B}^{\star} \end{bmatrix} \right)_{l::} \right\| \leq 126C \sqrt{\frac{(\mu r)^5 \kappa^5 n_3 \log^2(n \vee n_3) \sigma_{\max}}{mn^2}} \tag{28}$$

$$\max_{l \in [2n]} \left\| \begin{bmatrix} \boldsymbol{\mathcal{A}}^{0,l} \\ \boldsymbol{\mathcal{B}}^{0,l} \end{bmatrix} * \boldsymbol{\mathcal{T}}^{0,l} - \begin{bmatrix} \boldsymbol{\mathcal{A}}^0 \\ \boldsymbol{\mathcal{B}}^0 \end{bmatrix} * \boldsymbol{\mathcal{R}}^0 \right\| \le 52 C \sqrt{\frac{(\mu r)^4 \kappa^2 n_3 \log^2(n \vee n_3) \sigma_{\max}}{n^2 m}} \qquad (29)$$

with probability at least $1 - \frac{1}{(n \vee n_3)^{10}}$.

The idea of the proof for initialization is similar to that in [8, 32] and is provided in Appendix B.

## 5  Experimental results

We validate our theoretical findings on synthetic tensors (real-data results appear in Appendix C). First, we verify the linear convergence of Algorithm 1. Then we study its recovery performance under varying tube-wise sample complexity. Finally, we compare our method with existing works in terms of sample complexity and running time.

To validate convergence, we set $(n_1, n_2, n_3) = (100, 100, 50)$, tubal rank $r = 5$, and condition number $\kappa = 1$ for ground-truth tensor $\boldsymbol{\mathcal{X}}^\star$ that is generated from $i.i.d. \mathcal{N}(0,1)$ entries, performing an FFT along tube mode, truncating each frontal slice to rank $r$ via SVD (setting the top $r$ singular values to 1 and the rest to 0), and then applying the inverse FFT. We set the $\eta = 0.2$ and tube-wise sample complexity $m = 10$. Fig. 1 plots the recovery errors vs. iteration number in 20 Monte Carlo trials under different initializations. The recovery errors are plotted in three metrics: (1) the relative Frobenius norm error $\|\boldsymbol{\mathcal{X}}^t - \boldsymbol{\mathcal{X}}^\star\|_F / \|\boldsymbol{\mathcal{X}}^\star\|_F$; (1) the relative tube-wise error $\max_{ij} \|(\boldsymbol{\mathcal{X}}^t - \boldsymbol{\mathcal{X}}^\star)_{ij:}\|_F / \max_{ij} \|\boldsymbol{\mathcal{X}}^\star_{ij:}\|_F$; (3) the relative incoherence error $\max_{l \in [2n]} \left\| \left( \begin{bmatrix} \boldsymbol{\mathcal{A}}^t \\ \boldsymbol{\mathcal{B}}^t \end{bmatrix} * \boldsymbol{\mathcal{R}}^t - \begin{bmatrix} \boldsymbol{\mathcal{A}}^\star \\ \boldsymbol{\mathcal{B}}^\star \end{bmatrix} \right)_{l::} \right\| / \max_{l \in [2n]} \left\| \left( \begin{bmatrix} \boldsymbol{\mathcal{A}}^\star \\ \boldsymbol{\mathcal{B}}^\star \end{bmatrix} \right)_{l::} \right\|$.

Fig. 1(a) shows the linear convergence of our Algorithm 1 in all three relative error metrics. Fig. 1(b) and Fig. 1(c) show the results of the GD under random initialization $\boldsymbol{\mathcal{A}}^0, \boldsymbol{\mathcal{B}}^0 \sim \mathcal{N}(\mathbf{0}, \alpha^2 \boldsymbol{\mathcal{I}}_{n \times r \times n_3})$ with $\alpha = 1, 0.001$, respectively. For $\alpha = 1$, we select the largest $\eta = 0.0002$ to guarantee convergence. Fig. 1(b) shows large initialization cannot achieve exact recovery. For $\alpha = 0.001$, the $\eta = 0.2$ is the same as in the Fig. 1(a) setting. Fig. 1(c) shows that GD can achieve exact recovery under small initialization with the nearly same computational cost as Algorithm 1. Fig. 1(b) and Fig. 1(c) show that our spectral initialization is not necessary to achieve exact recovery, and GD with random initialization can also attain this in global convergence, but the small initialization is a necessary condition even under exact parameterization.

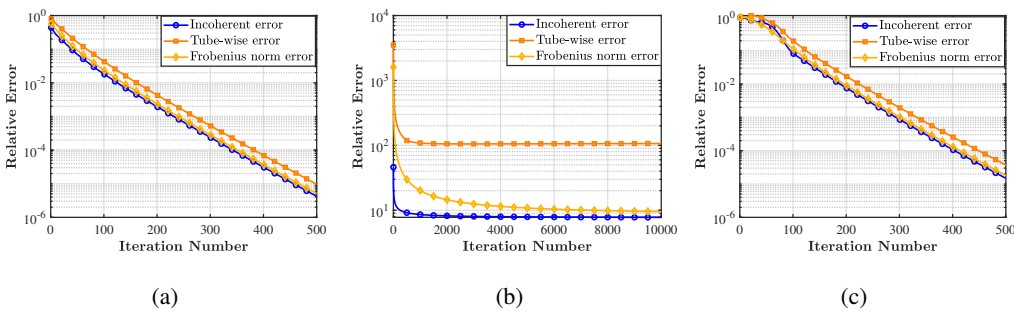

(a)           (b)           (c)

Figure 1: Relative recovery errors vs. iteration number. (a) Algorithm 1 with spectral initialization in Algorithm 2 . (b) Random initialization with $\alpha = 1$. (c) Random initialization with $\alpha = 0.001$.

To illustrate the sample efficiency of the method, we plot phase transition diagrams to examine the recovery phenomenon with varying spatial dimension $n$ and varying tube-wise sample complexity $m$. We fix the tube dimension as $n_3 = 50$ and consider the two settings: (1) $r = 5$; (2) $r = 10$. We generate ground truth $\boldsymbol{\mathcal{X}}^\star$ through the abovementioned ways. We choose $n$ in $[20 : 5 : 200]$ and $m = 1, 2, \cdots, 50$ for both $r = 5$ and $r = 10$ cases. For each $(n, m)$ pair, we simulate 10 test trials and assess a trial to be successful if the recovered $\hat{\boldsymbol{\mathcal{X}}}$ satisfies $\|\hat{\boldsymbol{\mathcal{X}}} - \boldsymbol{\mathcal{X}}^\star\|_F / \|\boldsymbol{\mathcal{X}}^\star\|_F \le 0.001$.

Fig. 2 plots the fraction of successful recovery for each $(n, m)$ pair, and white denotes perfect recovery (100%), while black denotes failure (0%). Fig. 2(a) and Fig. 2(b) plot $r = 5$ and $r = 10$ cases, respectively. In both cases, the white successful region is large, but it is noticeably larger for

Table 2: Comparison of runtime and required measurements $m$ (TCS). $m$: sample size in tensor unit; $n_3$: tube size; $n$: spatial dimension; $r$: tubal rank.

| $n_3(r)$ | $n$ | FGD ($m$ / time (s)) | TNN ($m$ / time (s)) | Schatten-2/3 TNN ($m$ / time (s)) | Alt-PGD-Min ($m$ / time (s)) | Ours ($m$ / time (s)) |
|---|---|---|---|---|---|---|
| 10 ($r$=2) | 20 | 840/7.1 | 1450/1.6 | 1020/5.1 | 43/13.8 | 3/1.1 |
| 10 ($r$=2) | 50 | 2200/175.7 | 4600/109.4 | 3100/181.7 | 45/15.6 | 2/3.7 |
| 20 ($r$=3) | 20 | 2400/86.9 | 3950/7.7 | 2750/20.7 | 130/13.8 | 7/4.5 |
| 20 ($r$=3) | 50 | 6800/442.4 | 13000/717.5 | 8500/988.2 | 135/80.4 | 3/11.6 |
| 30 ($r$=4) | 20 | 4900/181.2 | 7100/20.9 | 5250/90.0 | 250/47.3 | 12/16.1 |
| 30 ($r$=4) | 50 | 13500/1626.1 | 24000/2395.6 | 16500/2344.9 | 268/155.9 | 6/18.9 |

$r = 5$ than for $r = 10$, reflecting the greater difficulty for higher rank settings. Moreover, as the spatial dimension $n$ increases, the required tube-wise sample size $m$ for exact recovery decreases substantially, which corroborates our sample-efficient theoretical result.

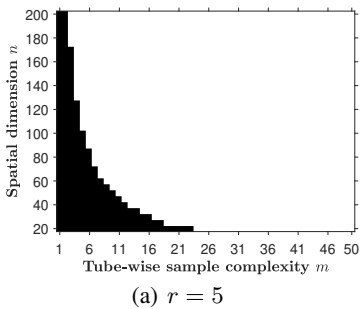
(a) $r = 5$

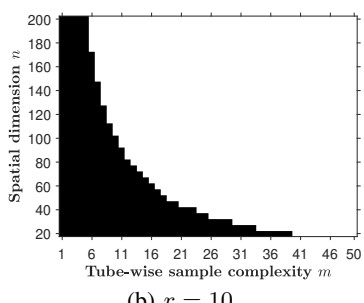
(b) $r = 10$

Figure 2: Phase transitions for tube-wise local TCS model.

To validate the sample efficiency of the proposed tube-wise model and method as given in Table 1, we compare with existing t-SVD-based TCS methods, which include global TCS methods: FGD [26], TNN [28], Schatten-2/3 TNN [20], and the latest slice-wise local TCS model Alt-PGD-Min [64]. We report results for $n = 20, 50$ only, as Schatten-2/3 TNN and TNN encounter out-of-memory issues for larger $n$ due to the huge sample size. For each method, we find the smallest number of random tensor samples $m$ (each tensor has the same size as $\mathcal{X}^\star$ and all entries generated from $i.i.d.$ standard Gaussian distribution ) that can achieve exact recovery in the sense of $\frac{\|\hat{\mathcal{X}} - \mathcal{X}^\star\|_F}{\|\mathcal{X}^\star\|_F} \leq 10^{-3}$. For the computational cost, we choose the largest step size for each method that can converge under each parameter setting.

The comparison results are summarized in the Table 2. We can observe that our tube-wise model and solving method can use both the smallest tensor samples and computing time to achieve exact recovery. This validates that our proposed model and method achieve both sample and computation efficiency. In addition, we can also find that for the fixed tube size $n_3$ and tubal rank $r$ with increasing size of spatial dimension $n$, all other methods need more samples to achieve the recovery. On the contrary, our model can use fewer samples to recover, which means that increasing the spatial dimension can reduce the tube-wise sample complexity. This result also validates our theoretical sample complexity in (12).

## 6 Conclusion and future work

In this paper, we proposed a tube-wise TCS model under the t-SVD framework. We prove that GD with spectral initialization achieves exact recovery at a linear rate. Our analysis reveals that the tube-wise sample complexity decreases linearly with respect to the spatial dimension. But several important questions remain open. The current linear-rate constant scales quadratically with the tensor condition number, motivating preconditioned or curvature-aware gradient methods to mitigate slow convergence on ill-conditioned instances. We also assume a known tubal rank and focus on third-order tensors; extending the theory to over-parameterized factorizations and to arbitrary orders under modern high-order tensor algebra is an important next direction.

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

# Appendices

## A  Proof for GD Stage

We first prove the tensor RIP result of Proposition 1 in Section A.1. Section A.2, A.3, A.4 and A.5 proves the (13), (22), (23) and (14) in Theorem 2 and Theorem 3, respectively. The final section A.6, presents the technical lemmas used throughout the main analysis. The core idea of the proof is an induction argument: assuming that equations (13), (22), (23), and (14) hold at the $t$-th iteration, we show they continue to hold at iteration $t+1$. Therefore, establishing that they are satisfied at initialization ($t = 0$) via spectral methods completes the proof.

### A.1  Proof of Proposition 1

We prove this proposition by expressing $\|\mathcal{Z}(\mathcal{X}^\star)\|_F^2 - m\|\mathcal{X}^\star\|_F^2$ as a sum of independent, mean-zero, sub-exponential random variables. We then apply an $\epsilon$-net argument together with the sub-exponential Bernstein inequality to derive the desired concentration bound. Crucially, independence of the sensing matrices in our tube-wise model and the tensor incoherence condition allow us to obtain a tight bound under the efficient sample size with high probability.

*Proof.*

$$
\begin{aligned}
\|\mathcal{Z}(\mathcal{X}^\star)\|_F^2 - m\|\mathcal{X}^\star\|_F^2 &= \sum_{i=1}^n \sum_{j=1}^n \left( \|\mathbf{Z}_{ij}\mathcal{X}^\star_{ij:}\|_F^2 - m\|\mathcal{X}^\star_{ij:}\|_F^2 \right) \\
&= \sum_{i=1}^n \sum_{j=1}^n \left( (\mathcal{X}^\star_{ij:})^T \mathbf{Z}_{ij}^T \mathbf{Z}_{ij} \mathcal{X}^\star_{ij:} - m(\mathcal{X}^\star_{ij:})^T \mathcal{X}^\star_{ij:} \right) \\
&= \sum_{i=1}^n \sum_{j=1}^n \sum_{s=1}^m \underbrace{\left( (\mathcal{X}^\star_{ij:})^T z_{ijs}^T z_{ijs} \mathcal{X}^\star_{ij:} - (\mathcal{X}^\star_{ij:})^T \mathcal{X}^\star_{ij:} \right)}_{\mathcal{T}_{ijs}} \qquad (30)
\end{aligned}
$$

It is apparent that $\{\mathcal{T}_{ijs}\}$ are independent sub-exponential random variables with zero mean and bounded sub-exponential norm $\|K_{ijs}\|_{\psi_1} \leq C\|\mathcal{X}^\star_{ij:}\|_F^2$. Thus, we can use sub-exponential Bernstein inequality with $\epsilon$-net argument to bound such deviation. Firstly, $\forall t > 0$ we have

$$
\begin{aligned}
\frac{t^2}{\sum_{i=1}^n \sum_{j=1}^n \sum_{s=1}^m \|\mathcal{T}_{ijs}\|_{\psi_1}^2} &\geq \frac{t^2}{mC^2 \sum_{i=1}^n \sum_{j=1}^n \|\mathcal{X}^\star_{ij:}\|_F^4} \\
&\geq \frac{t^2}{mC^2 \max_{ij} \|\mathcal{X}^\star_{ij:}\|_F^2 \sum_{i=1}^n \sum_{j=1}^n \|\mathcal{X}^\star_{ij:}\|_F^2} \\
&= \frac{t^2}{mC^2 \max_{ij} \|\mathcal{X}^\star_{ij:}\|_F^2 \|\mathcal{X}^\star\|_F^2} \qquad (31)
\end{aligned}
$$

Secondly, there is

$$
\frac{t}{\max_{ijk} \|\mathcal{T}_{ijs}\|_{\psi_1}} \geq \frac{t}{C \max_{ij} \|\mathcal{X}^\star_{ij:}\|_F^2}. \qquad (32)
$$

Setting $t := mC\delta_r \|\mathcal{X}^\star\|_F^2$, then there is

$$
\min \left\{ \frac{t^2}{mC^2 \max_{ij} \|\mathcal{X}^\star_{ij:}\|_F^2 \|\mathcal{X}^\star\|_F^2}, \frac{t}{C \max_{ij} \|\mathcal{X}^\star_{ij:}\|_F^2} \right\} \geq \min \left\{ \frac{m\delta_r^2 \|\mathcal{X}^\star\|_F^2}{\max_{ij} \|\mathcal{X}^\star_{ij:}\|_F^2}, \frac{m\delta_r \|\mathcal{X}^\star\|_F^2}{\max_{ij} \|\mathcal{X}^\star_{ij:}\|_F^2} \right\}
$$

$$
= \frac{m\delta_r^2 \|\mathcal{X}^\star\|_F^2}{\max_{ij} \|\mathcal{X}^\star_{ij:}\|_F^2} \overset{(i)}{\geq} \frac{m\delta_r^2 \|\mathcal{X}^\star\|_F^2}{\max_i \|\mathcal{U}^\star_{i::}\| \|\mathcal{S}^\star\|^2 \max_j \|\mathcal{V}^\star_{j::}\|} \overset{(ii)}{\geq} \frac{m\delta_r^2 \|\mathcal{X}^\star\|_F^2}{\frac{(\mu r)^2}{n^2} \|\mathcal{X}^\star\|^2} \overset{(iii)}{=} \frac{m\delta_r^2 \frac{\|\overline{\mathcal{X}^\star}\|_F^2}{n_3}}{\frac{(\mu r)^2}{n^2} \|\mathcal{X}^\star\|^2}
$$

$$\geq \frac{m\delta_r^2 \frac{n_3 r\sigma_{\min}^2(\mathcal{X}^\star)}{n_3}}{\frac{(\mu r)^2}{n^2} \|\mathcal{X}^\star\|^2} = \frac{m\delta_r^2 n^2}{\mu^2 r\kappa^2}, \tag{33}$$

where (i) is based on the tensor norm inequalities (93), (95), and the tensor incoherence condition in (10). Based on sub-exponential Bernstein inequality for fixed $\forall \mathcal{X}^\star$, we have

$$\mathbb{P}\left(\left|\|\mathcal{Z}(\mathcal{X}^\star)\|_F^2 - m\|\mathcal{X}^\star\|_F^2\right| \geq mC\delta_r\|\mathcal{X}^\star\|_F^2\right) \leq \exp\left\{-\frac{m\delta_r^2 n^2}{\mu^2 r\kappa^2}\right\} \tag{34}$$

Thus, for all the incoherence $\mathcal{X}^\star$ with tubal rank $r$, the covering number is $\mathcal{O}\left(r(2n-r)n_3\right)$, then based on the $\epsilon$-net argument in [56], there is uniform bound as

$$\mathbb{P}\left(\left|\|\mathcal{Z}(\mathcal{X}^\star)\|_F^2 - m\|\mathcal{X}^\star\|_F^2\right| \geq mC\delta_r\|\mathcal{X}^\star\|_F^2\right) \leq \exp\left\{-\frac{m\delta_r^2 n^2}{\mu^2 r\kappa^2} + r(2n-r)n_3\right\}. \tag{35}$$

Combining the tube-wise sample complexity in (16), we obtain the (17) with high probability. $\quad\square$

## A.2 Proof of (13)

Following the proof of [8], we construct the auxiliary sequence as follows:

$$\tilde{\mathcal{A}}^{t+1} := \mathcal{A}^t * \mathcal{R}^t - \eta\mathcal{Z}^c\mathcal{Z}\left(\Delta_{\mathcal{X}}^t\right) * \mathcal{B}^\star - \frac{m\eta}{2}\mathcal{A}^\star * \left(\mathcal{R}^t\right)^T \left(\left(\mathcal{A}^t\right)^T * \mathcal{A}^t - \left(\mathcal{B}^t\right)^T * \mathcal{B}^t\right) * \mathcal{R}^t \tag{36}$$

$$\tilde{\mathcal{B}}^{t+1} := \mathcal{B}^t * \mathcal{R}^t - \eta\left(\mathcal{Z}^c\mathcal{Z}\left(\Delta_{\mathcal{X}}^t\right)\right)^T * \mathcal{A}^\star - \frac{m\eta}{2}\mathcal{B}^\star * \left(\mathcal{R}^t\right)^T \left(\left(\mathcal{B}^t\right)^T * \mathcal{B}^t - \left(\mathcal{A}^t\right)^T * \mathcal{A}^t\right) * \mathcal{R}^t, \tag{37}$$

where we define the quantity $\Delta_{\mathcal{X}}^t := \mathcal{A}^t * \left(\mathcal{B}^t\right)^T - \mathcal{A}^\star * \left(\mathcal{B}^\star\right)^T$.

Next, we show that the expected auxiliary sequence contracts toward the global optimum in one gradient step. By controlling the deviation between the auxiliary sequence and its expectation via concentration bounds, we ensure the actual auxiliary iterates also exhibit this local contraction. Finally, we prove that the true GD iterates remain sufficiently close to the auxiliary sequence, and hence inherit its local contraction toward the ground-truth factors.

It is apparent that the above defined quantities expectations have

$$\mathbb{E}\left[\tilde{\mathcal{A}}^{t+1}\right] = \mathcal{A}^t * \mathcal{R}^t - \eta m\Delta_{\mathcal{X}}^t * \mathcal{B}^\star - \frac{m\eta}{2}\mathcal{A}^\star * \left(\mathcal{R}^t\right)^T \left(\left(\mathcal{A}^t\right)^T * \mathcal{A}^t - \left(\mathcal{B}^t\right)^T * \mathcal{B}^t\right) * \mathcal{R}^t, \tag{38}$$

$$\mathbb{E}\left[\tilde{\mathcal{B}}^{t+1}\right] = \mathcal{B}^t * \mathcal{R}^t - \eta m\left(\Delta_{\mathcal{X}}^t\right)^T * \mathcal{A}^\star - \frac{m\eta}{2}\mathcal{B}^\star * \left(\mathcal{R}^t\right)^T \left(\left(\mathcal{B}^t\right)^T * \mathcal{B}^t - \left(\mathcal{A}^t\right)^T * \mathcal{A}^t\right) * \mathcal{R}^t. \tag{39}$$

Based on triangle inequality and decomposition, there is

$$\left\|\begin{bmatrix}\mathcal{A}^{t+1}\\\mathcal{B}^{t+1}\end{bmatrix} * \mathcal{R}^{t+1} - \begin{bmatrix}\mathcal{A}^\star\\\mathcal{B}^\star\end{bmatrix}\right\| \leq \underbrace{\left\|\begin{bmatrix}\mathbb{E}\left[\tilde{\mathcal{A}}^{t+1}\right]\\\mathbb{E}\left[\tilde{\mathcal{B}}^{t+1}\right]\end{bmatrix} - \begin{bmatrix}\tilde{\mathcal{A}}^{t+1}\\\tilde{\mathcal{B}}^{t+1}\end{bmatrix}\right\|}_{\Pi_1} + \underbrace{\left\|\begin{bmatrix}\mathbb{E}\left[\tilde{\mathcal{A}}^{t+1}\right]\\\mathbb{E}\left[\tilde{\mathcal{B}}^{t+1}\right]\end{bmatrix} - \begin{bmatrix}\mathcal{A}^\star\\\mathcal{B}^\star\end{bmatrix}\right\|}_{\Pi_2}$$

$$+ \underbrace{\left\|\begin{bmatrix}\mathcal{A}^{t+1}\\\mathcal{B}^{t+1}\end{bmatrix} * \mathcal{R}^{t+1} - \begin{bmatrix}\tilde{\mathcal{A}}^{t+1}\\\tilde{\mathcal{B}}^{t+1}\end{bmatrix}\right\|}_{\Pi_3}. \tag{40}$$

Next, we need to bound $\Pi_1, \Pi_2, \Pi_3$, respectively.

### A.2.1 Bound for $\Pi_1$

We first give the concentration inequality for auxiliary sequence as follows.

$$\Pi_1 = \eta\left\|\begin{bmatrix}\left(\mathcal{Z}^c\mathcal{Z}\left(\Delta_{\mathcal{X}}^t\right) - m\Delta_{\mathcal{X}}^t\right) * \mathcal{B}^\star\\\left(\mathcal{Z}^c\mathcal{Z}\left(\Delta_{\mathcal{X}}^t\right) - m\Delta_{\mathcal{X}}^t\right)^T * \mathcal{A}^\star\end{bmatrix}\right\|$$

$$\leq \eta \left( \left\| \left( \boldsymbol{\mathcal{Z}}^c \boldsymbol{\mathcal{Z}} \left( \boldsymbol{\Delta}_{\boldsymbol{\mathcal{X}}}^t \right) - m \boldsymbol{\Delta}_{\boldsymbol{\mathcal{X}}}^t \right) * \boldsymbol{\mathcal{B}}^\star \right\| + \left\| \left( \boldsymbol{\mathcal{Z}}^c \boldsymbol{\mathcal{Z}} \left( \boldsymbol{\Delta}_{\boldsymbol{\mathcal{X}}}^t \right) - m \boldsymbol{\Delta}_{\boldsymbol{\mathcal{X}}}^t \right)^T * \boldsymbol{\mathcal{A}}^\star \right\| \right)$$

$$\overset{(i)}{\leq} 2\eta Cmr \sqrt{\frac{nn_3}{m}} \log(n \vee n_3) \max_{ij} \left\| \left( \boldsymbol{\Delta}_{\boldsymbol{\mathcal{X}}}^t \right)_{ij:} \right\|_F \left( \| \boldsymbol{\mathcal{B}}^\star \| + \| \boldsymbol{\mathcal{A}}^\star \| \right)$$

$$\overset{(ii)}{=} 4\eta Cmr \sqrt{\frac{nn_3 \sigma_{\max}}{m}} \log(n \vee n_3) \max_{ij} \left\| \left( \boldsymbol{\mathcal{A}}_{i::}^t * \boldsymbol{\mathcal{R}}^t - \boldsymbol{\mathcal{A}}_{i::}^\star \right) * \left( \boldsymbol{\mathcal{B}}_{j::}^\star \right)^T + \boldsymbol{\mathcal{A}}_{i::}^\star * \left( \boldsymbol{\mathcal{B}}_{j::}^t * \boldsymbol{\mathcal{R}}^t - \boldsymbol{\mathcal{B}}_{j::}^\star \right)^T \right.$$
$$\left. + \left( \boldsymbol{\mathcal{A}}_{i::}^t * \boldsymbol{\mathcal{R}}^t - \boldsymbol{\mathcal{A}}_{i::}^\star \right) * \left( \boldsymbol{\mathcal{B}}_{j::}^t * \boldsymbol{\mathcal{R}}^t - \boldsymbol{\mathcal{B}}_{j::}^\star \right)^T \right\|_F$$

$$\overset{(iii)}{\leq} 4\eta Cmr \sqrt{\frac{nn_3 \sigma_{\max}}{m}} \log(n \vee n_3) \left( \max_i \left\| \left( \boldsymbol{\mathcal{A}}^t * \boldsymbol{\mathcal{R}}^t - \boldsymbol{\mathcal{A}}^\star \right)_{i::} \right\| \max_j \left\| \boldsymbol{\mathcal{B}}_{j::}^\star \right\| \right.$$
$$\left. + \max_i \| \boldsymbol{\mathcal{A}}_{i::}^\star \| \max_j \left\| \left( \boldsymbol{\mathcal{B}}^t * \boldsymbol{\mathcal{R}}^t - \boldsymbol{\mathcal{B}}^\star \right)_{j::} \right\| + \max_i \left\| \left( \boldsymbol{\mathcal{A}}^t * \boldsymbol{\mathcal{R}}^t - \boldsymbol{\mathcal{A}}^\star \right)_{i::} \right\| \max_j \left\| \left( \boldsymbol{\mathcal{B}}^t * \boldsymbol{\mathcal{R}}^t - \boldsymbol{\mathcal{B}}^\star \right)_{j::} \right\| \right)$$

$$\overset{(iv)}{\leq} 4\eta Cmr \sqrt{\frac{\sigma_{\max} nn_3}{m}} \log(n \vee n_3) \left( 1188 \rho^t \sqrt{\frac{(\mu r)^5 \kappa^5 n_3 \log^2(n \vee n_3) \sigma_{\max}}{n^2 m}} \cdot \sqrt{\frac{\mu r \sigma_{\max}}{n}} \right.$$
$$\left. + \left( 594 \rho^t \sqrt{\frac{(\mu r)^5 \kappa^5 n_3 \log^2(n \vee n_3) \sigma_{\max}}{n^2 m}} \right)^2 \right)$$

$$\overset{(v)}{\leq} 4\eta Cmr \sqrt{\frac{\sigma_{\max} nn_3}{m}} \log(n \vee n_3) \cdot 0.3 \sqrt{\frac{\mu r}{n^2}} \sigma_{\max}$$

$$\leq 0.012 \eta m \sigma_{\min} \cdot 98 C \rho^t \sqrt{\frac{(\mu r)^3 \kappa^2 n_3 \log^2(n \vee n_3) \sigma_{\max}}{nm}}, \tag{41}$$

where (i) is based on Lemma 2 and (ii) is because

$$\left\| \left( \boldsymbol{\Delta}_{\boldsymbol{\mathcal{X}}}^t \right)_{ij:} \right\|_F = \left\| \boldsymbol{\mathcal{A}}_{i::}^t * \boldsymbol{\mathcal{R}}^t * (\boldsymbol{\mathcal{B}}_{j::}^t * \boldsymbol{\mathcal{R}}^t)^T - \boldsymbol{\mathcal{A}}_{i::}^\star * (\boldsymbol{\mathcal{B}}_{j::}^\star)^T \right\|_F$$
$$= \left\| \left( \boldsymbol{\mathcal{A}}_{i::}^t * \boldsymbol{\mathcal{R}}^t - \boldsymbol{\mathcal{A}}_{i::}^\star \right) * (\boldsymbol{\mathcal{B}}_{j::}^\star)^T + \boldsymbol{\mathcal{A}}_{i::}^\star * (\boldsymbol{\mathcal{B}}_{j::}^t - \boldsymbol{\mathcal{B}}_{j::}^\star)^T + \left( \boldsymbol{\mathcal{A}}_{i::}^t * \boldsymbol{\mathcal{R}}^t - \boldsymbol{\mathcal{A}}_{i::}^\star \right) * (\boldsymbol{\mathcal{B}}_{j::}^t - \boldsymbol{\mathcal{B}}_{j::}^\star)^T \right\|. \tag{42}$$

(iii) is due to the tensor norm inequality (95). (iv) is based on tensor incoherence condition (10) and condition in (14). The (v) is due to the tube-wise sample complexity in (12).

### A.2.2 Bound for $\Pi_2$

To bound the distance between expectation of auxiliary sequence and optimum factors, we firstly defined the following error terms

$$\boldsymbol{\Delta}_{\boldsymbol{\mathcal{A}}}^t := \boldsymbol{\mathcal{A}}^t * \boldsymbol{\mathcal{R}}^t - \boldsymbol{\mathcal{A}}^\star,$$
$$\boldsymbol{\Delta}_{\boldsymbol{\mathcal{B}}}^t := \boldsymbol{\mathcal{B}}^t * \boldsymbol{\mathcal{R}}^t - \boldsymbol{\mathcal{B}}^\star,$$
$$\boldsymbol{\Delta}^t := \begin{bmatrix} \boldsymbol{\Delta}_{\boldsymbol{\mathcal{A}}}^t \\ \boldsymbol{\Delta}_{\boldsymbol{\mathcal{B}}}^t \end{bmatrix},$$
$$\boldsymbol{\mathcal{E}}_1^t := -\boldsymbol{\Delta}_{\boldsymbol{\mathcal{A}}}^t * \left( \boldsymbol{\Delta}_{\boldsymbol{\mathcal{B}}}^t \right)^T * \boldsymbol{\mathcal{B}}^\star - \frac{1}{2} \boldsymbol{\mathcal{A}}^\star * \left( \boldsymbol{\Delta}_{\boldsymbol{\mathcal{A}}}^t \right)^T * \boldsymbol{\Delta}_{\boldsymbol{\mathcal{A}}}^t + \frac{1}{2} \boldsymbol{\mathcal{A}}^\star * \left( \boldsymbol{\Delta}_{\boldsymbol{\mathcal{B}}}^t \right)^T * \boldsymbol{\Delta}_{\boldsymbol{\mathcal{B}}}^t, \tag{43}$$
$$\boldsymbol{\mathcal{E}}_2^t := -\boldsymbol{\Delta}_{\boldsymbol{\mathcal{B}}}^t * \left( \boldsymbol{\Delta}_{\boldsymbol{\mathcal{A}}}^t \right)^T * \boldsymbol{\mathcal{A}}^\star - \frac{1}{2} \boldsymbol{\mathcal{B}}^\star * \left( \boldsymbol{\Delta}_{\boldsymbol{\mathcal{B}}}^t \right)^T * \boldsymbol{\Delta}_{\boldsymbol{\mathcal{B}}}^t + \frac{1}{2} \boldsymbol{\mathcal{B}}^\star * \left( \boldsymbol{\Delta}_{\boldsymbol{\mathcal{A}}}^t \right)^T * \boldsymbol{\Delta}_{\boldsymbol{\mathcal{A}}}^t. \tag{44}$$

The $\Pi_2$ can be bounded as follows.

$$\Pi_2 = \left\| \begin{bmatrix} \mathbb{E} \left[ \tilde{\boldsymbol{\mathcal{A}}}^{t+1} \right] \\ \mathbb{E} \left[ \tilde{\boldsymbol{\mathcal{B}}}^{t+1} \right] \end{bmatrix} - \begin{bmatrix} \boldsymbol{\mathcal{A}}^\star \\ \boldsymbol{\mathcal{B}}^\star \end{bmatrix} \right\|$$
$$= \left\| \begin{bmatrix} \boldsymbol{\Delta}_{\boldsymbol{\mathcal{A}}}^t - \eta m \boldsymbol{\Delta}_{\boldsymbol{\mathcal{X}}}^t * \boldsymbol{\mathcal{B}}^\star - \frac{\eta m}{2} \boldsymbol{\mathcal{A}}^\star * (\boldsymbol{\mathcal{R}})^T * \left( (\boldsymbol{\mathcal{A}}^t)^T * \boldsymbol{\mathcal{A}}^t - (\boldsymbol{\mathcal{B}}^t)^T * \boldsymbol{\mathcal{B}}^t \right) * \boldsymbol{\mathcal{R}}^t \\ \boldsymbol{\Delta}_{\boldsymbol{\mathcal{B}}}^t - \eta m \left( \boldsymbol{\Delta}_{\boldsymbol{\mathcal{X}}}^t \right)^T * \boldsymbol{\mathcal{A}}^\star - \frac{\eta m}{2} \boldsymbol{\mathcal{B}}^\star * (\boldsymbol{\mathcal{R}})^T * \left( (\boldsymbol{\mathcal{A}}^t)^T * \boldsymbol{\mathcal{A}}^t - (\boldsymbol{\mathcal{B}}^t)^T * \boldsymbol{\mathcal{B}}^t \right) * \boldsymbol{\mathcal{R}}^t \end{bmatrix} \right\|$$

$$
= \left\| \begin{bmatrix} \boldsymbol{\Delta}_{\mathcal{A}}^t \left( 1 - \eta m (\mathcal{B}^\star)^T * \mathcal{B}^\star \right) + \frac{\eta m}{2} \mathcal{A}^\star * \left( -(\mathcal{A}^\star)^T * \boldsymbol{\Delta}_{\mathcal{A}}^t + (\mathcal{B}^\star)^T * \boldsymbol{\Delta}_{\mathcal{B}}^t - \left( \boldsymbol{\Delta}_{\mathcal{B}}^t \right)^T * \mathcal{B}^\star - \left( \boldsymbol{\Delta}_{\mathcal{A}}^t \right)^T * \mathcal{A}^\star \right) + \eta m \boldsymbol{\mathcal{E}}_1^t \\ \boldsymbol{\Delta}_{\mathcal{B}}^t \left( 1 - \eta m (\mathcal{A}^\star)^T * \mathcal{A}^\star \right) + \frac{\eta m}{2} \mathcal{B}^\star * \left( -(\mathcal{B}^\star)^T * \boldsymbol{\Delta}_{\mathcal{B}}^t + (\mathcal{A}^\star)^T * \boldsymbol{\Delta}_{\mathcal{A}}^t - \left( \boldsymbol{\Delta}_{\mathcal{A}}^t \right)^T * \mathcal{A}^\star - \left( \boldsymbol{\Delta}_{\mathcal{B}}^t \right)^T * \mathcal{B}^\star \right) + \eta m \boldsymbol{\mathcal{E}}_2^t \end{bmatrix} \right\|
$$

$$
\overset{(i)}{=} \left\| \begin{bmatrix} \boldsymbol{\Delta}_{\mathcal{A}}^t - \eta m \boldsymbol{\Delta}_{\mathcal{A}}^t * (\mathcal{B}^\star)^T * \mathcal{B}^\star - \eta m \mathcal{A}^\star * (\mathcal{A}^\star)^T * \boldsymbol{\Delta}_{\mathcal{A}}^t + \eta m \boldsymbol{\mathcal{E}}_1^t \\ \boldsymbol{\Delta}_{\mathcal{B}}^t - \eta m \boldsymbol{\Delta}_{\mathcal{B}}^t * (\mathcal{A}^\star)^T * \mathcal{A}^\star - \eta m \mathcal{B}^\star * (\mathcal{B}^\star)^T * \boldsymbol{\Delta}_{\mathcal{B}}^t + \eta m \boldsymbol{\mathcal{E}}_2^t \end{bmatrix} \right\|
$$

$$
= \left\| \frac{1}{2} \begin{bmatrix} \boldsymbol{\Delta}_{\mathcal{A}}^t \\ \boldsymbol{\Delta}_{\mathcal{B}}^t \end{bmatrix} * \left( \mathcal{I}_r - 2\eta m (\mathcal{A}^\star)^T * \mathcal{A}^\star \right) + \frac{1}{2} \begin{bmatrix} \left( \mathcal{I}_n - 2\eta m \mathcal{A}^\star (\mathcal{A}^\star)^T \right) * \boldsymbol{\Delta}_{\mathcal{A}}^t \\ \left( \mathcal{I}_n - 2\eta m \mathcal{B}^\star (\mathcal{B}^\star)^T \right) * \boldsymbol{\Delta}_{\mathcal{B}}^t \end{bmatrix} + \eta m \begin{bmatrix} \boldsymbol{\mathcal{E}}_1^t \\ \boldsymbol{\mathcal{E}}_2^t \end{bmatrix} \right\|
$$

$$
\leq \frac{1}{2} \left\| \mathcal{I}_r - 2\eta m (\mathcal{A}^\star)^T * \mathcal{A}^\star \right\| \left\| \boldsymbol{\Delta}^t \right\| + \frac{1}{2} \left\| \mathcal{I}_{2n} - 2\eta m \begin{bmatrix} \mathcal{A}^\star * (\mathcal{A}^\star)^T & \mathcal{O} \\ \mathcal{O} & \mathcal{B}^\star * (\mathcal{B}^\star)^T \end{bmatrix} \right\| \left\| \boldsymbol{\Delta}^t \right\|
$$

$$
+ \eta m \left\| \begin{bmatrix} \boldsymbol{\mathcal{E}}_1^t \\ \boldsymbol{\mathcal{E}}_2^t \end{bmatrix} \right\|
$$

$$
\overset{(ii)}{\leq} (1 - \eta m \sigma_{\min}) \cdot \left\| \boldsymbol{\Delta}^t \right\| + 4\eta m \left\| \boldsymbol{\Delta}^t \right\|^2 \left\| \mathcal{A}^\star \right\|
$$

$$
\overset{(iii)}{\leq} (1 - \eta m \sigma_{\min}) \cdot \left\| \boldsymbol{\Delta}^t \right\| + 4m\eta \sqrt{\sigma_{\max}} \cdot 98C\rho^t \sqrt{\frac{(\mu r)^4 \kappa^5 n_3 \log^2(n \vee n_3) \sigma_{\max}}{nm}} \left\| \boldsymbol{\Delta}^t \right\|
$$

$$
= (1 - \eta m \sigma_{\min}) \cdot \left\| \boldsymbol{\Delta}^t \right\| + 4m\eta \sigma_{\min} \cdot 98C\rho^t \sqrt{\frac{(\mu r)^4 \kappa^7 n_3 \log^2(n \vee n_3) \sigma_{\max}}{nm}} \left\| \boldsymbol{\Delta}^t \right\|
$$

$$
\leq (1 - 0.9\eta m \sigma_{\min}) \cdot \left\| \boldsymbol{\Delta}^t \right\|, \tag{45}
$$

where (i) is based on Lemma 4 that $\left( \boldsymbol{\Delta}_{\mathcal{A}}^t \right)^T * \mathcal{A}^\star + \left( \boldsymbol{\Delta}_{\mathcal{B}}^t \right)^T * \mathcal{B}^\star$ is the symmetric tensor. (ii) is due to $\left\| \boldsymbol{\mathcal{E}}_1^t \right\| + \left\| \boldsymbol{\mathcal{E}}_2^t \right\| \leq 4 \left\| \boldsymbol{\Delta}^t \right\| \left\| \mathcal{A}^\star \right\|$. (iii) is based on (13), and the last inequality is because of the lower bound of tube-wise sample complexity in (12).

### A.2.3 Bounding for $\Pi_3$

To bound the distance between auxiliary sequence and true GD iterates, the proof follows the proof of claim 10 in [8]. Since we want to use Lemma 6 to bound the $\Pi_3$ as. For invoking Lemma 6, we can assign

$$
\boldsymbol{\mathcal{X}}_0 = \begin{bmatrix} \mathcal{A}^\star \\ \mathcal{B}^\star \end{bmatrix}, \quad \boldsymbol{\mathcal{X}}_1 = \begin{bmatrix} \tilde{\mathcal{A}}^{t+1} \\ \tilde{\mathcal{B}}^{t+1} \end{bmatrix}, \quad \text{and} \quad \boldsymbol{\mathcal{X}}_2 = \begin{bmatrix} \mathcal{A}^{t+1} \\ \mathcal{B}^{t+1} \end{bmatrix} * \mathcal{R}^t. \tag{46}
$$

The alignment tensor between $\boldsymbol{\mathcal{X}}_0$ and $\boldsymbol{\mathcal{X}}_2$ is actually $\mathcal{R}_1 = \left( \mathcal{R}^t \right)^T * \mathcal{R}^{t+1}$. Now we prove the alignment tensor between $\boldsymbol{\mathcal{X}}_0$ and $\boldsymbol{\mathcal{X}}_1$ is actually $\mathcal{R}_1 = \mathcal{I}_r$. Based on tensor orthogonal Procrustes in Lemma 4, this is equivalent to proving that $(\mathcal{A}^\star)^T * \tilde{\mathcal{A}}^{t+1} + (\mathcal{B}^\star)^T * \tilde{\mathcal{B}}^{t+1}$ is the symmetric positive semi-definite tensor. Based on definition in (38), there is

$$
\boldsymbol{\mathcal{X}}_0^T * \boldsymbol{\mathcal{X}}_1 = (\mathcal{A}^\star)^T * \mathcal{A}^t * \mathcal{R}^t + (\mathcal{B}^\star)^T * \mathcal{B}^t * \mathcal{R}^t - \eta (\mathcal{A}^\star)^T * \boldsymbol{\mathcal{Z}}^c \boldsymbol{\mathcal{Z}} \left( \mathcal{A}^t * (\mathcal{B}^t)^T - \mathcal{A}^\star * (\mathcal{B}^\star)^T \right) * \mathcal{B}^\star
$$

$$
- \eta (\mathcal{B}^\star)^T * \left( \boldsymbol{\mathcal{Z}}^c \boldsymbol{\mathcal{Z}} \left( \mathcal{A}^t * (\mathcal{B}^t)^T - \mathcal{A}^\star * (\mathcal{B}^\star)^T \right) \right)^T * \mathcal{A}^\star. \tag{47}
$$

Based on definition of $\mathcal{R}^t$ and Lemma 4, the $(\mathcal{A}^\star)^T * \mathcal{A}^t * \mathcal{R}^t + (\mathcal{B}^\star)^T * \mathcal{B}^t * \mathcal{R}^t$ is symmetric and semi-definite positive. Thus, $(\mathcal{A}^\star)^T * \tilde{\mathcal{A}}^{t+1} + (\mathcal{B}^\star)^T * \tilde{\mathcal{B}}^{t+1}$ is symmetric. In addition,

$$
\left\| (\mathcal{A}^\star)^T * \tilde{\mathcal{A}}^{t+1} + (\mathcal{B}^\star)^T * \tilde{\mathcal{B}}^{t+1} - \left( (\mathcal{A}^\star)^T * \mathcal{A}^\star + (\mathcal{B}^\star)^T * \mathcal{B}^\star \right) \right\| \overset{(i)}{\leq} 2\sqrt{\sigma_{\max}} (\Pi_1 + \Pi_2)
$$

$$
\overset{(ii)}{\leq} 2\sqrt{\sigma_{\max}} \left( 0.012\eta m \sigma_{\min} \cdot 98C\rho^t \sqrt{\frac{(\mu r)^3 \kappa^2 n_3 \log^2(n \vee n_3) \sigma_{\max}}{nm}} + (1 - 0.9\eta m \sigma_{\min}) \cdot \left\| \boldsymbol{\Delta}^t \right\| \right)
$$

$$
\overset{(iii)}{\leq} 2\sqrt{\sigma_{\max}} \left( 0.012\eta m \sigma_{\min} \cdot 98C\rho^t \sqrt{\frac{(\mu r)^3 \kappa^2 n_3 \log^2(n \vee n_3) \sigma_{\max}}{nm}} \right.
$$

$$+ (1 - 0.9\eta m\sigma_{\min}) \cdot 98C\rho^t \sqrt{\frac{(\mu r)^4 \kappa^5 n_3 \log^2(n \vee n_3)\sigma_{\max}}{nm}}\Bigg)$$

$$\leq 2\sigma_{\min} \cdot (1 - 0.888\eta m\sigma_{\min}) \cdot 98C\rho^t \sqrt{\frac{(\mu r)^4 \kappa^7 n_3 \log^2(n \vee n_3)}{nm}}$$

$$\leq 0.5\sigma_{\min}, \tag{48}$$

where (i) is based on definition of $\Pi_1, \Pi_2$ in (40) and (ii) is substituting results in (41) and (45). (iii) is due to condition (13), and the last inequality is due to the step size condition and tube-wise complexity in (12). Moreover, we know

$$\sigma_{\min}\left((\boldsymbol{\mathcal{A}}^\star)^T * \boldsymbol{\mathcal{A}}^\star + (\boldsymbol{\mathcal{B}}^\star)^T * \boldsymbol{\mathcal{B}}^\star\right) = \sigma_{\min}\left(\begin{bmatrix} \boldsymbol{\mathcal{A}}^\star \\ \boldsymbol{\mathcal{B}}^\star \end{bmatrix}^T * \begin{bmatrix} \boldsymbol{\mathcal{A}}^\star \\ \boldsymbol{\mathcal{B}}^\star \end{bmatrix}\right) = 2\sigma_{\min} \tag{49}$$

Thus, based on Wely's inequality for tensor, which applies matrix Wely's inequality for the block diagonal matrix in the Fourier domain, there is

$$\sigma_{\min}\left((\boldsymbol{\mathcal{A}}^\star)^T * \tilde{\boldsymbol{\mathcal{A}}}^{t+1} + (\boldsymbol{\mathcal{B}}^\star)^T * \tilde{\boldsymbol{\mathcal{B}}}^{t+1}\right) \geq 1.5\sigma_{\min}, \tag{50}$$

which means that $(\boldsymbol{\mathcal{A}}^\star)^T * \tilde{\boldsymbol{\mathcal{A}}}^{t+1} + (\boldsymbol{\mathcal{B}}^\star)^T * \tilde{\boldsymbol{\mathcal{B}}}^{t+1}$ is positive definite tensor.

To use Lemma 6, we should verify the conditions in Lemma 6. We have proved

$$\left\| \begin{bmatrix} \tilde{\boldsymbol{\mathcal{A}}}^{t+1} \\ \tilde{\boldsymbol{\mathcal{B}}}^{t+1} \end{bmatrix} - \begin{bmatrix} \boldsymbol{\mathcal{A}}^\star \\ \boldsymbol{\mathcal{B}}^\star \end{bmatrix} \right\| \left\| \begin{bmatrix} \boldsymbol{\mathcal{A}}^\star \\ \boldsymbol{\mathcal{B}}^\star \end{bmatrix} \right\| \leq \frac{\sigma_{\min}^2(\boldsymbol{\mathcal{X}}_0)}{2}. \tag{51}$$

The last thing is to verify that

$$\left\| \begin{bmatrix} \tilde{\boldsymbol{\mathcal{A}}}^{t+1} \\ \tilde{\boldsymbol{\mathcal{B}}}^{t+1} \end{bmatrix} - \begin{bmatrix} \boldsymbol{\mathcal{A}}^{t+1} \\ \boldsymbol{\mathcal{B}}^{t+1} \end{bmatrix} * \boldsymbol{\mathcal{R}}^t \right\| \left\| \begin{bmatrix} \boldsymbol{\mathcal{A}}^\star \\ \boldsymbol{\mathcal{B}}^\star \end{bmatrix} \right\| \leq \frac{\sigma_{\min}^2(\boldsymbol{\mathcal{A}}^\star)}{4} \tag{52}$$

as follows. Based on updating of (7) and (8), there is

$$\boldsymbol{\mathcal{A}}^{t+1} * \boldsymbol{\mathcal{R}}^t = \boldsymbol{\mathcal{A}}^t * \boldsymbol{\mathcal{R}}^t - \eta \boldsymbol{\mathcal{Z}}^c \boldsymbol{\mathcal{Z}}\left(\boldsymbol{\Delta}_{\boldsymbol{\mathcal{X}}}^t\right) * \boldsymbol{\mathcal{B}}^t * \boldsymbol{\mathcal{R}}^t - \frac{m\eta}{2}\left(\boldsymbol{\mathcal{A}}^t * \boldsymbol{\mathcal{R}}^t\right) * \left(\boldsymbol{\mathcal{R}}^t\right)^T * \left((\boldsymbol{\mathcal{A}}^t)^T * \boldsymbol{\mathcal{A}}^t - (\boldsymbol{\mathcal{B}}^t)^T * \boldsymbol{\mathcal{B}}^t\right) * \boldsymbol{\mathcal{R}}^t$$

$$\boldsymbol{\mathcal{B}}^{t+1} * \boldsymbol{\mathcal{R}}^t = \boldsymbol{\mathcal{B}}^t * \boldsymbol{\mathcal{R}}^t - \eta \left(\boldsymbol{\mathcal{Z}}^c \boldsymbol{\mathcal{Z}}\left(\boldsymbol{\Delta}_{\boldsymbol{\mathcal{X}}}^t\right)\right)^T * \boldsymbol{\mathcal{A}}^t * \boldsymbol{\mathcal{R}}^t - \frac{m\eta}{2}\left(\boldsymbol{\mathcal{B}}^t * \boldsymbol{\mathcal{R}}^t\right) * \left(\boldsymbol{\mathcal{R}}^t\right)^T * \left((\boldsymbol{\mathcal{B}}^t)^T * \boldsymbol{\mathcal{B}}^t - (\boldsymbol{\mathcal{A}}^t)^T * \boldsymbol{\mathcal{A}}^t\right) * \boldsymbol{\mathcal{R}}^t. \tag{53}$$

Thus,

$$\|\boldsymbol{\mathcal{X}}_1 - \boldsymbol{\mathcal{X}}_2\| = \left\| \begin{bmatrix} \boldsymbol{\mathcal{A}}^{t+1} \\ \boldsymbol{\mathcal{B}}^{t+1} \end{bmatrix} * \boldsymbol{\mathcal{R}}^t - \begin{bmatrix} \tilde{\boldsymbol{\mathcal{A}}}^{t+1} \\ \tilde{\boldsymbol{\mathcal{B}}}^{t+1} \end{bmatrix} \right\|$$

$$= \eta \left\| \begin{bmatrix} \boldsymbol{\mathcal{Z}}^c \boldsymbol{\mathcal{Z}}\left(\boldsymbol{\Delta}_{\boldsymbol{\mathcal{X}}}^t\right) * \boldsymbol{\Delta}_{\boldsymbol{\mathcal{B}}}^t + \frac{m}{2}\boldsymbol{\Delta}_{\boldsymbol{\mathcal{A}}}^t * \left(\boldsymbol{\mathcal{R}}^t\right)^T * \left((\boldsymbol{\mathcal{A}}^t)^T * \boldsymbol{\mathcal{A}}^t - (\boldsymbol{\mathcal{B}}^t)^T * \boldsymbol{\mathcal{B}}^t\right) * \boldsymbol{\mathcal{R}}^t \\ \left(\boldsymbol{\mathcal{Z}}^c \boldsymbol{\mathcal{Z}}\left(\boldsymbol{\Delta}_{\boldsymbol{\mathcal{X}}}^t\right)\right)^T * \boldsymbol{\Delta}_{\boldsymbol{\mathcal{A}}}^t + \frac{m}{2}\boldsymbol{\Delta}_{\boldsymbol{\mathcal{B}}}^t * \left(\boldsymbol{\mathcal{R}}^t\right)^T * \left((\boldsymbol{\mathcal{B}}^t)^T * \boldsymbol{\mathcal{B}}^t - (\boldsymbol{\mathcal{A}}^t)^T * \boldsymbol{\mathcal{A}}^t\right) * \boldsymbol{\mathcal{R}}^t \end{bmatrix} \right\|$$

$$\leq \eta \left(\left\|\boldsymbol{\mathcal{Z}}^c \boldsymbol{\mathcal{Z}}\left(\boldsymbol{\Delta}_{\boldsymbol{\mathcal{X}}}^t\right) * \boldsymbol{\Delta}_{\boldsymbol{\mathcal{B}}}^t\right\| + \left\|\left(\boldsymbol{\mathcal{Z}}^c \boldsymbol{\mathcal{Z}}\left(\boldsymbol{\Delta}_{\boldsymbol{\mathcal{X}}}^t\right)\right)^T * \boldsymbol{\Delta}_{\boldsymbol{\mathcal{A}}}^t\right\|\right) + m\eta \left\|(\boldsymbol{\mathcal{A}}^t)^T * \boldsymbol{\mathcal{A}}^t - (\boldsymbol{\mathcal{B}}^t)^T * \boldsymbol{\mathcal{B}}^t\right\| \left\|\boldsymbol{\Delta}^t\right\|$$

$$\leq \eta \left(\left\|(\boldsymbol{\mathcal{Z}}^c \boldsymbol{\mathcal{Z}} - m\boldsymbol{\mathcal{I}})\left(\boldsymbol{\Delta}_{\boldsymbol{\mathcal{X}}}^t\right) * \boldsymbol{\Delta}_{\boldsymbol{\mathcal{B}}}^t\right\| + m \left\|\boldsymbol{\Delta}_{\boldsymbol{\mathcal{X}}}^t * \boldsymbol{\Delta}_{\boldsymbol{\mathcal{B}}}^t\right\|\right.$$
$$\left. + \left\|\left((\boldsymbol{\mathcal{Z}}^c \boldsymbol{\mathcal{Z}} - m\boldsymbol{\mathcal{I}})\left(\boldsymbol{\Delta}_{\boldsymbol{\mathcal{X}}}^t\right)\right)^T * \boldsymbol{\Delta}_{\boldsymbol{\mathcal{A}}}^t\right\| + m \left\|\left(\boldsymbol{\Delta}_{\boldsymbol{\mathcal{X}}}^t\right)^T * \boldsymbol{\Delta}_{\boldsymbol{\mathcal{A}}}^t\right\|\right)$$
$$+ m\eta \left(2\|\boldsymbol{\mathcal{A}}^\star\| \|\boldsymbol{\Delta}_{\boldsymbol{\mathcal{A}}}^t\| + 2\|\boldsymbol{\mathcal{B}}^\star\| \|\boldsymbol{\Delta}_{\boldsymbol{\mathcal{B}}}^t\| + \|\boldsymbol{\Delta}_{\boldsymbol{\mathcal{A}}}^t\|^2 + \|\boldsymbol{\Delta}_{\boldsymbol{\mathcal{B}}}^t\|^2\right) \|\boldsymbol{\Delta}^t\|$$

$$\leq \eta \left(\left\|(\boldsymbol{\mathcal{Z}}^c \boldsymbol{\mathcal{Z}} - m\boldsymbol{\mathcal{I}})\left(\boldsymbol{\Delta}_{\boldsymbol{\mathcal{X}}}^t\right) * \boldsymbol{\Delta}_{\boldsymbol{\mathcal{B}}}^t\right\| + \left\|\left((\boldsymbol{\mathcal{Z}}^c \boldsymbol{\mathcal{Z}} - m\boldsymbol{\mathcal{I}})\left(\boldsymbol{\Delta}_{\boldsymbol{\mathcal{X}}}^t\right)\right)^T * \boldsymbol{\Delta}_{\boldsymbol{\mathcal{A}}}^t\right\| + 2m \left\|\boldsymbol{\Delta}_{\boldsymbol{\mathcal{X}}}^t\right\| \|\boldsymbol{\Delta}^t\|\right).$$
$$+ m\eta \left(4\|\boldsymbol{\mathcal{A}}^\star\| \|\boldsymbol{\Delta}^t\| + 2\|\boldsymbol{\Delta}^t\|^2\right) \|\boldsymbol{\Delta}^t\|$$

$$\overset{(i)}{\leq} \eta \left(\left\|(\boldsymbol{\mathcal{Z}}^c \boldsymbol{\mathcal{Z}} - m\boldsymbol{\mathcal{I}})\left(\boldsymbol{\Delta}_{\boldsymbol{\mathcal{X}}}^t\right) * \boldsymbol{\Delta}_{\boldsymbol{\mathcal{B}}}^t\right\| + \left\|\left((\boldsymbol{\mathcal{Z}}^c \boldsymbol{\mathcal{Z}} - m\boldsymbol{\mathcal{I}})\left(\boldsymbol{\Delta}_{\boldsymbol{\mathcal{X}}}^t\right)\right)^T * \boldsymbol{\Delta}_{\boldsymbol{\mathcal{A}}}^t\right\|\right)$$

$$+ 4m\eta \left(2 \left\|\boldsymbol{\mathcal{A}}^\star\right\| \left\|\boldsymbol{\Delta}^t\right\| + \left\|\boldsymbol{\Delta}^t\right\|^2\right) \left\|\boldsymbol{\Delta}^t\right\|$$

$$\overset{(ii)}{\leq} 4Cm\eta r \sqrt{\frac{nn_3}{m}} \log(n \vee n_3) \max_{ij} \left\|\left(\boldsymbol{\Delta}^t_{\boldsymbol{\mathcal{X}}}\right)_{ij:}\right\| \left\|\boldsymbol{\Delta}^t\right\| + 12m\eta \left\|\boldsymbol{\mathcal{A}}^\star\right\| \left\|\boldsymbol{\Delta}^t\right\|^2$$

$$\leq 4Cm\eta r \sqrt{\frac{nn_3}{m}} \log(n \vee n_3) \left\|\boldsymbol{\Delta}^t\right\| \left(\max_i \left\|\left(\boldsymbol{\Delta}^t_{\boldsymbol{\mathcal{A}}}\right)_{i::}\right\| \max_j \left\|\boldsymbol{\mathcal{B}}^\star_{j::}\right\| + \max_j \left\|\left(\boldsymbol{\Delta}^t_{\boldsymbol{\mathcal{B}}}\right)_{j::}\right\| \max_i \left\|\boldsymbol{\mathcal{A}}^\star_{i::}\right\|\right.$$

$$\left. + \max_i \left\|\left(\boldsymbol{\Delta}^t_{\boldsymbol{\mathcal{A}}}\right)_{i::}\right\| \max_j \left\|\left(\boldsymbol{\Delta}^t_{\boldsymbol{\mathcal{B}}}\right)_{j::}\right\|\right) + 12m\eta \left\|\boldsymbol{\mathcal{A}}^\star\right\| \left\|\boldsymbol{\Delta}^t\right\|^2$$

$$\leq 12Cm\eta r \sqrt{\frac{nn_3}{m}} \log(n \vee n_3) \max_i \left\|\boldsymbol{\mathcal{A}}^\star_{i::}\right\| \left\|\boldsymbol{\Delta}^t\right\| \max_l \left\|\boldsymbol{\Delta}^t_{l::}\right\| + 12m\eta \left\|\boldsymbol{\mathcal{A}}^\star\right\| \left\|\boldsymbol{\Delta}^t\right\|^2$$

$$\leq \left(12Cm\eta r \sqrt{\frac{nn_3 \log^2(n \vee n_3)}{m}} \cdot 594C\rho^t \sqrt{\frac{(\mu r)^5 \kappa^6 n_3 \log^2(n \vee n_3)\sigma_{\max}}{n^2 m}} \cdot \sqrt{\frac{\mu r \sigma_{\max}}{n}}\right.$$

$$\left. + 12\sqrt{\sigma_{\max}} m\eta \cdot 98C\rho^t \sqrt{\frac{(\mu r)^4 \kappa^5 n_3 \log^2(n \vee n_3)\sigma_{\max}}{nm}}\right) \left\|\boldsymbol{\Delta}^t\right\|$$

$$\leq 0.08 \frac{\sigma_{\min}}{\kappa} m\eta \left\|\boldsymbol{\Delta}^t\right\|, \tag{54}$$

where (i) is because

$$\left\|\boldsymbol{\Delta}^t_{\boldsymbol{\mathcal{X}}}\right\| = \left\|\boldsymbol{\mathcal{A}}^t * \left(\boldsymbol{\mathcal{B}}^t\right)^T - \boldsymbol{\mathcal{A}}^\star * \left(\boldsymbol{\mathcal{B}}^\star\right)^T\right\|$$

$$\leq \left\|\boldsymbol{\mathcal{A}}^t * \boldsymbol{\mathcal{R}}^t - \boldsymbol{\mathcal{A}}^\star\right\| \left\|\boldsymbol{\mathcal{B}}^\star\right\| + \left\|\boldsymbol{\mathcal{B}}^t * \boldsymbol{\mathcal{R}}^t - \boldsymbol{\mathcal{B}}^\star\right\| \left\|\boldsymbol{\mathcal{A}}^\star\right\| + \left\|\boldsymbol{\mathcal{A}}^t * \boldsymbol{\mathcal{R}}^t - \boldsymbol{\mathcal{A}}^\star\right\| \left\|\boldsymbol{\mathcal{B}}^t * \boldsymbol{\mathcal{R}}^t - \boldsymbol{\mathcal{B}}^\star\right\|$$

$$\leq 2 \left\|\boldsymbol{\Delta}^t\right\| \left\|\boldsymbol{\mathcal{A}}^\star\right\| + \left\|\boldsymbol{\Delta}^t\right\|^2. \tag{55}$$

(ii) is based on Lemma 2 and last two inequalities are due to the condition (13), (14) and tube-wise sample complexity (12).

Now the condition

$$\left\|\boldsymbol{\mathcal{X}}_1 - \boldsymbol{\mathcal{X}}_2\right\| \left\|\begin{bmatrix}\boldsymbol{\mathcal{A}}^\star \\ \boldsymbol{\mathcal{B}}^\star\end{bmatrix}\right\| \leq 0.08 \frac{\sigma_{\min}}{\kappa} m\eta \left\|\boldsymbol{\Delta}^t\right\| \cdot \sqrt{\sigma_{\max}}$$

$$\leq \frac{\sigma^2_{\min}(\boldsymbol{\mathcal{A}}^\star)}{4} \tag{56}$$

is satisfied. The last inequality is due to the step size condition in Theorem 2, condition (13) and tube-wise sample complexity (12). Finally, based on Lemma 6, there is

$$\Pi_3 = \left\|\begin{bmatrix}\boldsymbol{\mathcal{A}}^{t+1} \\ \boldsymbol{\mathcal{B}}^{t+1}\end{bmatrix} * \boldsymbol{\mathcal{R}}^{t+1} - \begin{bmatrix}\tilde{\boldsymbol{\mathcal{A}}}^{t+1} \\ \tilde{\boldsymbol{\mathcal{B}}}^{t+1}\end{bmatrix}\right\|$$

$$\leq 9\kappa \left\|\boldsymbol{\mathcal{X}}_1 - \boldsymbol{\mathcal{X}}_2\right\|$$

$$\leq 0.72\sigma_{\min} m\eta \left\|\boldsymbol{\Delta}^t\right\|. \tag{57}$$

Combining the bound in (41), (45) and (57), there is

$$\left\|\begin{bmatrix}\boldsymbol{\mathcal{A}}^{t+1} \\ \boldsymbol{\mathcal{B}}^{t+1}\end{bmatrix} * \boldsymbol{\mathcal{R}}^{t+1} - \begin{bmatrix}\boldsymbol{\mathcal{A}}^\star \\ \boldsymbol{\mathcal{B}}^\star\end{bmatrix}\right\| \leq \Pi_1 + \Pi_2 + \Pi_3$$

$$\leq 0.012\eta m \sigma_{\min} \cdot 98C\rho^t \sqrt{\frac{(\mu r)^3 \kappa^2 n_3 \log^2(n \vee n_3)\sigma_{\max}}{nm}}$$

$$+ (1 - 0.9\eta m \sigma_{\min}) \cdot \left\|\boldsymbol{\Delta}^t\right\| + 0.72\eta m \sigma_{\min} \left\|\boldsymbol{\Delta}^t\right\|$$

$$\overset{(i)}{\leq} (1 - 0.168\eta m \sigma_{\min}) \cdot 98C\rho^t \sqrt{\frac{(\mu r)^2 \kappa^2 n_3 \log^2(n \vee n_3)\sigma_{\max}}{nm}}$$

$$\overset{(ii)}{\leq} 98C\rho^t \sqrt{\frac{(\mu r)^4 \kappa^5 n_3 \log^2(n \vee n_3)\sigma_{\max}}{nm}}. \tag{58}$$

The inequality (i) uses (13) and (ii) is due to $\rho = (1 - 0.1\eta m\sigma_{\min})$.

### A.3  Proof of (22)

The leave-one-out sequence with respect to the $l$-th horizontal slice is generated by the gradient descent method for solving (19) as

$$\boldsymbol{\mathcal{A}}^{t+1,l} = \boldsymbol{\mathcal{A}}^{t,l} - \eta\boldsymbol{\mathcal{Z}}^c_{-l,\cdot}\boldsymbol{\mathcal{Z}}_{-l,\cdot}\left(\boldsymbol{\mathcal{A}}^{t,l} * \left(\boldsymbol{\mathcal{B}}^{t,l}\right)^T - \boldsymbol{\mathcal{X}}^\star\right) * \boldsymbol{\mathcal{B}}^{t,l} - \eta\boldsymbol{\mathcal{P}}_{l,\cdot}\left(\boldsymbol{\mathcal{A}}^{t,l} * \left(\boldsymbol{\mathcal{B}}^{t,l}\right)^T - \boldsymbol{\mathcal{X}}^\star\right) * \boldsymbol{\mathcal{B}}^{t,l}$$
$$- \frac{m\eta}{2}\boldsymbol{\mathcal{A}}^{t,l} * \left(\left(\boldsymbol{\mathcal{A}}^{t,l}\right)^T * \boldsymbol{\mathcal{A}}^{t,l} - \left(\boldsymbol{\mathcal{B}}^{t,l}\right)^T * \boldsymbol{\mathcal{B}}^{t,l}\right),$$

$$\boldsymbol{\mathcal{B}}^{t+1,l} = \boldsymbol{\mathcal{B}}^{t,l} - \eta\left(\boldsymbol{\mathcal{Z}}^c_{-l,\cdot}\boldsymbol{\mathcal{Z}}_{-l,\cdot}\left(\boldsymbol{\mathcal{A}}^{t,l} * \left(\boldsymbol{\mathcal{B}}^{t,l}\right)^T - \boldsymbol{\mathcal{X}}^\star\right)\right)^T * \boldsymbol{\mathcal{A}}^{t,l} - \eta\left(\boldsymbol{\mathcal{P}}_{l,\cdot}\left(\boldsymbol{\mathcal{A}}^{t,l} * \left(\boldsymbol{\mathcal{B}}^{t,l}\right)^T - \boldsymbol{\mathcal{X}}^\star\right)\right)^T * \boldsymbol{\mathcal{A}}^{t,l}$$
$$- \frac{m\eta}{2}\boldsymbol{\mathcal{B}}^{t,l} * \left(\left(\boldsymbol{\mathcal{B}}^{t,l}\right)^T * \boldsymbol{\mathcal{B}}^{t,l} - \left(\boldsymbol{\mathcal{A}}^{t,l}\right)^T * \boldsymbol{\mathcal{A}}^{t,l}\right). \tag{59}$$

For $n + 1 \leq l \leq 2n$, the modified loss function is

$$\min_{\substack{\boldsymbol{\mathcal{A}}\in\mathbb{R}^{n\times r\times n_3}\\ \boldsymbol{\mathcal{B}}\in\mathbb{R}^{n\times r\times n_3}}} \mathcal{L}^{(l)}(\boldsymbol{\mathcal{A}}, \boldsymbol{\mathcal{B}}) := \left\|\boldsymbol{\mathcal{Z}}_{\cdot,-(l-n)}\left(\boldsymbol{\mathcal{A}} * \boldsymbol{\mathcal{B}}^T - \boldsymbol{\mathcal{X}}^\star\right)\right\|_F^2 + \left\|\boldsymbol{\mathcal{P}}_{\cdot,l-n}\left(\boldsymbol{\mathcal{A}} * \boldsymbol{\mathcal{B}}^T - \boldsymbol{\mathcal{X}}^\star\right)\right\|_F^2$$

$$+ \frac{m}{8}\left\|\boldsymbol{\mathcal{A}}^T * \boldsymbol{\mathcal{A}} - \boldsymbol{\mathcal{B}}^T * \boldsymbol{\mathcal{B}}\right\|_F^2 \tag{60}$$

The leave-one-out sequence generated with respect to the lateral slice is the gradient sequence as

$$\boldsymbol{\mathcal{A}}^{t+1,l} = \boldsymbol{\mathcal{A}}^{t,l} - \eta\boldsymbol{\mathcal{Z}}^c_{\cdot,-(l-n)}\boldsymbol{\mathcal{Z}}_{\cdot,-(l-n)}\left(\boldsymbol{\mathcal{A}}^{t,l} * \left(\boldsymbol{\mathcal{B}}^{t,l}\right)^T - \boldsymbol{\mathcal{X}}^\star\right) * \boldsymbol{\mathcal{B}}^{t,l}$$

$$- \eta\boldsymbol{\mathcal{P}}_{\cdot,-(l-n)}\left(\boldsymbol{\mathcal{A}}^{t,l} * \left(\boldsymbol{\mathcal{B}}^{t,l}\right)^T - \boldsymbol{\mathcal{X}}^\star\right) * \boldsymbol{\mathcal{B}}^{t,l}\star$$

$$- \frac{m\eta}{2}\boldsymbol{\mathcal{A}}^{t,l} * \left(\left(\boldsymbol{\mathcal{A}}^{t,l}\right)^T * \boldsymbol{\mathcal{A}}^{t,l} - \left(\boldsymbol{\mathcal{B}}^{t,l}\right)^T * \boldsymbol{\mathcal{B}}^{t,l}\right),$$

$$\boldsymbol{\mathcal{B}}^{t+1,l} = \boldsymbol{\mathcal{B}}^{t,l} - \eta\left(\boldsymbol{\mathcal{Z}}^c_{\cdot,-(l-n)}\boldsymbol{\mathcal{Z}}_{\cdot,-(l-n)}\left(\boldsymbol{\mathcal{A}}^{t,l} * \left(\boldsymbol{\mathcal{B}}^{t,l}\right)^T - \boldsymbol{\mathcal{X}}^\star\right)\right)^T * \boldsymbol{\mathcal{A}}^{t,l}$$

$$- \eta\left(\boldsymbol{\mathcal{P}}_{\cdot,-(l-n)}\left(\boldsymbol{\mathcal{A}}^{t,l} * \left(\boldsymbol{\mathcal{B}}^{t,l}\right)^T - \boldsymbol{\mathcal{X}}^\star\right)\right)^T * \boldsymbol{\mathcal{A}}^{t,l}$$

$$- \frac{m\eta}{2}\boldsymbol{\mathcal{B}}^{t,l} * \left(\left(\boldsymbol{\mathcal{B}}^{t,l}\right)^T * \boldsymbol{\mathcal{B}}^{t,l} - \left(\boldsymbol{\mathcal{A}}^{t,l}\right)^T * \boldsymbol{\mathcal{A}}^{t,l}\right). \tag{61}$$

Without loss of generality, we consider the case where $l \in [n]$, then the formula has

$$\left(\begin{bmatrix}\boldsymbol{\mathcal{A}}^{t+1,l}\\ \boldsymbol{\mathcal{B}}^{t+1,l}\end{bmatrix} * \boldsymbol{\mathcal{R}}^{t+1,l} - \begin{bmatrix}\boldsymbol{\mathcal{A}}^\star\\ \boldsymbol{\mathcal{B}}^\star\end{bmatrix}\right)_{l::} = \underbrace{\boldsymbol{\mathcal{A}}^{t,l}_{l::} * \boldsymbol{\mathcal{R}}^{t,l} - \boldsymbol{\mathcal{A}}^\star_{l::} - \eta m\left(\boldsymbol{\mathcal{A}}^{t,l}_{l::} * \left(\boldsymbol{\mathcal{B}}^{t,l}\right)^T - \boldsymbol{\mathcal{A}}^\star_{l::} * (\boldsymbol{\mathcal{B}}^\star)^T\right) * \boldsymbol{\mathcal{B}}^{t,l} * \boldsymbol{\mathcal{R}}^{t,l}}_{\boldsymbol{\varepsilon}_1}$$

$$+ \underbrace{\left(\boldsymbol{\mathcal{A}}^{t,l}_{l::} * \boldsymbol{\mathcal{R}}^{t,l} - \boldsymbol{\mathcal{A}}^\star_{l::} - \eta m\left(\boldsymbol{\mathcal{A}}^{t,l}_{l::} * \left(\boldsymbol{\mathcal{B}}^{t,l}\right)^T - \boldsymbol{\mathcal{A}}^\star_{l::} * (\boldsymbol{\mathcal{B}}^\star)^T\right) * \boldsymbol{\mathcal{B}}^{t,l} * \boldsymbol{\mathcal{R}}^{t,l}\right) * \left(\left(\boldsymbol{\mathcal{R}}^{t,l}\right)^{-1} * \boldsymbol{\mathcal{R}}^{t+1,l} - \boldsymbol{\mathcal{I}}_r\right)}_{\boldsymbol{\varepsilon}_2}$$

$$\underbrace{- \frac{\eta m}{2}\boldsymbol{\mathcal{A}}^{t,l}_{l::} * \left(\left(\boldsymbol{\mathcal{A}}^{t,l}\right)^T * \boldsymbol{\mathcal{A}}^{t,l} - \left(\boldsymbol{\mathcal{B}}^{t,l}\right)^T * \boldsymbol{\mathcal{B}}^{t,l}\right) * \boldsymbol{\mathcal{R}}^{t+1,l}}_{\boldsymbol{\varepsilon}_3}. \tag{62}$$

For bounding the spectral norm of $\boldsymbol{\mathcal{E}}_1$, with defining the notation $\boldsymbol{\Delta}_{\boldsymbol{\mathcal{A}}}^{t,l} := \boldsymbol{\mathcal{A}}^{t,l} * \boldsymbol{\mathcal{R}}^{t,l} - \boldsymbol{\mathcal{A}}^\star, \boldsymbol{\Delta}_{\boldsymbol{\mathcal{B}}}^{t,l} :=$
$\boldsymbol{\mathcal{B}}^{t,l} * \boldsymbol{\mathcal{R}}^{t,l} - \boldsymbol{\mathcal{B}}^\star$, then there is

$$\|\boldsymbol{\mathcal{E}}_1\| = \left\| \left( \boldsymbol{\mathcal{I}}_r - m\eta \left(\boldsymbol{\mathcal{B}}^\star\right)^T * \boldsymbol{\mathcal{B}}^\star \right) * \left(\boldsymbol{\Delta}_{\boldsymbol{\mathcal{A}}}^{t,l}\right)_{l::} - m\eta \left( \left(\boldsymbol{\Delta}_{\boldsymbol{\mathcal{A}}}^{t,l}\right)_{l::} * \left(\boldsymbol{\Delta}_{\boldsymbol{\mathcal{B}}}^{t,l}\right)^T + \boldsymbol{\mathcal{A}}_{l::}^\star * \left(\boldsymbol{\Delta}_{\boldsymbol{\mathcal{B}}}^{t,l}\right)^T \right) * \boldsymbol{\mathcal{B}}^{t,l} * \boldsymbol{\mathcal{R}}^{t,l} \right.$$

$$\left. - \eta m \left(\boldsymbol{\Delta}_{\boldsymbol{\mathcal{A}}}^{t,l}\right)_{l::} * (\boldsymbol{\mathcal{B}}^\star)^T * \boldsymbol{\Delta}_{\boldsymbol{\mathcal{B}}}^{t,l} \right\|$$

$$\leq \left\| \boldsymbol{\mathcal{I}}_r - m\eta \left(\boldsymbol{\mathcal{B}}^\star\right)^T * \boldsymbol{\mathcal{B}}^\star \right\| \left\| \left(\boldsymbol{\Delta}_{\boldsymbol{\mathcal{A}}}^{t,l}\right)_{l::} \right\| + \eta m \left\| \left(\boldsymbol{\Delta}_{\boldsymbol{\mathcal{A}}}^{t,l}\right)_{l::} \right\| \left\| \boldsymbol{\Delta}_{\boldsymbol{\mathcal{B}}}^{t,l} \right\| \left( \left\| \boldsymbol{\mathcal{B}}^{t,l} \right\| + \|\boldsymbol{\mathcal{B}}^\star\| \right)$$

$$+ \eta m \left\| \boldsymbol{\mathcal{A}}_{l::}^\star \right\| \left\| \boldsymbol{\Delta}_{\boldsymbol{\mathcal{B}}}^{t,l} \right\| \left\| \boldsymbol{\mathcal{B}}^{t,l} \right\|$$

$$\leq \left\| \boldsymbol{\mathcal{I}}_r - m\eta \left(\boldsymbol{\mathcal{B}}^\star\right)^T * \boldsymbol{\mathcal{B}}^\star \right\| \left\| \left(\boldsymbol{\Delta}_{\boldsymbol{\mathcal{A}}}^{t,l}\right)_{l::} \right\| + \eta m \left\| \left(\boldsymbol{\Delta}_{\boldsymbol{\mathcal{A}}}^{t,l}\right)_{l::} \right\| \left\| \boldsymbol{\Delta}_{\boldsymbol{\mathcal{B}}}^{t,l} \right\| \left( \left\| \boldsymbol{\Delta}_{\boldsymbol{\mathcal{B}}}^{t,l} \right\| + 2\|\boldsymbol{\mathcal{B}}^\star\| \right)$$

$$+ \eta m \left\| \boldsymbol{\mathcal{A}}_{l::}^\star \right\| \left\| \boldsymbol{\Delta}_{\boldsymbol{\mathcal{B}}}^{t,l} \right\| \left( \left\| \boldsymbol{\Delta}_{\boldsymbol{\mathcal{B}}}^{t,l} \right\| + \|\boldsymbol{\mathcal{B}}^\star\| \right). \tag{63}$$

Now comes to bound the $\left\| \boldsymbol{\Delta}_{\boldsymbol{\mathcal{B}}}^{t,l} \right\|$ as follows

$$\left\| \boldsymbol{\Delta}_{\boldsymbol{\mathcal{B}}}^{t,l} \right\| \leq \left\| \boldsymbol{\Delta}^{t,l} \right\| = \left\| \begin{bmatrix} \boldsymbol{\mathcal{A}}^{t,l} \\ \boldsymbol{\mathcal{B}}^{t,l} \end{bmatrix} * \boldsymbol{\mathcal{R}}^{t,l} - \begin{bmatrix} \boldsymbol{\mathcal{A}}^\star \\ \boldsymbol{\mathcal{B}}^\star \end{bmatrix} \right\|$$

$$\leq \left\| \begin{bmatrix} \boldsymbol{\mathcal{A}}^{t,l} \\ \boldsymbol{\mathcal{B}}^{t,l} \end{bmatrix} * \boldsymbol{\mathcal{R}}^{t,l} - \begin{bmatrix} \boldsymbol{\mathcal{A}}^t \\ \boldsymbol{\mathcal{B}}^t \end{bmatrix} * \boldsymbol{\mathcal{R}}^t \right\| + \left\| \begin{bmatrix} \boldsymbol{\mathcal{A}}^t \\ \boldsymbol{\mathcal{B}}^t \end{bmatrix} * \boldsymbol{\mathcal{R}}^t - \begin{bmatrix} \boldsymbol{\mathcal{A}}^\star \\ \boldsymbol{\mathcal{B}}^\star \end{bmatrix} \right\|$$

$$\overset{(i)}{\leq} 9\kappa \left\| \begin{bmatrix} \boldsymbol{\mathcal{A}}^{t,l} \\ \boldsymbol{\mathcal{B}}^{t,l} \end{bmatrix} * \boldsymbol{\mathcal{T}}^{t,l} - \begin{bmatrix} \boldsymbol{\mathcal{A}}^t \\ \boldsymbol{\mathcal{B}}^t \end{bmatrix} * \boldsymbol{\mathcal{R}}^t \right\| + \left\| \begin{bmatrix} \boldsymbol{\mathcal{A}}^t \\ \boldsymbol{\mathcal{B}}^t \end{bmatrix} * \boldsymbol{\mathcal{R}}^t - \begin{bmatrix} \boldsymbol{\mathcal{A}}^\star \\ \boldsymbol{\mathcal{B}}^\star \end{bmatrix} \right\|$$

$$\overset{(ii)}{\leq} 9\kappa \cdot 52C\rho^t \sqrt{\frac{(\mu r)^4 \kappa^2 n_3 \log^2(n \vee n_3)\sigma_{\max}}{n^2 m}} + 98C\rho^t \sqrt{\frac{(\mu r)^4 \kappa^5 n_3 \log^2(n \vee n_3)\sigma_{\max}}{nm}}$$

$$\leq 12\kappa \cdot 52C\rho^t \sqrt{\frac{(\mu r)^4 \kappa^5 n_3 \log^2(n \vee n_3)\sigma_{\max}}{nm}}. \tag{64}$$

(i) reuses the Lemma 6 to bound the first term of RHS of the above inequality as $\boldsymbol{\mathcal{X}}_0 = \begin{bmatrix} \boldsymbol{\mathcal{A}}^\star \\ \boldsymbol{\mathcal{B}}^\star \end{bmatrix}$, $\boldsymbol{\mathcal{X}}_1 = \begin{bmatrix} \boldsymbol{\mathcal{A}}^t \\ \boldsymbol{\mathcal{B}}^t \end{bmatrix} * \boldsymbol{\mathcal{R}}^t$, and $\boldsymbol{\mathcal{X}}_2 = \begin{bmatrix} \boldsymbol{\mathcal{A}}^{t,l} \\ \boldsymbol{\mathcal{B}}^{t,l} \end{bmatrix} * \boldsymbol{\mathcal{T}}^{t,l}$. It is apparent that $\|\boldsymbol{\mathcal{X}}_1 - \boldsymbol{\mathcal{X}}_0\| \|\boldsymbol{\mathcal{X}}_0\| \leq \frac{\sigma_{\max}^2(\boldsymbol{\mathcal{A}}^\star)}{2}$ and $\|\boldsymbol{\mathcal{X}}_1 - \boldsymbol{\mathcal{X}}_2\| \|\boldsymbol{\mathcal{X}}_0\| \leq \frac{\sigma_{\max}^2(\boldsymbol{\mathcal{A}}^\star)}{4}$ based on (13), (23) and tube-wise sample complexity (12). (ii) is due to (13) and (23). Substituting above inequality into (63) would have

$$\|\boldsymbol{\mathcal{E}}_1\| \leq (1 - \eta m \sigma_{\min}) \left\| \left(\boldsymbol{\Delta}_{\boldsymbol{\mathcal{A}}}^{t,l}\right)_{l::} \right\| + \eta m \left\| \left(\boldsymbol{\Delta}_{\boldsymbol{\mathcal{A}}}^{t,l}\right)_{l::} \right\| \cdot 12\kappa \cdot 52C\rho^t \sqrt{\frac{(\mu r)^4 \kappa^5 n_3 \log^2(n \vee n_3)\sigma_{\max}}{nm}}$$

$$\cdot \left( 2\sqrt{\sigma_{\max}} + 12\kappa \cdot 52C\rho^t \sqrt{\frac{(\mu r)^4 \kappa^5 n_3 \log^2(n \vee n_3)\sigma_{\max}}{nm}} \right)$$

$$+ \eta m \sqrt{\frac{\mu r \sigma_{\max}}{n}} \cdot 12\kappa \cdot 52C\rho^t \sqrt{\frac{(\mu r)^4 \kappa^5 n_3 \log^2(n \vee n_3)\sigma_{\max}}{nm}}$$

$$\cdot \left( \sqrt{\sigma_{\max}} + 12\kappa \cdot 52C\rho^t \sqrt{\frac{(\mu r)^4 \kappa^5 n_3 \log^2(n \vee n_3)\sigma_{\max}}{nm}} \right)$$

$$\overset{(i)}{\leq} (1 - \eta m \sigma_{\min}) \left\| \left(\boldsymbol{\Delta}_{\boldsymbol{\mathcal{A}}}^{t,l}\right)_{l::} \right\| + 1872C\eta m \sigma_{\max}\kappa\rho^t \sqrt{\frac{(\mu r)^4 \kappa^5 n_3 \log^2(n \vee n_3)}{nm}} \left\| \left(\boldsymbol{\Delta}_{\boldsymbol{\mathcal{A}}}^{t,l}\right)_{l::} \right\|$$

$$+ \eta m \sigma_{\max} \sqrt{\frac{\mu r}{n}} \cdot 1248C\kappa\rho^t \sqrt{\frac{(\mu r)^4 \kappa^5 n_3 \log^2(n \vee n_3)\sigma_{\max}}{nm}}$$

$$\overset{(ii)}{\leq} (1 - 0.85\eta m\sigma_{\min}) \left\| \left(\mathbf{\Delta}_{\mathcal{A}}^{t,l}\right)_{l::} \right\| + 0.5\eta m\sigma_{\min} \cdot 2496C\rho^t \sqrt{\frac{(\mu r)^5 \kappa^9 n_3 \log^2(n \vee n_3)\sigma_{\max}}{n^2 m}}$$

$$\overset{(iii)}{\leq} (1 - 0.6\eta m\sigma_{\min}) \cdot 2496C\rho^t \sqrt{\frac{(\mu r)^5 \kappa^9 n_3 \log^2(n \vee n_3)\sigma_{\max}}{n^2 m}}$$

$$\leq \sqrt{\frac{\mu r \sigma_{\max}}{n}}, \tag{65}$$

where the (i) and (ii) are due to (12). (iii) is due to (22) and step size condition in Theorem 2.

For the $\mathcal{E}_2$, we have the similar bound as (48) in [8] as

$$\|\mathcal{E}_2\| \leq \left\| \mathcal{E}_1 + (\mathcal{A}^\star)_{l::} \right\| \left\| \left(\mathcal{R}^{t,l}\right)^{-1} * \mathcal{R}^{t+1,l} - \mathcal{I}_r \right\|$$

$$\overset{(i)}{\leq} 2\sqrt{\frac{\mu r \sigma_{\max}}{n}} \left\| \left(\mathcal{R}^{t,l}\right)^{-1} * \mathcal{R}^{t+1,l} - \mathcal{I}_r \right\|$$

$$\leq 2\sqrt{\frac{\mu r \sigma_{\max}}{n}} \cdot 2 \times 1250^2 m\eta C^2 \rho^{2t} \left( \sqrt{\frac{(\mu r)^4 \kappa^8 n_3 \log^2(n \vee n_3)\sigma_{\max}}{nm}} \right)^2 \tag{66}$$

where the (i) is due to (65) and the last inequality is due to Lemma 8.

For the $\mathcal{E}_3$, there is

$$\|\mathcal{E}_3\| \leq \frac{\eta m}{2} \left\| \left(\mathcal{A}^{t,l}\right)_{l::} \right\| \left( 2\left\|\mathbf{\Delta}_{\mathcal{A}}^{t,l}\right\| \|\mathcal{A}^\star\| + 2\left\|\mathbf{\Delta}_{\mathcal{B}}^{t,l}\right\| \|\mathcal{B}^\star\| + \left\|\mathbf{\Delta}_{\mathcal{A}}^{t,l}\right\|^2 + \left\|\mathbf{\Delta}_{\mathcal{B}}^{t,l}\right\|^2 \right)$$

$$\leq \frac{\eta m}{2} \left( \left\| \left(\mathbf{\Delta}^{t,l}\right)_{l::} \right\| + \|\mathcal{A}_{l::}^\star\| \right) \cdot \left( 4\left\|\mathbf{\Delta}^{t,l}\right\| \|\mathcal{A}^\star\| + 2\left\|\mathbf{\Delta}^{t,l}\right\|^2 \right)$$

$$\overset{(i)}{\leq} \frac{\eta m}{2} \left( 2496C\sqrt{\frac{(\mu r)^5 \kappa^9 n_3 \log^2(n \vee n_3)\sigma_{\max}}{n^2 m}} + \sqrt{\frac{\mu r \sigma_{\max}}{n}} \right) \cdot 6\sqrt{\sigma_{\max}} \left\|\mathbf{\Delta}^{t,l}\right\|$$

$$\overset{(ii)}{\leq} 6\eta m\sigma_{\max} \sqrt{\frac{\mu r}{n}} \left\|\mathbf{\Delta}^{t,l}\right\|$$

$$\leq 1872C m\eta\sigma_{\min}\rho^t \sqrt{\frac{(\mu r)^5 \kappa^9 n_3 \log^2(n \vee n_3)\sigma_{\max}}{n^2 m}}, \tag{67}$$

where (i) uses the result in (22) and tensor incoherence condition in (10). (ii) and last inequality use the tube-wise sample complexity and (64), respectively.

Finally, combing the (65), (66) and (67) would induce

$$\left\| \left( \begin{bmatrix} \mathcal{A}^{t+1,l} \\ \mathcal{B}^{t+1,l} \end{bmatrix} * \mathcal{R}^{t+1,l} - \begin{bmatrix} \mathcal{A}^\star \\ \mathcal{B}^\star \end{bmatrix} \right)_{l::} \right\| \leq (1 - 0.6\eta m\sigma_{\min}) \cdot 2496C\rho^t \sqrt{\frac{(\mu r)^5 \kappa^9 n_3 \log^2(n \vee n_3)\sigma_{\max}}{n^2 m}}$$

$$+ (2500C)^2 m\eta\sigma_{\min}\rho^t \sqrt{\frac{(\mu r)^4 \kappa^9 n_3 \log^2(n \vee n_3)}{nm}}$$

$$\cdot \sqrt{\frac{(\mu r)^5 \kappa^9 n_3 \log^2(n \vee n_3)\sigma_{\max}}{n^2 m}}$$

$$+ 1872C m\eta\sigma_{\min}\rho^t \sqrt{\frac{(\mu r)^5 \kappa^9 n_3 \log^2(n \vee n_3)\sigma_{\max}}{n^2 m}}$$

$$\leq 2496C\rho^{t+1} \sqrt{\frac{(\mu r)^5 \kappa^9 n_3 \log^2(n \vee n_3)\sigma_{\max}}{n^2 m}}. \tag{68}$$

The last inequality is due to the tube-wise sample complexity in (12).

## A.4 Proof of (23)

Based on definition of $\mathcal{T}^{t,l}$ in (21), there is

$$\left\| \begin{bmatrix} \mathcal{A}^{t+1} \\ \mathcal{B}^{t+1} \end{bmatrix} * \mathcal{R}^{t+1} - \begin{bmatrix} \mathcal{A}^{t+1,l} \\ \mathcal{B}^{t+1,l} \end{bmatrix} * \mathcal{T}^{t+1,l} \right\| \leq \left\| \begin{bmatrix} \mathcal{A}^{t+1} \\ \mathcal{B}^{t+1} \end{bmatrix} * \mathcal{R}^{t} - \begin{bmatrix} \mathcal{A}^{t+1,l} \\ \mathcal{B}^{t+1,l} \end{bmatrix} * \mathcal{T}^{t,l} \right\| \tag{69}$$

Without loss of generality, we can consider the leave-one-out sequence under $l \in [n]$. Based on (7), (8) and (59), there is decomposition

$$\begin{bmatrix} \mathcal{A}^{t+1} \\ \mathcal{B}^{t+1} \end{bmatrix} * \mathcal{R}^{t} - \begin{bmatrix} \mathcal{A}^{t+1,l} \\ \mathcal{B}^{t+1,l} \end{bmatrix} * \mathcal{T}^{t,l} = \mathcal{E}_1 + \eta \begin{bmatrix} \mathcal{E}_2 \\ \mathcal{E}_3 \end{bmatrix}, \tag{70}$$

where the $\mathcal{E}_1, \mathcal{E}_2$ and $\mathcal{E}_3$ are redefined as

$$\mathcal{E}_1 := \left( \begin{bmatrix} \mathcal{A}^t \\ \mathcal{B}^t \end{bmatrix} - \eta \nabla \mathcal{L} \left( \mathcal{A}^t, \mathcal{B}^t \right) \right) * \mathcal{R}^t - \left( \begin{bmatrix} \mathcal{A}^{t,l} \\ \mathcal{B}^{t,l} \end{bmatrix} - \eta \nabla \mathcal{L} \left( \mathcal{A}^{t,l}, \mathcal{B}^{t,l} \right) \right) * \mathcal{T}^{t,l} \tag{71}$$

$$\mathcal{E}_2 := \left[ m \mathcal{P}_{l,\cdot} \left( \mathcal{A}^{t,l} * \left( \mathcal{B}^{t,l} \right)^T - \mathcal{A}^\star * (\mathcal{B}^\star)^T \right) - \mathcal{Z}^c_{l,\cdot} \mathcal{Z}_{l,\cdot} \left( \mathcal{A}^{t,l} * \left( \mathcal{B}^{t,l} \right)^T - \mathcal{A}^\star * (\mathcal{B}^\star)^T \right) \right] * \mathcal{B}^{t,l} * \mathcal{T}^{t,l} \tag{72}$$

$$\mathcal{E}_3 := \left[ m \mathcal{P}_{l,\cdot} \left( \mathcal{A}^{t,l} * \left( \mathcal{B}^{t,l} \right)^T - \mathcal{A}^\star * (\mathcal{B}^\star)^T \right) - \mathcal{Z}^c_{l,\cdot} \mathcal{Z}_{l,\cdot} \left( \mathcal{A}^{t,l} * \left( \mathcal{B}^{t,l} \right)^T - \mathcal{A}^\star * (\mathcal{B}^\star)^T \right) \right]^T * \mathcal{A}^{t,l} * \mathcal{T}^{t,l}, \tag{73}$$

where $\mathcal{Z}_{i,\cdot}, i \in [n_1]$ denotes the tube-wise sensing operator that only tubes in $i$-th horizontal slice are sensed based on $\mathcal{Z}$ and all other tubes are not sensed, i.e., for given $i \in [n]$, there is

$$[\mathcal{Z}_{i,\cdot} (\boldsymbol{\mathcal{X}})]_{kj:} := \begin{cases} \boldsymbol{Z}_{kj} \boldsymbol{\mathcal{X}}_{kj:}, & k = i, \forall j \in [n], \\ \boldsymbol{0}_m, & k \neq i, \forall j \in [n]. \end{cases} \tag{74}$$

Now we bound each of the error terms $\|\mathcal{E}_1\|$, $\|\mathcal{E}_2\|$, and $\|\mathcal{E}_3\|$ in turn. For $\|\mathcal{E}_1\|$, we apply Lemma 7, which provides restricted strong convexity and smoothness of the non-convex loss, to establish a local contraction of the distance between the auxiliary sequence and the true GD iterates. The terms $\|\mathcal{E}_2\|$ and $\|\mathcal{E}_3\|$ are then controlled by our concentration bounds, ensuring they remain sufficiently small.

For $\mathcal{E}_1$, there exists $\tau \in [0, 1]$ such that

$$\|\mathcal{E}_1\| = \left\| \left( \mathcal{I} - \eta \nabla^2 \mathcal{L} \left( \tau \mathcal{A}^{t,l} * \mathcal{T}^{t,l} + (1-\tau) \mathcal{A}^t * \mathcal{R}^t, \tau \mathcal{B}^{t,l} * \mathcal{T}^{t,l} + (1-\tau) \mathcal{B}^t * \mathcal{R}^t \right) \right) \begin{bmatrix} \mathcal{A}^{t,l} * \mathcal{T}^{t,l} - \mathcal{A}^t * \mathcal{R}^t \\ \mathcal{B}^{t,l} * \mathcal{T}^{t,l} - \mathcal{B}^t * \mathcal{R}^t \end{bmatrix} \right\|$$

$$\leq \left( 1 - \frac{m \eta \sigma_{\min}}{5} \right) \left\| \begin{bmatrix} \mathcal{A}^{t,l} * \mathcal{T}^{t,l} - \mathcal{A}^t * \mathcal{R}^t \\ \mathcal{B}^{t,l} * \mathcal{T}^{t,l} - \mathcal{B}^t * \mathcal{R}^t \end{bmatrix} \right\|. \tag{75}$$

The inequality is based on Lemma 7 as follows. Based on (79), (14) and tube-wise sample complexity (12), we can conclude that

$$\max_l \left\| \begin{bmatrix} \tau \mathcal{A}^{t,l} * \mathcal{T}^{t,l} + (1-\tau) \mathcal{A}^t * \mathcal{R}^t - \mathcal{U}^\star \\ \tau \mathcal{B}^{t,l} * \mathcal{T}^{t,l} + (1-\tau) \mathcal{B}^t * \mathcal{R}^t - \mathcal{V}^\star \end{bmatrix}_{l::} \right\| \leq \tau \max_l \left\| \begin{bmatrix} \mathcal{A}^{t,l} * \mathcal{T}^{t,l} - \mathcal{U}^\star \\ \mathcal{B}^{t,l} * \mathcal{T}^{t,l} - \mathcal{V}^\star \end{bmatrix}_{l::} \right\|$$

$$+ (1-\tau) \max_l \left\| \begin{bmatrix} \mathcal{A}^t * \mathcal{R}^t - \mathcal{U}^\star \\ \mathcal{B}^t * \mathcal{R}^t - \mathcal{V}^\star \end{bmatrix}_{l::} \right\|$$

$$\leq \frac{\sqrt{\sigma_{\max}}}{500 \kappa \sqrt{2n}}$$

$$\left\| \begin{bmatrix} \mathcal{A}^t * \mathcal{R}^t - \mathcal{U}^\star \\ \mathcal{B}^t * \mathcal{R}^t - \mathcal{V}^\star \end{bmatrix} \right\| \leq \sqrt{\sum_{i=1}^{2n} \left\| \left( \begin{bmatrix} \mathcal{A}^t * \mathcal{R}^t - \mathcal{U}^\star \\ \mathcal{B}^t * \mathcal{R}^t - \mathcal{V}^\star \end{bmatrix} \right)_{i::} \right\|^2}$$

$$\leq \frac{\sqrt{\sigma_{\max}}}{500\kappa}, \tag{76}$$

where the third inequality is due to norm inequality (92). The above two conditions are satisfied in Lemma 7.

For $\mathcal{E}_2, \mathcal{E}_3$, we have

$$\|\mathcal{E}_2\| = \left\| \sum_{j=1}^{n} \left( \boldsymbol{Z}_{lj}^T \boldsymbol{Z}_{lj} - m\boldsymbol{I} \right) \left( \boldsymbol{\Delta}_{\boldsymbol{\mathcal{X}}}^{t,l} \right)_{lj:} * \boldsymbol{\mathcal{B}}_{j::}^{t,l} \right\|,$$

$$\|\mathcal{E}_3\| = \left\| \begin{bmatrix} \left( \boldsymbol{Z}_{l1}^T \boldsymbol{Z}_{l1} - m\boldsymbol{I} \right) \left( \boldsymbol{\Delta}_{\boldsymbol{\mathcal{X}}}^{t,l} \right)_{l1:} \\ \left( \boldsymbol{Z}_{l2}^T \boldsymbol{Z}_{l2} - m\boldsymbol{I} \right) \left( \boldsymbol{\Delta}_{\boldsymbol{\mathcal{X}}}^{t,l} \right)_{l2:} \\ \vdots \\ \left( \boldsymbol{Z}_{ln}^T \boldsymbol{Z}_{ln} - m\boldsymbol{I} \right) \left( \boldsymbol{\Delta}_{\boldsymbol{\mathcal{X}}}^{t,l} \right)_{ln:} \end{bmatrix} * \boldsymbol{\mathcal{A}}_{l::}^{t,l} \right\|$$

$$\leq \left\| \begin{bmatrix} \left( \boldsymbol{Z}_{l1}^T \boldsymbol{Z}_{l1} - m\boldsymbol{I} \right) \left( \boldsymbol{\Delta}_{\boldsymbol{\mathcal{X}}}^{t,l} \right)_{l1:} \\ \left( \boldsymbol{Z}_{l2}^T \boldsymbol{Z}_{l2} - m\boldsymbol{I} \right) \left( \boldsymbol{\Delta}_{\boldsymbol{\mathcal{X}}}^{t,l} \right)_{l2:} \\ \vdots \\ \left( \boldsymbol{Z}_{ln}^T \boldsymbol{Z}_{ln} - m\boldsymbol{I} \right) \left( \boldsymbol{\Delta}_{\boldsymbol{\mathcal{X}}}^{t,l} \right)_{ln:} \end{bmatrix} \right\| * \left\| \boldsymbol{\mathcal{A}}_{l::}^{t,l} \right\| \tag{77}$$

where $\boldsymbol{\Delta}_{\boldsymbol{\mathcal{X}}}^{t,l} := \boldsymbol{\mathcal{A}}^{t,l} * \left( \boldsymbol{\mathcal{B}}^{t,l} \right)^T - \boldsymbol{\mathcal{A}}^{\star} * (\boldsymbol{\mathcal{B}}^{\star})^T$ and we would bound the $\|\mathcal{E}_2\|, \|\mathcal{E}_3\|$ separately. Before that, we focus on uniform tube-wise bound for $\boldsymbol{\Delta}_{\boldsymbol{\mathcal{X}}}^{t,l}. \ \forall j \in [n]$,

$$\left\| \left( \boldsymbol{\Delta}_{\boldsymbol{\mathcal{X}}}^{t,l} \right)_{lj:} \right\|_F = \left\| \boldsymbol{\mathcal{A}}_{l::}^{t,l} * \left( \boldsymbol{\mathcal{B}}_{j::}^{t,l} \right)^T - \boldsymbol{\mathcal{A}}_{l::}^{\star} * (\boldsymbol{\mathcal{B}}_{j::}^{\star})^T \right\|_F$$

$$= \left\| \left( \boldsymbol{\mathcal{A}}_{l::}^{t,l} * \boldsymbol{\mathcal{T}}^{t,l} - \boldsymbol{\mathcal{A}}_{l::}^{\star} \right) * (\boldsymbol{\mathcal{B}}_{j::}^{\star})^T + \boldsymbol{\mathcal{A}}_{l::}^{\star} * \left( \boldsymbol{\mathcal{B}}_{j::}^{t,l} * \boldsymbol{\mathcal{T}}^{t,l} - \boldsymbol{\mathcal{B}}_{j::}^{\star} \right)^T \right.$$
$$\left. + \left( \boldsymbol{\mathcal{A}}_{l::}^{t,l} * \boldsymbol{\mathcal{T}}^{t,l} - \boldsymbol{\mathcal{A}}_{l::}^{\star} \right) * \left( \boldsymbol{\mathcal{B}}_{j::}^{t,l} * \boldsymbol{\mathcal{T}}^{t,l} - \boldsymbol{\mathcal{B}}_{j::}^{\star} \right)^T \right\|_F$$

$$\stackrel{(i)}{\leq} \left\| \boldsymbol{\mathcal{A}}_{l::}^{t,l} * \boldsymbol{\mathcal{T}}^{t,l} - \boldsymbol{\mathcal{A}}_{l::}^{\star} \right\| \left\| \boldsymbol{\mathcal{B}}_{j::}^{\star} \right\| + \left\| \boldsymbol{\mathcal{A}}_{l::}^{\star} \right\| \left\| \boldsymbol{\mathcal{B}}_{j::}^{t,l} * \boldsymbol{\mathcal{T}}^{t,l} - \boldsymbol{\mathcal{B}}_{j::}^{\star} \right\|$$
$$+ \left\| \boldsymbol{\mathcal{A}}_{l::}^{t,l} * \boldsymbol{\mathcal{T}}^{t,l} - \boldsymbol{\mathcal{A}}_{l::}^{\star} \right\| \left\| \boldsymbol{\mathcal{B}}_{j::}^{t,l} * \boldsymbol{\mathcal{T}}^{t,l} - \boldsymbol{\mathcal{B}}_{j::}^{\star} \right\|$$

$$\stackrel{(ii)}{\leq} 1292 C \rho^t \sqrt{\frac{(\mu r)^5 \kappa^9 n_3 \log^2(n_3 \vee n) \sigma_{\max}}{n^2 m}} \cdot \sqrt{\frac{\mu r \sigma_{\max}}{n}}$$
$$+ \left( 646 C \rho^t \sqrt{\frac{(\mu r)^5 \kappa^9 n_3 \log^2(n_3 \vee n) \sigma_{\max}}{n^2 m}} \right)^2$$

$$\leq 1300 C \rho^t \sqrt{\frac{(\mu r)^5 \kappa^9 n_3 \log^2(n_3 \vee n) \sigma_{\max}}{n^2 m}} \cdot \sqrt{\frac{\mu r \sigma_{\max}}{n}}. \tag{78}$$

(i) is based on (95), and the last inequality is due to the tube-wise sample complexity in (12). (ii) is based on the tensor incoherence condition (10) and the following inequality.

$$\max_{l} \left\| \left( \begin{bmatrix} \boldsymbol{\mathcal{A}}^{t,l} \\ \boldsymbol{\mathcal{B}}^{t,l} \end{bmatrix} * \boldsymbol{\mathcal{T}}^{t,l} - \begin{bmatrix} \boldsymbol{\mathcal{A}}^{\star} \\ \boldsymbol{\mathcal{B}}^{\star} \end{bmatrix} \right)_{l::} \right\| \leq \max_{l} \left\| \left( \begin{bmatrix} \boldsymbol{\mathcal{A}}^{t,l} \\ \boldsymbol{\mathcal{B}}^{t,l} \end{bmatrix} * \boldsymbol{\mathcal{T}}^{t,l} - \begin{bmatrix} \boldsymbol{\mathcal{A}}^{t} \\ \boldsymbol{\mathcal{B}}^{t} \end{bmatrix} * \boldsymbol{\mathcal{R}}^{t} \right)_{l::} \right\|$$
$$+ \max_{l} \left\| \left( \begin{bmatrix} \boldsymbol{\mathcal{A}}^{t} \\ \boldsymbol{\mathcal{B}}^{t} \end{bmatrix} * \boldsymbol{\mathcal{R}}^{t} - \begin{bmatrix} \boldsymbol{\mathcal{A}}^{\star} \\ \boldsymbol{\mathcal{B}}^{\star} \end{bmatrix} \right)_{l::} \right\|$$

$$\overset{(iv)}{\leq} \left\| \begin{bmatrix} \boldsymbol{\mathcal{A}}^{t,l} \\ \boldsymbol{\mathcal{B}}^{t,l} \end{bmatrix} * \boldsymbol{\mathcal{T}}^{t,l} - \begin{bmatrix} \boldsymbol{\mathcal{A}}^{t} \\ \boldsymbol{\mathcal{B}}^{t} \end{bmatrix} * \boldsymbol{\mathcal{R}}^{t} \right\| + \max_{l} \left\| \left( \begin{bmatrix} \boldsymbol{\mathcal{A}}^{t} \\ \boldsymbol{\mathcal{B}}^{t} \end{bmatrix} * \boldsymbol{\mathcal{R}}^{t} - \begin{bmatrix} \boldsymbol{\mathcal{A}}^{\star} \\ \boldsymbol{\mathcal{B}}^{\star} \end{bmatrix} \right)_{l::} \right\|$$

$$\overset{(v)}{\leq} 52C\rho^{t}\sqrt{\frac{(\mu r)^4 \kappa^2 n_3 \log^2(n_3 \vee n)\sigma_{\max}}{n^2 m}}$$

$$+ 594C\rho^{t}\sqrt{\frac{(\mu r)^5 \kappa^9 n_3 \log^2(n_3 \vee n)\sigma_{\max}}{n^2 m}}$$

$$\leq 646C\rho^{t}\sqrt{\frac{(\mu r)^5 \kappa^9 n_3 \log^2(n_3 \vee n)\sigma_{\max}}{n^2 m}}, \tag{79}$$

where (iv) we use inequality in (92) and (v) are due to (23) and (13). Now we bound the $\|\boldsymbol{\mathcal{E}}_2\|$ based on Lemma 2 as

$$\|\boldsymbol{\mathcal{E}}_2\| \leq 2Cmr\sqrt{\frac{n_3}{m}}\log(n \vee n_3) \max_{ij} \left\| \left( \boldsymbol{\Delta}_{\boldsymbol{\mathcal{X}}}^{t,l} \right)_{ij:} \right\|_F \left\| \boldsymbol{\mathcal{B}}^{t,l} \right\|$$

$$\overset{(vi)}{\leq} 2Cmr\sqrt{\frac{n_3}{m}}\log(n \vee n_3) \cdot 1300C\rho^{t}\sqrt{\frac{(\mu r)^5 \kappa^9 n_3 \log^2(n_3 \vee n)\sigma_{\max}}{n^2 m}} \cdot \sqrt{\frac{\mu r \sigma_{\max}}{n}}$$

$$\cdot \left( \left\| \boldsymbol{\mathcal{B}}^{t,l} * \boldsymbol{\mathcal{T}}^{t,l} - \boldsymbol{\mathcal{B}}^{\star} \right\| + \|\boldsymbol{\mathcal{B}}^{\star}\| \right)$$

$$\overset{(vii)}{\leq} 2600C^2 m \sqrt{\frac{(\mu r)^3 \log^2(n \vee n_3) n_3 \sigma_{\max}}{nm}} \cdot \rho^{t}\sqrt{\frac{(\mu r)^5 \kappa^9 n_3 \log^2(n_3 \vee n)\sigma_{\max}}{n^2 m}}$$

$$\cdot \left( 646C\rho^{t}\sqrt{\frac{(\mu r)^5 \kappa^9 n_3 \log^2(n_3 \vee n)\sigma_{\max}}{nm}} + \sqrt{\sigma_{\max}} \right)$$

$$\leq 5200C^2 m \sigma_{\min} \rho^{t}\sqrt{\frac{(\mu r)^3 \kappa^2 \log^2(n \vee n_3) n_3}{nm}} \cdot \rho^{t}\sqrt{\frac{(\mu r)^5 \kappa^9 n_3 \log^2(n_3 \vee n)\sigma_{\max}}{n^2 m}}$$

$$\leq 2Cm\sigma_{\min} \rho^{t}\sqrt{\frac{(\mu r)^3 \kappa^2 n_3 \log^2(n \vee n_3)\sigma_{\max}}{n^2 m}}, \tag{80}$$

where (vi) and (vii) are due to (78), (79), respectively. The last two inequalities are due to tube-wise sample complexity (12).

Now comes to bound the $\|\boldsymbol{\mathcal{E}}_3\|$. To obtain a tight bound, we also bound the spectral norm of each tube, $\left( \boldsymbol{Z}_{lj}^T \boldsymbol{Z}_{lj} - m\boldsymbol{I} \right) \left( \boldsymbol{\Delta}_{\boldsymbol{\mathcal{X}}}^{t,l} \right)_{lj:}$. $\forall j \in [n]$, there is

$$\frac{t^2}{\sum_{s=1}^{m} K_s^2} \geq \frac{t^2}{\sum_{s=1}^{m} \left\| \left( \boldsymbol{\Delta}_{\boldsymbol{\mathcal{X}}}^{t,l} \right)_{lj:} \right\|_F^2 \|\boldsymbol{\mathcal{T}}\|_F^2} \geq \frac{t^2}{m \max_{l,j} \left\| \left( \boldsymbol{\Delta}_{\boldsymbol{\mathcal{X}}}^{t,l} \right)_{lj:} \right\|_F^2 \|\boldsymbol{\mathcal{T}}\|_F^2} = \frac{t^2}{mn_3 \max_{l,j} \left\| \left( \boldsymbol{\Delta}_{\boldsymbol{\mathcal{X}}}^{t,l} \right)_{lj:} \right\|_F^2}$$

$$\frac{t}{\max_s K_s} \geq \frac{t}{\max_{l,j} \left\| \left( \boldsymbol{\Delta}_{\boldsymbol{\mathcal{X}}}^{t,l} \right)_{lj:} \right\|_F \|\boldsymbol{\mathcal{T}}\|_F} = \frac{t}{\sqrt{n_3} \max_{l,j} \left\| \left( \boldsymbol{\Delta}_{\boldsymbol{\mathcal{X}}}^{t,l} \right)_{lj:} \right\|_F}. \tag{81}$$

Based on the sub-exponential Bernstein inequality and $\epsilon$-net argument,

$$\left\| \left( \boldsymbol{Z}_{lj}^T \boldsymbol{Z}_{lj} - m\boldsymbol{I} \right) \left( \boldsymbol{\Delta}_{\boldsymbol{\mathcal{X}}}^{t,l} \right)_{lj:} \right\| \leq \sqrt{mn_3} \max_{l,j} \left\| \left( \boldsymbol{\Delta}_{\boldsymbol{\mathcal{X}}}^{t,l} \right)_{lj:} \right\|_F \sqrt{\log(n \vee n_3)} + \sqrt{n_3} \max_{l,j} \left\| \left( \boldsymbol{\Delta}_{\boldsymbol{\mathcal{X}}}^{t,l} \right)_{lj:} \right\|_F \log(n \vee n_3)$$

$$\tag{82}$$

holds with at least $1 - \exp(-c \log(n \vee n_3))$ probability. Finally using the union bound and inequality (92), we can obtain the bound for $\boldsymbol{\mathcal{E}}_3$ as

$$\|\boldsymbol{\mathcal{E}}_3\| \leq C\sqrt{n} \cdot \left( \sqrt{mn_3} \max_{l,j} \left\| \left( \boldsymbol{\Delta}_{\boldsymbol{\mathcal{X}}}^{t,l} \right)_{lj:} \right\|_F \sqrt{\log(n \vee n_3)} + \sqrt{n_3} \max_{l,j} \left\| \left( \boldsymbol{\Delta}_{\boldsymbol{\mathcal{X}}}^{t,l} \right)_{lj:} \right\|_F \log(n \vee n_3) \right) \cdot \left\| \boldsymbol{\mathcal{A}}_{l::}^{t,l} \right\|$$

$$\leq 2Cm\sqrt{\frac{nn_3}{m}}\log(n \vee n_3)\max_{l,j}\left\|\left(\boldsymbol{\Delta}_{\boldsymbol{\mathcal{X}}}^{t,l}\right)_{lj:}\right\|_F \cdot \left\|\boldsymbol{\mathcal{A}}_{l::}^{t,l}\right\|$$

$$\overset{(i)}{\leq} 2Cm\sqrt{\frac{nn_3\log^2(n \vee n_3)}{m}}\cdot 1300C\rho^t\sqrt{\frac{(\mu r)^5\kappa^9 n_3\log^2(n_3 \vee n)\sigma_{\max}}{n^2 m}}\cdot\sqrt{\frac{\mu r\sigma_{\max}}{n}}$$

$$\cdot\left(\left\|\boldsymbol{\mathcal{A}}_{l::}^{t,l}*\boldsymbol{\mathcal{T}}^{t,l}-\boldsymbol{\mathcal{A}}_{l::}^{\star}\right\|+\|\boldsymbol{\mathcal{A}}_{l::}^{\star}\|\right)$$

$$\overset{(ii)}{\leq} 2600C^2 m\rho^t\sqrt{\frac{\mu r n_3\log^2(n \vee n_3)\sigma_{\max}}{m}}\cdot\rho^t\sqrt{\frac{(\mu r)^5\kappa^9 n_3\log^2(n_3 \vee n)\sigma_{\max}}{n^2 m}}$$

$$\cdot\left(646C\rho^t\sqrt{\frac{(\mu r)^5\kappa^9 n_3\log^2(n_3 \vee n)\sigma_{\max}}{n^2 m}}+\sqrt{\frac{\mu r\sigma_{\max}}{n}}\right)$$

$$\leq 5200C^2 m\sigma_{\min}\sqrt{\frac{(\mu r)^2\kappa^2 n_3\log^2(n \vee n_3)}{nm}}\cdot\rho^t\sqrt{\frac{(\mu r)^5\kappa^9 n_3\log^2(n_3 \vee n)\sigma_{\max}}{n^2 m}}$$

$$\leq 2Cm\sigma_{\min}\rho^t\sqrt{\frac{(\mu r)^2\kappa^2 n_3\log^2(n \vee n_3)\sigma_{\max}}{n^2 m}}, \tag{83}$$

where (i) is due to (78) and triangle inequality. (ii) is because (79) and tensor incoherence condition (10). The last two inequalities are due to tube-wise sample complexity in (12).

Finally, combining the bounds in (75), (80) and (83) with (70) would have

$$\left\|\begin{bmatrix}\boldsymbol{\mathcal{A}}^{t+1}\\\boldsymbol{\mathcal{B}}^{t+1}\end{bmatrix}*\boldsymbol{\mathcal{R}}^{t+1}-\begin{bmatrix}\boldsymbol{\mathcal{A}}^{t+1,l}\\\boldsymbol{\mathcal{B}}^{t+1,l}\end{bmatrix}*\boldsymbol{\mathcal{T}}^{t+1,l}\right\|\leq\left(1-\frac{m\eta\sigma_{\min}}{5}\right)\left\|\begin{bmatrix}\boldsymbol{\mathcal{A}}^{t,l}*\boldsymbol{\mathcal{T}}^{t,l}-\boldsymbol{\mathcal{A}}^t*\boldsymbol{\mathcal{R}}^t\\\boldsymbol{\mathcal{B}}^{t,l}*\boldsymbol{\mathcal{T}}^{t,l}-\boldsymbol{\mathcal{B}}^t*\boldsymbol{\mathcal{R}}^t\end{bmatrix}\right\|$$

$$+2Cm\sigma_{\min}\rho^t\sqrt{\frac{(\mu r)^3\kappa^2 n_3\log^2(n \vee n_3)\sigma_{\max}}{n^2 m}}$$

$$+2Cm\sigma_{\min}\rho^t\sqrt{\frac{(\mu r)^2\kappa^2 n_3\log^2(n \vee n_3)\sigma_{\max}}{n^2 m}}$$

$$\leq\left(1-\frac{m\eta\sigma_{\min}}{5}\right)\cdot 52C\rho^t\sqrt{\frac{(\mu r)^4\kappa^2 n_3\log^2(n \vee n_3)\sigma_{\max}}{n^2 m}}$$

$$+0.08\times 52C\rho^t\sqrt{\frac{(\mu r)^3\kappa^2 n_3\log^2(n \vee n_3)\sigma_{\max}}{n^2 m}}$$

$$\leq 52C\rho^{t+1}\sqrt{\frac{(\mu r)^4\kappa^2 n_3\log^2(n \vee n_3)\sigma_{\max}}{n^2 m}}, \tag{84}$$

where the last inequality is due to the $\rho=(1-0.01m\eta\sigma_{\min})$.

## A.5 Proof of (14)

This proof is combining (23) and (22) by triangle inequality. Thus. there is decomposition as

$$\left\|\left(\begin{bmatrix}\boldsymbol{\mathcal{A}}^t\\\boldsymbol{\mathcal{B}}^t\end{bmatrix}*\boldsymbol{\mathcal{R}}^t-\begin{bmatrix}\boldsymbol{\mathcal{A}}^\star\\\boldsymbol{\mathcal{B}}^\star\end{bmatrix}\right)_{l::}\right\|\leq\left\|\left(\begin{bmatrix}\boldsymbol{\mathcal{A}}^t\\\boldsymbol{\mathcal{B}}^t\end{bmatrix}*\boldsymbol{\mathcal{R}}^t-\begin{bmatrix}\boldsymbol{\mathcal{A}}^{t,l}\\\boldsymbol{\mathcal{B}}^{t,l}\end{bmatrix}*\boldsymbol{\mathcal{R}}^{t,l}\right)_{l::}\right\|+\left\|\left(\begin{bmatrix}\boldsymbol{\mathcal{A}}^{t,l}\\\boldsymbol{\mathcal{B}}^{t,l}\end{bmatrix}*\boldsymbol{\mathcal{R}}^{t,l}-\begin{bmatrix}\boldsymbol{\mathcal{A}}^\star\\\boldsymbol{\mathcal{B}}^\star\end{bmatrix}\right)_{l::}\right\|$$

$$\leq\left\|\begin{bmatrix}\boldsymbol{\mathcal{A}}^t\\\boldsymbol{\mathcal{B}}^t\end{bmatrix}*\boldsymbol{\mathcal{R}}^t-\begin{bmatrix}\boldsymbol{\mathcal{A}}^{t,l}\\\boldsymbol{\mathcal{B}}^{t,l}\end{bmatrix}*\boldsymbol{\mathcal{R}}^{t,l}\right\|+\left\|\left(\begin{bmatrix}\boldsymbol{\mathcal{A}}^{t,l}\\\boldsymbol{\mathcal{B}}^{t,l}\end{bmatrix}*\boldsymbol{\mathcal{R}}^{t,l}-\begin{bmatrix}\boldsymbol{\mathcal{A}}^\star\\\boldsymbol{\mathcal{B}}^\star\end{bmatrix}\right)_{l::}\right\|, \tag{85}$$

where the last inequality is due to tensor norm inequality (92). For bound the $\left\| \begin{bmatrix} \mathcal{A}^t \\ \mathcal{B}^t \end{bmatrix} * \mathcal{R}^t - \begin{bmatrix} \mathcal{A}^{t,l} \\ \mathcal{B}^{t,l} \end{bmatrix} * \mathcal{R}^{t,l} \right\|$, we apply the Lemma 6 again as

$$\boldsymbol{\mathcal{X}}_0 = \begin{bmatrix} \mathcal{A}^\star \\ \mathcal{B}^\star \end{bmatrix}, \quad \boldsymbol{\mathcal{X}}_1 = \begin{bmatrix} \mathcal{A}^t \\ \mathcal{B}^t \end{bmatrix} * \mathcal{R}^t, \quad \boldsymbol{\mathcal{X}}_2 = \begin{bmatrix} \mathcal{A}^{t,l} \\ \mathcal{B}^{t,l} \end{bmatrix} * \mathcal{T}^{t,l}. \tag{86}$$

Based on tube-wise sample complexity, there is

$$\|\boldsymbol{\mathcal{X}}_1 - \boldsymbol{\mathcal{X}}_0\| \|\boldsymbol{\mathcal{X}}_0\| \overset{(i)}{\leq} 98 C \rho^t \sqrt{\frac{(\mu r)^4 \kappa^5 n_3 \log^2(n \vee n_3) \sigma_{\max}}{nm}} \cdot \sqrt{\sigma_{\max}}$$

$$= 98 C \sigma_{\min} \rho^t \sqrt{\frac{(\mu r)^4 \kappa^7 n_3 \log^2(n \vee n_3)}{nm}}$$

$$\leq \frac{\sigma_{\min}}{2}, \tag{87}$$

where (i) is due to (13) and the last is due to tube-wise sample complexity. In addition,

$$\|\boldsymbol{\mathcal{X}}_1 - \boldsymbol{\mathcal{X}}_2\| \|\boldsymbol{\mathcal{X}}_0\| \leq 52 C \rho^t \sqrt{\frac{(\mu r)^4 \kappa^2 n_3 \log^2(n \vee n_3) \sigma_{\max}}{n^2 m}} \cdot \sqrt{\sigma_{\max}}$$

$$= 52 C \sigma_{\min} \rho^t \sqrt{\frac{(\mu r)^4 \kappa^4 n_3 \log^2(n \vee n_3)}{n^2 m}}$$

$$\leq \frac{\sigma_{\min}}{4}, \tag{88}$$

where (i) the first inequality is due to (23) and the last is due to tube-wise sample complexity. Thus,

$$\left\| \left( \begin{bmatrix} \mathcal{A}^t \\ \mathcal{B}^t \end{bmatrix} * \mathcal{R}^t - \begin{bmatrix} \mathcal{A}^\star \\ \mathcal{B}^\star \end{bmatrix} \right)_{l::} \right\| \leq 9\kappa \left\| \begin{bmatrix} \mathcal{A}^t \\ \mathcal{B}^t \end{bmatrix} * \mathcal{R}^t - \begin{bmatrix} \mathcal{A}^{t,l} \\ \mathcal{B}^{t,l} \end{bmatrix} * \mathcal{T}^{t,l} \right\| + \left\| \left( \begin{bmatrix} \mathcal{A}^{t,l} \\ \mathcal{B}^{t,l} \end{bmatrix} * \mathcal{R}^{t,l} - \begin{bmatrix} \mathcal{A}^\star \\ \mathcal{B}^\star \end{bmatrix} \right)_{l::} \right\|$$

$$\leq 468 C \rho^t \sqrt{\frac{(\mu r)^4 \kappa^4 n_3 \log^2(n \vee n_3) \sigma_{\max}}{n^2 m}}$$

$$+ 2496 C \rho^t \sqrt{\frac{(\mu r)^5 \kappa^9 n_3 \log^2(n \vee n_3) \sigma_{\max}}{n^2 m}}$$

$$\leq 2964 C \rho^t \sqrt{\frac{(\mu r)^5 \kappa^9 n_3 \log^2(n \vee n_3) \sigma_{\max}}{n^2 m}}. \tag{89}$$

## A.6  Technical Lemmas

This section presents the technical lemmas used in proofs of Section A. The first lemma is some important tensor norm inequalities under the t-SVD framework, which will be repeatedly used throughout the proof.

**Lemma 1.** *For $\forall \boldsymbol{\mathcal{X}} \in \mathbb{R}^{n_1 \times n_2 \times n_3}, \boldsymbol{\mathcal{Y}} \in \mathbb{R}^{n_2 \times n_4 \times n_3}$, there are tensor norm inequalities as*

$$\|\boldsymbol{\mathcal{X}}\| \leq \sqrt{n_3} \|\boldsymbol{\mathcal{X}}\|_F; \tag{90}$$

$$\|\boldsymbol{\mathcal{X}}(i,:,:)\|_F \leq \|\boldsymbol{\mathcal{X}}(i,:,:)\|, \quad \forall i \in [n_1]; \tag{91}$$

$$\max_{i \in [n_1]} \|\boldsymbol{\mathcal{X}}_{i::}\| \leq \|\boldsymbol{\mathcal{X}}\| \leq \sqrt{\sum_{i=1}^{n_1} \|\boldsymbol{\mathcal{X}}(i,:,:)\|^2}; \tag{92}$$

$$\|\boldsymbol{\mathcal{X}} * \boldsymbol{\mathcal{Y}}\|_F \leq \|\boldsymbol{\mathcal{X}}\| \|\boldsymbol{\mathcal{Y}}\|_F \leq \sqrt{n_3} \|\boldsymbol{\mathcal{X}}\|_F \|\boldsymbol{\mathcal{Y}}\|_F; \tag{93}$$

$$\|\boldsymbol{\mathcal{X}} * \boldsymbol{\mathcal{Y}}\| \leq \|\boldsymbol{\mathcal{X}}\| \|\boldsymbol{\mathcal{Y}}\|; \tag{94}$$

$$\|\boldsymbol{\mathcal{X}}(i,:,:) * \boldsymbol{\mathcal{Y}}(:,j,:)\|_F \leq \|\boldsymbol{\mathcal{X}}(i,:,:)\| \|\boldsymbol{\mathcal{Y}}(:,j,:)\|, \quad \forall i \in [n_1], \forall j \in [n_4]. \tag{95}$$

**Remark 3.** *The relationship between the spectral norm and the Frobenius norm in the t-SVD tensor algebra differs from the matrix case. For a matrix $X$, one has*

$$\|X\| \leq \|X\|_F,$$

*whereas in the tensor setting an extra factor of $\sqrt{n_3}$ appears in (90). Similarly, the matrix inequality*

$$\|XY\|_F \leq \|X\|_F \|Y\|_F$$

*also acquires a factor of $\sqrt{n_3}$ in (93). This $\sqrt{n_3}$ factor is unavoidable in the general third-order tensor context. Interestingly, for a horizontal slice tensor $\boldsymbol{\mathcal{X}}(i,:,:) \in \mathbb{R}^{1 \times n_2 \times n_3}$, the Frobenius norm is bounded by its spectral norm, reversing the usual matrix inequality.*

*Proof.* For the (90), we have

$$\|\boldsymbol{\mathcal{X}}\|^2 \overset{(a)}{=} \|\text{bcirc}\,(\boldsymbol{\mathcal{X}})\|^2 \overset{(b)}{\leq} \|\text{bcirc}\,(\boldsymbol{\mathcal{X}})\|_F^2 \overset{(c)}{=} \sqrt{n_3}\,\|\boldsymbol{\mathcal{X}}\|_F^2\,, \tag{96}$$

where we use the definition of tensor spectral norm in (a), matrix norm inequality in (b). (c) is due to the definition of the bcirc operator in (3) and equality (102).

For (91),

$$\|\boldsymbol{\mathcal{X}}(i,:,:)\|_F = \sqrt{\frac{\sum_{k=1}^{n_3} \left\|\overline{\boldsymbol{X}}_{i:}^{(k)}\right\|^2}{n_3}} \leq \max_{k \in [n_3]} \left\|\overline{\boldsymbol{X}}_{i:}^{(k)}\right\| = \|\boldsymbol{\mathcal{X}}(i,:,:)\| \tag{97}$$

For (92),

$$\|\boldsymbol{\mathcal{X}}\| = \left\|\text{bdiag}\,\left(\overline{\boldsymbol{\mathcal{X}}}\right)\right\|$$

$$= \max_{k \in [n_3]} \left\|\overline{\boldsymbol{X}}^{(k)}\right\|$$

$$\overset{(d)}{\leq} \max_{k \in [n_3]} \sqrt{\sum_{i=1}^{n_1} \left\|\overline{\boldsymbol{X}}_{i,:}^{(k)}\right\|_F^2}$$

$$\leq \sqrt{\sum_{i=1}^{n_1} \max_{k \in [n_3]} \left\|\overline{\boldsymbol{X}}_{i,:}^{(k)}\right\|_F^2}$$

$$\overset{(e)}{=} \sqrt{\sum_{i=1}^{n_1} \max_{k \in [n_3]} \left\|\overline{\boldsymbol{X}}_{i,:}^{(k)}\right\|^2}$$

$$\overset{(f)}{=} \sqrt{\sum_{i=1}^{n_1} \|\boldsymbol{\mathcal{X}}(i,:,:)\|^2}, \tag{98}$$

where (d) uses matrix norm inequality that $\|\boldsymbol{X}\| \leq \|\boldsymbol{X}\|_F$. (e) is based on the fact that for a matrix with only one row, its spectral norm is equal to its Frobenius norm. (f) is due to the definition of the tensor spectral norm.

For (93),

$$\|\boldsymbol{\mathcal{X}} * \boldsymbol{\mathcal{Y}}\|_F = \frac{1}{\sqrt{n_3}} \left\|\overline{\boldsymbol{XY}}\right\|_F$$

$$= \frac{\sqrt{\sum_{k=1}^{n_3} \left\|\overline{\boldsymbol{X}}^{(k)}\overline{\boldsymbol{Y}}^{(k)}\right\|_F^2}}{\sqrt{n_3}}$$

$$\overset{(g)}{\leq} \frac{\sqrt{\sum_{k=1}^{n_3} \left\|\overline{\boldsymbol{Y}}^{(k)}\right\|_F^2 \left\|\overline{\boldsymbol{X}}^{(k)}\right\|^2}}{\sqrt{n_3}}$$

$$\leq \max_{k\in[n_3]} \left\|\overline{\boldsymbol{X}}^{(k)}\right\| \frac{\sqrt{\sum_{k=1}^{n_3}\left\|\overline{\boldsymbol{Y}}^{(k)}\right\|_F^2}}{\sqrt{n_3}}$$

$$= \|\boldsymbol{\mathcal{X}}\|\,\|\boldsymbol{\mathcal{Y}}\|_F. \tag{99}$$

(g) is based on matrix norm inequality that $\|\boldsymbol{X}\boldsymbol{Y}\|_F \leq \|\boldsymbol{Y}\|_F \|\boldsymbol{X}\|$.

For (94), $\|\boldsymbol{\mathcal{X}}*\boldsymbol{\mathcal{Y}}\| = \max_{k\in[n_3]}\left\|\overline{\boldsymbol{X}}^{(k)}\overline{\boldsymbol{Y}}^{(k)}\right\| \leq \max_{k\in[n_3]}\left\|\overline{\boldsymbol{X}}^{(k)}\right\|\max_{k\in[n_3]}\left\|\overline{\boldsymbol{Y}}^{(k)}\right\| = \|\boldsymbol{\mathcal{X}}\|\,\|\boldsymbol{\mathcal{Y}}\|$.

For (95), it is directly result from (92) and (93) as $\|\boldsymbol{\mathcal{X}}(i,:,:)*\boldsymbol{\mathcal{Y}}(:,j,:)\|_F \leq \|\boldsymbol{\mathcal{X}}(i,:,:)\|_F\|\boldsymbol{\mathcal{Y}}(:,j,:)\| \leq \|\boldsymbol{\mathcal{X}}(i,:,:)\|\,\|\boldsymbol{\mathcal{Y}}(:,j,:)\|$. $\qquad\square$

The following lemma is the key lemma in our proof, which measures the concentration of the tube-wise sensing map $\boldsymbol{\mathcal{Z}}:\mathbb{R}^{n\times n\times n_3}\to\mathbb{R}^{n\times n\times m}$ of Definition 1 in the tensor spectral norm metric.

**Lemma 2.** *For the defined tube-wise sensing operator $\boldsymbol{\mathcal{Z}}$, all entries in $\{\boldsymbol{Z}_{ij}\}$ obey i.i.d standard Gaussian distribution, then $\forall$ tensor $\boldsymbol{\Delta}\in\mathbb{R}^{n\times n\times n_3}, \boldsymbol{\mathcal{F}}\in\mathbb{R}^{n\times r\times n_3}$, the*

$$\|(m\boldsymbol{\mathcal{I}}-\boldsymbol{\mathcal{Z}}^c\boldsymbol{\mathcal{Z}})(\boldsymbol{\Delta})*\boldsymbol{\mathcal{F}}\| \leq 2Cmr\sqrt{\frac{nn_3}{m}}\log(n\vee n_3)\max_{ij}\|\boldsymbol{\Delta}_{ij:}\|_F\|\boldsymbol{\mathcal{F}}\| \tag{100}$$

*holds with high probability at least $1-\frac{1}{(n\vee n_3)^{10}}$.*

*Proof.* Due to the independence of each horizontal slice from $\boldsymbol{\mathcal{Z}}$, the core idea is to bound each horizontal slice separately and utilize the union bound to control concentration. Denote $\boldsymbol{\mathcal{C}} := (m\boldsymbol{\mathcal{I}}-\boldsymbol{\mathcal{Z}}^c\boldsymbol{\mathcal{Z}})(\boldsymbol{\Delta})*\boldsymbol{\mathcal{F}}$, then $\boldsymbol{\mathcal{C}}_{i::} = \sum_{j=1}^n\left(\boldsymbol{Z}_{ij}^T\boldsymbol{Z}_{ij}\boldsymbol{\Delta}_{ij:}-m\boldsymbol{I}\right)*\boldsymbol{\mathcal{F}}(j,:,:)\in\mathbb{R}^{1\times r\times n_3}$. Based on the variational formula of the matrix spectral norm, there is

$$
\begin{aligned}
\|\boldsymbol{\mathcal{C}}_{i::}\| &\overset{(i)}{=} \max_{\boldsymbol{w}\in\Theta^1,\boldsymbol{z}\in\Theta^r}\langle\overline{\boldsymbol{C}_{i::}},\boldsymbol{w}\boldsymbol{z}^T\rangle\\
&\overset{(ii)}{=} \max_{\boldsymbol{w}\in\Theta^1,\boldsymbol{z}\in\Theta^r}\langle(\boldsymbol{F}_{n_3}\otimes\boldsymbol{I}_1)\,\mathrm{bcirc}(\boldsymbol{\mathcal{C}}_{i::})\left(\boldsymbol{F}_{n_3}^{-1}\otimes\boldsymbol{I}_r\right),\boldsymbol{w}\boldsymbol{z}^T\rangle\\
&= \max_{\boldsymbol{w}\in\Theta^1,\boldsymbol{z}\in\Theta^r}\left\langle\sum_{j=1}^n\left(\boldsymbol{Z}_{ij}^T\boldsymbol{Z}_{ij}\boldsymbol{\Delta}_{ij:}-m\boldsymbol{\Delta}_{ij:}\right)*\boldsymbol{\mathcal{F}}(j,:,:),\underbrace{\mathrm{bcirc}^*\left(\left(\boldsymbol{F}_{n_3}^{-1}\otimes\boldsymbol{I}_1\right)\boldsymbol{w}\boldsymbol{z}^T\left(\boldsymbol{F}_{n_3}\otimes\boldsymbol{I}_r\right)\right)}_{\boldsymbol{\mathcal{T}}\in\mathbb{R}^{1\times r\times n_3}}\right\rangle\\
&= \max_{\boldsymbol{w}\in\Theta^1,\boldsymbol{z}\in\Theta^r}\sum_{j=1}^n\left\langle\boldsymbol{Z}_{ij}\boldsymbol{\Delta}_{ij:},\boldsymbol{Z}_{ij}\left(\boldsymbol{\mathcal{T}}*(\boldsymbol{\mathcal{F}}(j,:,:))^T\right)\right\rangle - m\left\langle\boldsymbol{\Delta}_{ij:},\boldsymbol{\mathcal{T}}*(\boldsymbol{\mathcal{F}}(j,:,:))^T\right\rangle\\
&= \max_{\boldsymbol{w}\in\Theta^1,\boldsymbol{z}\in\Theta^r}\sum_{j=1}^n\sum_{s=1}^m\left(\underbrace{\boldsymbol{z}_{ijs}^T\boldsymbol{\Delta}_{ij:}\boldsymbol{z}_{ijs}^T\left(\boldsymbol{\mathcal{T}}*(\boldsymbol{\mathcal{F}}(j,:,:))^T\right)-\left\langle\boldsymbol{\Delta}_{ij:},\boldsymbol{\mathcal{T}}*(\boldsymbol{\mathcal{F}}(j,:,:))^T\right\rangle}_{K_{js}}\right).
\end{aligned}
\tag{101}
$$

(i) is due to the definition of the tensor spectral norm. (ii) is because the block circulant matrix in (3) can be block diagonalized, i.e.,

$$(\boldsymbol{F}_{n_3}\otimes\boldsymbol{I}_{n_1})\mathrm{bcirc}(\boldsymbol{\mathcal{X}})(\boldsymbol{F}_{n_3}^{-1}\otimes\boldsymbol{I}_{n_2}) = \overline{\boldsymbol{X}}, \tag{102}$$

where $F_n$ is the Fourier matrix and $\overline{\boldsymbol{X}}$ is the block diagonal of $\overline{\boldsymbol{\mathcal{X}}}$ as

$$\overline{\boldsymbol{X}} = \begin{bmatrix}\overline{\boldsymbol{X}}^{(1)} & & & \\ & \overline{\boldsymbol{X}}^{(2)} & & \\ & & \ddots & \\ & & & \overline{\boldsymbol{X}}^{(n_3)}\end{bmatrix} \tag{103}$$

The set $\Theta^k$ is defined as $\Theta^k := \mathcal{B}^k \bigcap \mathcal{S}^k$. The set $\mathcal{B}^k$ denote the block sparse vectors, specially, $\mathcal{B}^k := \left\{ \boldsymbol{x} \in \mathbb{R}^{kn_3} | \boldsymbol{x} = \left[ \boldsymbol{x}_1^T, \cdots, \boldsymbol{x}_i^T, \cdots, \boldsymbol{x}_{n_3}^T \right] \right\}$, where $\boldsymbol{x}_i \in \mathbb{R}^k$ and there exists a $j$ such that $\boldsymbol{x}_j \neq \boldsymbol{0}$ and $\boldsymbol{x}_i = \boldsymbol{0}$ for all $i \neq j$. The $\mathcal{S}^k$ is defined as $\mathcal{S}^k := \left\{ \boldsymbol{x} \in \mathbb{R}^{kn_3} | \, \|\boldsymbol{x}\| = 1 \right\}$.

Due to independence of $\{\boldsymbol{z}_{ijs}\}_{j=1,s=1}^{j=n,s=m}, \forall i \in [n]$, the $K_{js}$ are independent sub-exponential scalars with zero means and have bounded sub-exponential norm as $\|K_{js}\|_{\psi_1} \leq \|\boldsymbol{\Delta}_{ij:}\|_F \left\| \boldsymbol{\mathcal{T}} * (\boldsymbol{\mathcal{F}}(j,:,:))^T \right\|_F$. $\forall t > 0$, there are

$$
\frac{t^2}{\sum_{j=1}^n \sum_{s=1}^m \|K_{js}\|_{\psi_1}^2} \geq \frac{t^2}{\sum_{j=1}^n \sum_{s=1}^m \|\boldsymbol{\Delta}_{ij:}\|_F^2 \left\| \boldsymbol{\mathcal{T}} * (\boldsymbol{\mathcal{F}}(j,:,:))^T \right\|_F^2}
$$

$$
\geq \frac{t^2}{m \max_{ij} \|\boldsymbol{\Delta}_{ij:}\|_F^2 \sum_{j=1}^n \left\| \boldsymbol{\mathcal{T}} * (\boldsymbol{\mathcal{F}}(j,:,:))^T \right\|_F^2}
$$

$$
= \frac{t^2}{m \max_{ij} \|\boldsymbol{\Delta}_{ij:}\|_F^2 \left\| \boldsymbol{\mathcal{T}} * \boldsymbol{\mathcal{F}}^T \right\|_F^2}
$$

$$
\overset{(i)}{\geq} \frac{t^2}{m \max_{ij} \|\boldsymbol{\Delta}_{ij:}\|_F^2 \|\boldsymbol{\mathcal{T}}\|_F^2 \|\boldsymbol{\mathcal{F}}\|^2}
$$

$$
\overset{(ii)}{=} \frac{t^2}{mn_3 \max_{ij} \|\boldsymbol{\Delta}_{ij:}\|_F^2 \|\boldsymbol{\mathcal{F}}\|^2}, \tag{104}
$$

$$
\frac{t}{\max_{js} \|K_{js}\|_{\psi_1}} \geq \frac{t}{\max_{ij} \|\boldsymbol{\Delta}_{ij:}\|_F \max_j \left\| \boldsymbol{\mathcal{T}} * (\boldsymbol{\mathcal{F}}(j,:,:))^T \right\|_F}
$$

$$
\overset{(iii)}{\geq} \frac{t}{\max_{ij} \|\boldsymbol{\Delta}_{ij:}\|_F \|\boldsymbol{\mathcal{T}}\|_F \max_j \|\boldsymbol{\mathcal{F}}_{j::}\|}
$$

$$
\overset{(iv)}{=} \frac{t}{\sqrt{n_3} \max_{ij} \|\boldsymbol{\Delta}_{ij:}\|_F \max_j \|\boldsymbol{\mathcal{F}}_{j::}\|}. \tag{105}
$$

(i) and (iii) are due to norm inequality in (93). (ii) and (iv) are because $\|\boldsymbol{\mathcal{T}}\|_F = \sqrt{n_3}$. Thus, based on the sub-exponential Bernstein's inequality and $\epsilon$-net argument, $\forall i \in [n]$ there is

$$
\mathbb{P}\left( \|\boldsymbol{\mathcal{C}}_{i::}\| \geq t \right) \leq \exp\left( r \log n_3 - c \min\left\{ \frac{t^2}{mn_3 \max_{ij} \|\boldsymbol{\Delta}_{ij:}\|_F^2 \|\boldsymbol{\mathcal{F}}\|^2}, \frac{t}{\sqrt{n_3} \max_{ij} \|\boldsymbol{\Delta}_{ij:}\|_F \max_j \|\boldsymbol{\mathcal{F}}_{j::}\|} \right\} \right).
\tag{106}
$$

The $\exp(r \log n_3)$ is due to the covering number of for union set $\Theta^1 \times \Theta^2$ is $\mathcal{O}\left( \exp\left( (r+1) \log n_3 \right) \right)$. Based on the independence among $\boldsymbol{\mathcal{C}}_{i::}, \forall i \in [n]$ due to tube-wise sensing operator $\boldsymbol{\mathcal{Z}}$, there is union bound

$$
\mathbb{P}\left( \sqrt{\sum_{i=1}^n \|\boldsymbol{\mathcal{C}}_{i::}\|^2} \geq \sqrt{n}t \right) \leq \sum_{i=1}^n \mathbb{P}\left( \|\boldsymbol{\mathcal{C}}_{i::}\| \geq t \right)
$$

$$
\leq n \exp(r \log n_3) \cdot \exp\left( -c \min\left\{ \frac{t^2}{mn_3 \max_{ij} \|\boldsymbol{\Delta}_{ij:}\|_F^2 \|\boldsymbol{\mathcal{F}}\|^2}, \frac{t}{\sqrt{n_3} \max_{ij} \|\boldsymbol{\Delta}_{ij:}\|_F \max_j \|\boldsymbol{\mathcal{F}}_{j::}\|} \right\} \right).
\tag{107}
$$

Thus,

$$
\|\boldsymbol{\mathcal{C}}\| \leq \sqrt{\sum_{i=1}^n \|\boldsymbol{\mathcal{C}}_{i::}\|^2}
$$

$$
\leq C\sqrt{n} \cdot \left( \sqrt{mn_3 r \log(n \vee n_3)} \max_{ij} \|\boldsymbol{\Delta}_{ij:}\|_F \|\boldsymbol{\mathcal{F}}\| + \sqrt{n_3} r \log(n \vee n_3) \max_{ij} \|\boldsymbol{\Delta}_{ij:}\|_F \max_j \|\boldsymbol{\mathcal{F}}_{j::}\| \right)
$$

$$\leq 2Cmr\sqrt{\frac{nn_3}{m}}\log(n \vee n_3)\max_{ij}\|\boldsymbol{\Delta}_{ij:}\|_F\|\boldsymbol{\mathcal{F}}\| \tag{108}$$

holds with probability at least $1 - \frac{1}{(n\vee n_3)^{10}}$. The first and last inequalities are due to the tensor norm inequality (92). $\qquad\square$

Besides the above concentration inequality in the tensor spectral norm metric, we also need the following concentration inequality in the inner product metric, which would be used in proving the restricted strong convexity and smooth property of the non-convex loss landscape in (6).

**Lemma 3.** *For the tube-wise sensing map* $\boldsymbol{\mathcal{Z}} : \mathbb{R}^{n \times n \times n_3} \to \mathbb{R}^{n \times n \times m}$ *in Definition 1,* $\forall \boldsymbol{\mathcal{A}}, \boldsymbol{\mathcal{B}}, \boldsymbol{\mathcal{C}}, \boldsymbol{\mathcal{D}} \in \mathbb{R}^{n \times r \times n_3}$ *and the defined concentration distance*

$$D\left(\boldsymbol{\mathcal{A}} * \boldsymbol{\mathcal{C}}^T, \boldsymbol{\mathcal{B}} * \boldsymbol{\mathcal{D}}^T\right) := \left\langle \boldsymbol{\mathcal{Z}}\left(\boldsymbol{\mathcal{A}} * \boldsymbol{\mathcal{C}}^T\right), \boldsymbol{\mathcal{Z}}\left(\boldsymbol{\mathcal{B}} * \boldsymbol{\mathcal{D}}^T\right)\right\rangle - m\left\langle \boldsymbol{\mathcal{A}} * \boldsymbol{\mathcal{C}}^T, \boldsymbol{\mathcal{B}} * \boldsymbol{\mathcal{D}}^T\right\rangle, \tag{109}$$

*there is*

$$\left|D\left(\boldsymbol{\mathcal{A}} * \boldsymbol{\mathcal{C}}^T, \boldsymbol{\mathcal{B}} * \boldsymbol{\mathcal{D}}^T\right)\right| \leq \sqrt{m}\log(n \vee n_3)\sqrt{\sum_{i=1}^n\sum_{j=1}^n \left\|\boldsymbol{\mathcal{A}}_{i::} * (\boldsymbol{\mathcal{C}}_{j::})^T\right\|_F^2 \left\|\boldsymbol{\mathcal{B}}_{i::} * (\boldsymbol{\mathcal{D}}_{j::})^T\right\|_F^2}$$

$$\leq \sqrt{mn_3}\log(n \vee n_3)\left(\left(\|\boldsymbol{\mathcal{A}}\|_F \max_i \|\boldsymbol{\mathcal{C}}_{i::}\|\right) \wedge \left(\|\boldsymbol{\mathcal{C}}\|_F \max_i \|\boldsymbol{\mathcal{C}}_{i::}\|\right)\right)\|\boldsymbol{\mathcal{B}}\|_F\|\boldsymbol{\mathcal{D}}\|_F \tag{110}$$

*with probability at least* $1 - \frac{1}{(n\vee n_3)^{10}}$.

*Proof.*

$$D\left(\boldsymbol{\mathcal{A}} * \boldsymbol{\mathcal{C}}^T, \boldsymbol{\mathcal{B}} * \boldsymbol{\mathcal{D}}^T\right) = \sum_{i=1}^n\sum_{j=1}^n \left\langle \boldsymbol{Z}_{ij}\left(\boldsymbol{\mathcal{A}}_{i::} * (\boldsymbol{\mathcal{C}}_{j::})^T\right), \boldsymbol{Z}_{ij}\left(\boldsymbol{\mathcal{B}}_{i::} * (\boldsymbol{\mathcal{D}}_{j::})^T\right)\right\rangle$$

$$- m\left\langle \boldsymbol{\mathcal{A}}_{i::} * (\boldsymbol{\mathcal{C}}_{j::})^T, \boldsymbol{\mathcal{B}}_{i::} * (\boldsymbol{\mathcal{D}}_{j::})^T\right\rangle$$

$$= \sum_{i=1}^n\sum_{j=1}^n\sum_{s=1}^m \left\{\underbrace{\boldsymbol{z}_{ijs}^T\left(\boldsymbol{\mathcal{A}}_{i::} * (\boldsymbol{\mathcal{C}}_{j::})^T\right)\boldsymbol{z}_{ijs}^T\left(\boldsymbol{\mathcal{B}}_{i::} * (\boldsymbol{\mathcal{D}}_{j::})^T\right) - \left\langle \boldsymbol{\mathcal{A}}_{i::} * (\boldsymbol{\mathcal{C}}_{j::})^T, \boldsymbol{\mathcal{B}}_{i::} * (\boldsymbol{\mathcal{D}}_{j::})^T\right\rangle}_{g_{ijs}}\right\}. \tag{111}$$

It is apparent that $g_{ijs}$ are independent sub-exponential variables with zero means and bounded sub-exponential norm $\|g_{ijs}\|_{\psi_1} \leq \left\|\boldsymbol{\mathcal{A}}_{i::} * (\boldsymbol{\mathcal{C}}_{j::})^T\right\|_F \left\|\boldsymbol{\mathcal{B}}_{i::} * (\boldsymbol{\mathcal{D}}_{j::})^T\right\|_F$. Thus, there is

$$\frac{t^2}{\sum_{i=1}^n\sum_{j=1}^n\sum_{s=1}^m \|g_{ijs}\|_{\psi_1}^2} \geq \frac{t^2}{m\sum_{i=1}^n\sum_{j=1}^n \left\|\boldsymbol{\mathcal{A}}_{i::} * (\boldsymbol{\mathcal{C}}_{j::})^T\right\|_F^2 \left\|\boldsymbol{\mathcal{B}}_{i::} * (\boldsymbol{\mathcal{D}}_{j::})^T\right\|_F^2},$$

$$\frac{t}{\max_{ijs}\|g_{ijs}\|_{\psi_1}} \geq \frac{t}{\max_{ij} \left\|\boldsymbol{\mathcal{A}}_{i::} * (\boldsymbol{\mathcal{C}}_{j::})^T\right\|_F \left\|\boldsymbol{\mathcal{B}}_{i::} * (\boldsymbol{\mathcal{D}}_{j::})^T\right\|_F}. \tag{112}$$

Based on the sub-exponential Bernstein inequality, there is

$$\left|D\left(\boldsymbol{\mathcal{A}} * \boldsymbol{\mathcal{C}}^T, \boldsymbol{\mathcal{B}} * \boldsymbol{\mathcal{D}}^T\right)\right| \leq \frac{\sqrt{m\log(n \vee n_3)}}{2}\sqrt{\sum_{i=1}^n\sum_{j=1}^n \left\|\boldsymbol{\mathcal{A}}_{i::} * (\boldsymbol{\mathcal{C}}_{j::})^T\right\|_F^2 \left\|\boldsymbol{\mathcal{B}}_{i::} * (\boldsymbol{\mathcal{D}}_{j::})^T\right\|_F^2}$$

$$+ \frac{\log(n \vee n_3)}{2}\max_{ij} \left\|\boldsymbol{\mathcal{A}}_{i::} * (\boldsymbol{\mathcal{C}}_{j::})^T\right\|_F \left\|\boldsymbol{\mathcal{B}}_{i::} * (\boldsymbol{\mathcal{D}}_{j::})^T\right\|_F$$

$$\leq \sqrt{m}\log(n \vee n_3)\sqrt{\sum_{i=1}^n\sum_{j=1}^n \left\|\boldsymbol{\mathcal{A}}_{i::} * (\boldsymbol{\mathcal{C}}_{j::})^T\right\|_F^2 \left\|\boldsymbol{\mathcal{B}}_{i::} * (\boldsymbol{\mathcal{D}}_{j::})^T\right\|_F^2} \tag{113}$$

holds with probability at least $1 - \frac{1}{(n \vee n_3)^{10}}$. Further, we have

$$\sqrt{\sum_{i=1}^{n}\sum_{j=1}^{n}\left\|\mathcal{A}_{i::}*(\mathcal{C}_{j::})^T\right\|_F^2\left\|\mathcal{B}_{i::}*(\mathcal{D}_{j::})^T\right\|_F^2} \leq \max_{ij}\left\|\mathcal{A}_{i::}*(\mathcal{C}_{j::})^T\right\|_F\sqrt{\sum_{i=1}^{n}\sum_{j=1}^{n}\left\|\mathcal{B}_{i::}*(\mathcal{D}_{j::})^T\right\|_F^2}$$

$$\leq \left(\left(\|\mathcal{A}\|_F \max_i\|\mathcal{C}_{i::}\|\right) \wedge \left(\|\mathcal{C}\|_F \max_i\|\mathcal{A}_{i::}\|\right)\right)\sqrt{n_3\sum_{i=1}^{n}\sum_{j=1}^{n}\|\mathcal{B}_{i::}\|_F^2\|\mathcal{D}_{j::}\|_F^2}$$

$$= \sqrt{n_3}\left(\left(\|\mathcal{A}\|_F \max_i\|\mathcal{C}_{i::}\|\right) \wedge \left(\|\mathcal{C}\|_F \max_i\|\mathcal{A}_{i::}\|\right)\right)\|\mathcal{B}\|_F\|\mathcal{D}\|_F, \tag{114}$$

where the inequality is due to norm inequalities (93). $\qquad\square$

The following lemma gives the property of alignment tensor $\mathcal{R}^t$ in (11), which shares a similar result in the matrix alignment setting [32].

**Lemma 4.** *For the minimization problem* (11)*, the*

$$\left(\mathcal{A}^t * \mathcal{R}^t\right)^T * \mathcal{A}^\star + \left(\mathcal{B}^t * \mathcal{R}^t\right)^T * \mathcal{B}^\star \tag{115}$$

*is the symmetric positive semidefinite tensor and* $\left(\mathcal{A}^t * \mathcal{R}^t - \mathcal{A}^\star\right)^T * \mathcal{A}^\star + \left(\mathcal{B}^t * \mathcal{R}^t - \mathcal{B}^\star\right)^T * \mathcal{B}^\star \in \mathbb{R}^{r \times r \times n_3}$ *is also symmetric tensor.*

*Proof.* Solving (11) is equivalent to solving the following problem in the frequency domain as

$$\overline{\mathbf{R}}^t := \arg\min_{\mathcal{R} \in \mathcal{O}_r}\left\|\begin{bmatrix}\begin{bmatrix}\overline{\mathbf{A}^t}^{(1)}\\\overline{\mathbf{B}^t}^{(1)}\end{bmatrix}\overline{\mathbf{R}}^{t(1)} - \begin{bmatrix}\overline{\mathbf{A}^\star}^{(1)}\\\overline{\mathbf{B}^\star}^{(1)}\end{bmatrix} & & \\ & \ddots & \\ & & \begin{bmatrix}\overline{\mathbf{A}^t}^{(n_3)}\\\overline{\mathbf{B}^t}^{(n_3)}\end{bmatrix}\overline{\mathbf{R}}^{t(n_3)} - \begin{bmatrix}\overline{\mathbf{A}^\star}^{(n_3)}\\\overline{\mathbf{B}^\star}^{(n_3)}\end{bmatrix}\end{bmatrix}\right\|_F \tag{116}$$

Due to the block diagonal structure, to find the orthogonal alignment tensor. We just need to find each orthogonal alignment matrix for each frontal slice in the Fourier domain as

$$\overline{\mathbf{R}^t}^{(k)} := \arg\min_{\mathbf{R} \in \mathcal{O}_r}\left\|\begin{bmatrix}\overline{\mathbf{A}^t}^{(k)}\\\overline{\mathbf{B}^t}^{(k)}\end{bmatrix}\mathbf{R} - \begin{bmatrix}\overline{\mathbf{A}^\star}^{(k)}\\\overline{\mathbf{B}^\star}^{(k)}\end{bmatrix}\right\|_F, \quad \forall k \in [n_3]. \tag{117}$$

$\forall k \in [n_3]$, the (117) is the matrix orthogonal Procrustes problem [14] which concludes that the each optimal alignment matrix as

$$\overline{\mathbf{R}^t}^{(k)} = \mathbf{L}_t^{(k)}\left(\mathbf{P}_t^{(k)}\right)^T,$$

$$\left(\overline{\mathbf{A}^t}^{(k)}\right)^T\overline{\mathbf{A}^\star}^{(k)} + \left(\overline{\mathbf{B}^t}^{(k)}\right)^T\overline{\mathbf{B}^\star}^{(k)} = \overline{\mathbf{L}}_t^{(k)}\overline{\mathbf{S}}_t^{(k)}\left(\overline{\mathbf{P}}_t^{(k)}\right)^T, \tag{118}$$

where the second equation denotes the SVD decomposition of LHS. Also there is

$$\left(\overline{\mathbf{A}^\star}^{(k)}\right)^T\overline{\mathbf{A}^t}^{(k)}\overline{\mathbf{R}^t}^{(k)} + \left(\overline{\mathbf{B}^\star}^{(k)}\right)^T\overline{\mathbf{B}^t}^{(k)}\overline{\mathbf{R}^t}^{(k)} = \left(\overline{\mathbf{A}^t}^{(k)}\overline{\mathbf{R}^t}^{(k)}\right)^T\overline{\mathbf{A}^\star}^{(k)} + \left(\overline{\mathbf{B}^t}^{(k)}\overline{\mathbf{R}^t}^{(k)}\right)^T\overline{\mathbf{B}^\star}^{(k)}$$

$$= \overline{\mathbf{P}}_t^{(k)}\overline{\mathbf{S}}_t^{(k)}\left(\overline{\mathbf{P}}_t^{(k)}\right)^T. \tag{119}$$

Translating the above equation into the original domain, we obtain

$$\left(\mathcal{A}^\star\right)^T * \left(\mathcal{A}^t * \mathcal{R}^t\right) + \left(\mathcal{B}^\star\right)^T * \left(\mathcal{B}^t * \mathcal{R}^t\right) = \left(\mathcal{A}^t * \mathcal{R}^t\right)^T * \mathcal{A}^\star + \left(\mathcal{B}^t * \mathcal{R}^t\right)^T * \mathcal{B}^\star$$

$$= \boldsymbol{\mathcal{P}}_t * \boldsymbol{\mathcal{S}}_t * \boldsymbol{\mathcal{P}}_t^T, \tag{120}$$

which is the symmetric positive semidefinite tensor.

$$\left(\boldsymbol{\mathcal{A}}^t * \boldsymbol{\mathcal{R}}^t - \boldsymbol{\mathcal{A}}^\star\right)^T * \boldsymbol{\mathcal{A}}^\star + \left(\boldsymbol{\mathcal{B}}^t * \boldsymbol{\mathcal{R}}^t - \boldsymbol{\mathcal{B}}^\star\right)^T * \boldsymbol{\mathcal{B}}^\star = \left(\boldsymbol{\mathcal{A}}^t * \boldsymbol{\mathcal{R}}^t\right)^T * \boldsymbol{\mathcal{A}}^\star + \left(\boldsymbol{\mathcal{B}}^t * \boldsymbol{\mathcal{R}}^t\right)^T * \boldsymbol{\mathcal{B}}^\star$$
$$- \left((\boldsymbol{\mathcal{A}}^\star)^T * \boldsymbol{\mathcal{A}}^\star + (\boldsymbol{\mathcal{B}}^\star)^T * \boldsymbol{\mathcal{B}}^\star\right), \tag{121}$$

where the RHS of the above equation is the symmetric tensor due to (120). The general result for tensor is similar as matrix that $\boldsymbol{\mathcal{R}}^t$ is the minimizer of (11) iff that $\left(\boldsymbol{\mathcal{A}}^t * \boldsymbol{\mathcal{R}}^t\right)^T * \boldsymbol{\mathcal{A}}^\star + \left(\boldsymbol{\mathcal{B}}^t * \boldsymbol{\mathcal{R}}^t\right)^T * \boldsymbol{\mathcal{B}}^\star$ is symmetric and positive semidefinite. $\qquad \square$

The next lemma gives the perturbation bound in the tensor spectral norm under the t-SVD algebra.

**Lemma 5.** *Let* $\boldsymbol{\mathcal{C}} \in \mathbb{R}^{r \times r \times n_3}$ *be a nonsingular tensor, i.e.,* $\sigma_r(\overline{\boldsymbol{C}}^{(i)}) > 0, \forall i \in [n_3]$. *Then for any tensor* $\boldsymbol{\mathcal{E}} \in \mathbb{R}^{r \times r \times n_3}$ *with* $\|\boldsymbol{\mathcal{E}}\| \leq \sigma_{\min}(\boldsymbol{\mathcal{C}})$, *there is*

$$\|sgn(\boldsymbol{\mathcal{C}} + \boldsymbol{\mathcal{E}}) - sgn(\boldsymbol{\mathcal{C}})\| \leq \frac{\|\boldsymbol{\mathcal{E}}\|}{\sigma_{\min}(\boldsymbol{\mathcal{C}}) - \|\boldsymbol{\mathcal{E}}\|}, \tag{122}$$

*where* $sgn(\boldsymbol{\mathcal{X}}) := \boldsymbol{\mathcal{X}} * (\boldsymbol{\mathcal{X}}^T * \boldsymbol{\mathcal{X}})^{-\frac{1}{2}}$.

*Proof.*

$$\|\text{sgn}(\boldsymbol{\mathcal{C}} + \boldsymbol{\mathcal{E}}) - \text{sgn}(\boldsymbol{\mathcal{C}})\| = \left\| \overline{\boldsymbol{C}} \left(\overline{\boldsymbol{C}}^c \overline{\boldsymbol{C}}\right)^{-\frac{1}{2}} - \left(\overline{\boldsymbol{C}} + \overline{\boldsymbol{E}}\right) \left(\left(\overline{\boldsymbol{C}} + \overline{\boldsymbol{E}}\right)^c \left(\overline{\boldsymbol{C}} + \overline{\boldsymbol{E}}\right)\right)^{-\frac{1}{2}} \right\|$$

$$\stackrel{(i)}{\leq} -\log\left(1 - \frac{\|\overline{\boldsymbol{E}}\|}{\sigma_{\min}(\overline{\boldsymbol{C}})}\right)$$

$$\stackrel{(ii)}{\leq} \frac{\frac{\|\overline{\boldsymbol{E}}\|}{\sigma_{\min}(\overline{\boldsymbol{C}})}}{1 - \frac{\|\overline{\boldsymbol{E}}\|}{\sigma_{\min}(\overline{\boldsymbol{C}})}}$$

$$= \frac{\|\boldsymbol{\mathcal{E}}\|}{\sigma_{\min}(\boldsymbol{\mathcal{C}}) - \|\boldsymbol{\mathcal{E}}\|}, \tag{123}$$

where (i) is based on Theorem 3.2 in [35], which is a perturbation bound for the complex matrices. (ii) is based on logarithm inequality $\log(1 + x) \geq \frac{x}{1+x}, \forall x > -1$. $\qquad \square$

To bound the distance between two tensors after transformation based on the corresponding orthogonal alignment tensor, we need the following lemma, which is a generalized version of the matrix setting in [32] and based on Lemma 5.

**Lemma 6.** *Assume the tensors* $\boldsymbol{\mathcal{X}}_0, \boldsymbol{\mathcal{X}}_1, \boldsymbol{\mathcal{X}}_2 \in \mathbb{R}^{n \times r \times n_3}$ *satisfying*

$$\|\boldsymbol{\mathcal{X}}_1 - \boldsymbol{\mathcal{X}}_0\| \|\boldsymbol{\mathcal{X}}_0\| \leq \frac{\sigma_{\min}^2(\boldsymbol{\mathcal{X}}_0)}{2}, \quad \|\boldsymbol{\mathcal{X}}_1 - \boldsymbol{\mathcal{X}}_2\| \|\boldsymbol{\mathcal{X}}_0\| \leq \frac{\sigma_{\min}^2(\boldsymbol{\mathcal{X}}_0)}{4}. \tag{124}$$

*Denote the alignment tensors as*

$$\boldsymbol{\mathcal{R}}_1 := \underset{\boldsymbol{\mathcal{R}} \in \mathcal{O}_r}{\arg\min} \|\boldsymbol{\mathcal{X}}_1 * \boldsymbol{\mathcal{R}} - \boldsymbol{\mathcal{X}}_0\|_F$$
$$\boldsymbol{\mathcal{R}}_2 := \underset{\boldsymbol{\mathcal{R}} \in \mathcal{O}_r}{\arg\min} \|\boldsymbol{\mathcal{X}}_2 * \boldsymbol{\mathcal{R}} - \boldsymbol{\mathcal{X}}_0\|_F. \tag{125}$$

*Then there is*

$$\|\boldsymbol{\mathcal{X}}_1 * \boldsymbol{\mathcal{R}}_1 - \boldsymbol{\mathcal{X}}_2 * \boldsymbol{\mathcal{R}}_2\| \leq 9 \left(\frac{\sigma_{\max}(\boldsymbol{\mathcal{X}}_0)}{\sigma_{\min}(\boldsymbol{\mathcal{X}}_0)}\right)^2 \|\boldsymbol{\mathcal{X}}_1 - \boldsymbol{\mathcal{X}}_2\|. \tag{126}$$

This can be proved based on the proof for the matrix setting in [32].

$$\|\boldsymbol{\mathcal{X}}_1 * \boldsymbol{\mathcal{R}}_1 - \boldsymbol{\mathcal{X}}_2 * \boldsymbol{\mathcal{R}}_2\| = \|(\boldsymbol{\mathcal{X}}_1 - \boldsymbol{\mathcal{X}}_2) * \boldsymbol{\mathcal{R}}_2 + \boldsymbol{\mathcal{X}}_1 * (\boldsymbol{\mathcal{R}}_1 - \boldsymbol{\mathcal{R}}_2)\|$$

$$\leq \|\mathcal{X}_1 - \mathcal{X}_2\| + \|\mathcal{X}_1\| \|\mathcal{R}_1 - \mathcal{R}_2\|$$
$$\leq \|\mathcal{X}_1 - \mathcal{X}_2\| + 2\|\mathcal{X}_0\| \|\mathcal{R}_1 - \mathcal{R}_2\|, \tag{127}$$

where the last inequality is due to the second condition in (124). Now we come to bound the distance between two alignment tensors. Based on (125), $\forall k \in [n_3]$, there is

$$\overline{\boldsymbol{R}}_1^{(k)} = \underset{\boldsymbol{R} \in \boldsymbol{O}_r}{\arg\min} \left\| \overline{\boldsymbol{X}_1}^{(k)} \boldsymbol{R} - \overline{\boldsymbol{X}_0}^{(k)} \right\|_F \tag{128}$$

$$\overline{\boldsymbol{R}}_2^{(k)} = \underset{\boldsymbol{R} \in \boldsymbol{O}_r}{\arg\min} \left\| \overline{\boldsymbol{X}_2}^{(k)} \boldsymbol{R} - \overline{\boldsymbol{X}_0}^{(k)} \right\|_F. \tag{129}$$

Denote the $\mathcal{C} := \mathcal{X}_1^T * \mathcal{X}_0$ and $\mathcal{C} + \mathcal{E} := \mathcal{X}_2^T * \mathcal{X}_0$, there is

$$\left\| \overline{\boldsymbol{C}} - \overline{\boldsymbol{X}_0^T \boldsymbol{X}_0} \right\| = \left\| \mathcal{C} - \mathcal{X}_0^T * \mathcal{X}_0 \right\| \leq \frac{\sigma_{\min}^2(\mathcal{X}_0)}{2}. \tag{130}$$

Based on Wely's inequality, we can know $\sigma_{\min}(\overline{\boldsymbol{C}}) \geq \frac{\sigma_{\min}^2(\mathcal{X}_0)}{2}$ which means that all $\overline{\boldsymbol{C}}^{(k))}$ are invertible. Thus, $\mathcal{C}$ is an invertible tensor. In addition,

$$\left\| \overline{\boldsymbol{E}} \right\| = \left\| \left( \overline{\boldsymbol{X}_2}^T - \overline{\boldsymbol{X}_1} \right)^T \overline{\boldsymbol{X}_0} \right\| \leq \left\| \overline{\boldsymbol{X}_2} - \overline{\boldsymbol{X}_1} \right\| \left\| \overline{\boldsymbol{X}_0} \right\| = \|\mathcal{X}_1 - \mathcal{X}_2\| \|\mathcal{X}_0\| \leq \frac{\sigma_{\min}^2(\mathcal{X}_0)}{4}. \tag{131}$$

Thus, $\overline{\boldsymbol{C}} + \overline{\boldsymbol{E}}$ is invertible and $\overline{\boldsymbol{R}}_1^{(k)}, \overline{\boldsymbol{R}}_2^{(k)}$ have closed formulation as

$$\overline{\boldsymbol{R}}_1^{(k)} = \overline{\boldsymbol{C}}^{(k)} \left( \left( \overline{\boldsymbol{C}}^{(k)} \right)^T \overline{\boldsymbol{C}}^{(k)} \right)^{-\frac{1}{2}}, \quad \overline{\boldsymbol{R}}_2^{(k)} = \left( \overline{\boldsymbol{C}}^{(k)} + \overline{\boldsymbol{E}}^{(k)} \right) \left( \left( \overline{\boldsymbol{C}}^{(k)} + \overline{\boldsymbol{E}}^{(k)} \right)^T \left( \overline{\boldsymbol{C}}^{(k)} + \overline{\boldsymbol{E}}^{(k)} \right) \right)^{-\frac{1}{2}} \tag{132}$$

Therefore, there is

$$\|\mathcal{R}_1 - \mathcal{R}_2\| = \left\| \overline{\boldsymbol{R}_1} - \overline{\boldsymbol{R}_2} \right\|$$
$$= \left\| \overline{\boldsymbol{C}} \left( \overline{\boldsymbol{C}}^T \overline{\boldsymbol{C}} \right)^{-\frac{1}{2}} - \left( \overline{\boldsymbol{C}} + \overline{\boldsymbol{E}} \right) \left( \left( \overline{\boldsymbol{C}} + \overline{\boldsymbol{E}} \right)^T \left( \overline{\boldsymbol{C}} + \overline{\boldsymbol{E}} \right) \right)^{-\frac{1}{2}} \right\|$$
$$\overset{(i)}{\leq} \frac{\|\mathcal{E}\|}{\sigma_{\min}(\mathcal{C}) - \|\mathcal{E}\|}$$
$$\leq \frac{\|\mathcal{X}_1 - \mathcal{X}_2\| \|\mathcal{X}_0\|}{\frac{\sigma_{\min}^2(\mathcal{X}_0)}{2} - \frac{\sigma_{\min}^2(\mathcal{X}_0)}{4}}, \tag{133}$$

where (i) is based on Lemma 5 and $\left\| \overline{\boldsymbol{E}} \right\| \leq \frac{\sigma_{\min}^2(\mathcal{X}_0)}{4} < \frac{\sigma_{\min}^2(\mathcal{X}_0)}{2} \leq \sigma_{\min}(\overline{\boldsymbol{C}})$, which also induce the last inequality with (131). Substituting above inequality into (127) would have

$$\|\mathcal{X}_1 * \mathcal{R}_1 - \mathcal{X}_2 * \mathcal{R}_2\| \leq \|\mathcal{X}_1 - \mathcal{X}_2\| + \frac{2\|\mathcal{X}_0\|^2}{\frac{\sigma_{\min}^2(\mathcal{X}_0)}{4}} \|\mathcal{X}_1 - \mathcal{X}_2\|$$
$$\leq \left( 1 + 8\frac{\sigma_{\max}^2(\mathcal{X}_0)}{\sigma_{\min}^2(\mathcal{X}_0)} \right) \|\mathcal{X}_1 - \mathcal{X}_2\|$$
$$\leq 9\frac{\sigma_{\max}^2(\mathcal{X}_0)}{\sigma_{\min}^2(\mathcal{X}_0)} \|\mathcal{X}_1 - \mathcal{X}_2\|. \tag{134}$$

The following lemma provides the benign landscape of the non-convex problem (6), which has restricted strong convexity and smoothness properties in incoherent regions near the global optimum $\mathcal{X}^\star$. The sample complexity in (12) is sufficient to guarantee the existence of this benign landscape, which can obtain the linear rate.

**Lemma 7.** *With the same setting as Theorem 2, if the tube-wise sample complexity satisfies* (12)*, then*

$$Vec\left( \begin{bmatrix} \mathcal{D}_{\mathcal{A}} \\ \mathcal{D}_{\mathcal{B}} \end{bmatrix} \right)^T Mat\left( \nabla^2 f(\mathcal{A}, \mathcal{B}) \right) Vec\left( \begin{bmatrix} \mathcal{D}_{\mathcal{A}} \\ \mathcal{D}_{\mathcal{B}} \end{bmatrix} \right) \geq \frac{\sigma_{\min}}{5} \left\| \begin{bmatrix} \mathcal{D}_{\mathcal{A}} \\ \mathcal{D}_{\mathcal{B}} \end{bmatrix} \right\|_F^2 \quad and \quad \left\| \nabla^2 f(\mathcal{A}, \mathcal{B}) \right\| \leq 5\sigma_{\max} \tag{135}$$

*hold with probability at least $1 - \frac{1}{(n \vee n_3)^{10}}$ for $\forall \mathcal{A}, \mathcal{B} \in \mathbb{R}^{n \times r \times n_3}$ that satisfy*

$$\max_{l \in [2n]} \left\| \left( \begin{bmatrix} \mathcal{A} - \mathcal{U}^\star \\ \mathcal{B} - \mathcal{V}^\star \end{bmatrix} \right)_{l::} \right\| \leq \frac{\sqrt{\sigma_{\max}}}{500\kappa\sqrt{2n}} \tag{136}$$

*and $\forall \mathcal{D}_{\mathcal{A}}, \mathcal{D}_{\mathcal{B}} \in \mathbb{R}^{n \times r \times n_3}$ stay in set*

$$\left\{ \begin{bmatrix} \mathcal{A}_1 \\ \mathcal{B}_1 \end{bmatrix} * \hat{\mathcal{R}} - \begin{bmatrix} \mathcal{A}_2 \\ \mathcal{B}_2 \end{bmatrix} : \left\| \begin{bmatrix} \mathcal{A}_2 \\ \mathcal{B}_2 \end{bmatrix} - \begin{bmatrix} \mathcal{U}^\star \\ \mathcal{V}^\star \end{bmatrix} \right\| \leq \frac{\sqrt{\sigma_{\max}}}{500\kappa}, \hat{\mathcal{R}} := \arg\min_{\mathcal{R} \in \mathcal{O}_r} \left\| \begin{bmatrix} \mathcal{A}_1 \\ \mathcal{B}_1 \end{bmatrix} * \mathcal{R} - \begin{bmatrix} \mathcal{A}_2 \\ \mathcal{B}_2 \end{bmatrix} \right\|_F \right\}. \tag{137}$$

*Proof.* Based on (6), there is definition $\Delta_{\mathcal{A}} := \mathcal{A} - \mathcal{U}^\star$, $\Delta_{\mathcal{B}} := \mathcal{B} - \mathcal{V}^\star$ and population level has expression as

$$\mathbb{E}\left[ \text{Vec}\left( \begin{bmatrix} \mathcal{D}_{\mathcal{A}} \\ \mathcal{D}_{\mathcal{B}} \end{bmatrix} \right)^T \text{Mat}\left( \nabla^2 f(\mathcal{A}, \mathcal{B}) \right) \text{Vec}\left( \begin{bmatrix} \mathcal{D}_{\mathcal{A}} \\ \mathcal{D}_{\mathcal{B}} \end{bmatrix} \right) \right] = m \left( \|\mathcal{D}_{\mathcal{A}} * \mathcal{V}^\star\|_F^2 + \left\| \mathcal{U}^\star * \mathcal{D}_{\mathcal{B}}^T \right\|_F^2 \right.$$

$$\left. + \frac{1}{2}\left\| \mathcal{D}_{\mathcal{A}}^T * \mathcal{U}^\star - \mathcal{D}_{\mathcal{B}}^T * \mathcal{V}^\star \right\|_F^2 + \frac{1}{2}\left\| \mathcal{A}_2^T * \mathcal{D}_{\mathcal{A}} + \mathcal{B}_2^T * \mathcal{D}_{\mathcal{B}} \right\|_F^2 + \mathcal{W}_1 + \mathcal{W}_2 \right), \tag{138}$$

where the $\mathcal{W}_1 + \mathcal{W}_2$ has following bound

$$|\mathcal{W}_1 + \mathcal{W}_2| \leq 9 \left[ (\|\mathcal{U}^\star - \mathcal{A}_2\| + \|\mathcal{V}^\star - \mathcal{B}_2\|) \cdot (\|\mathcal{A}_2\| + \|\mathcal{B}_2\|) + (\|\mathcal{U}^\star - \mathcal{A}_2\| + \|\mathcal{V}^\star - \mathcal{B}_2\|)^2 \right.$$

$$\left. + (\|\Delta_{\mathcal{A}}\| + \|\Delta_{\mathcal{B}}\|) \cdot (\|\mathcal{U}^\star\| + \|\mathcal{V}^\star\|) + (\|\Delta_{\mathcal{A}}\| + \|\Delta_{\mathcal{B}}\|)^2 \right] \cdot \left( \|\mathcal{D}_{\mathcal{A}}\|_F^2 + \|\mathcal{D}_{\mathcal{B}}\|_F^2 \right). \tag{139}$$

Due to spectral norm inequality (92), there is

$$\left\| \begin{bmatrix} \mathcal{A} - \mathcal{U}^\star \\ \mathcal{B} - \mathcal{V}^\star \end{bmatrix} \right\| \leq \sqrt{\sum_{i=1}^{2n} \left\| \left( \begin{bmatrix} \mathcal{A} - \mathcal{U}^\star \\ \mathcal{B} - \mathcal{V}^\star \end{bmatrix} \right)_{i::} \right\|^2}$$

$$\leq \max_{l \in [2n]} \sqrt{2n} \left\| \left( \begin{bmatrix} \mathcal{A} - \mathcal{U}^\star \\ \mathcal{B} - \mathcal{V}^\star \end{bmatrix} \right)_{l::} \right\|$$

$$\leq \frac{\sqrt{\sigma_{\max}}}{500\kappa}. \tag{140}$$

Thus, based on $\left\| \begin{bmatrix} \mathcal{A}_2 \\ \mathcal{B}_2 \end{bmatrix} - \begin{bmatrix} \mathcal{U}^\star \\ \mathcal{V}^\star \end{bmatrix} \right\| \leq \frac{\sqrt{\sigma_{\max}}}{500\kappa}$ and above inequality, there is

$$|\mathcal{W}_1 + \mathcal{W}_2| \leq \frac{\sigma_{\min}}{5} \left\| \begin{bmatrix} \mathcal{D}_{\mathcal{A}} \\ \mathcal{D}_{\mathcal{B}} \end{bmatrix} \right\|_F^2. \tag{141}$$

Next, we need to bound the difference between the population level and the empirical level by the following decomposition as

$$\text{Vec}\left( \begin{bmatrix} \mathcal{D}_{\mathcal{A}} \\ \mathcal{D}_{\mathcal{B}} \end{bmatrix} \right)^T \text{Mat}\left( \nabla^2 f(\mathcal{A}, \mathcal{B}) \right) \text{Vec}\left( \begin{bmatrix} \mathcal{D}_{\mathcal{A}} \\ \mathcal{D}_{\mathcal{B}} \end{bmatrix} \right) - \mathbb{E}\left[ \text{Vec}\left( \begin{bmatrix} \mathcal{D}_{\mathcal{A}} \\ \mathcal{D}_{\mathcal{B}} \end{bmatrix} \right)^T \text{Mat}\left( \nabla^2 f(\mathcal{A}, \mathcal{B}) \right) \text{Vec}\left( \begin{bmatrix} \mathcal{D}_{\mathcal{A}} \\ \mathcal{D}_{\mathcal{B}} \end{bmatrix} \right) \right]$$

$$= 2 \left\langle \mathcal{Z}^c \mathcal{Z} \left( \mathcal{A} * \mathcal{B}^T - \mathcal{U}^\star * (\mathcal{V}^\star)^T \right), \mathcal{D}_{\mathcal{A}} * \mathcal{D}_{\mathcal{B}}^T \right\rangle - 2 \left\langle \mathcal{A} * \mathcal{B}^T - \mathcal{U}^\star * (\mathcal{V}^\star)^T, \mathcal{D}_{\mathcal{A}} * \mathcal{D}_{\mathcal{B}}^T \right\rangle$$

$$+ \left\| \mathcal{Z} \left( \mathcal{D}_{\mathcal{A}} * \mathcal{B}^T + \mathcal{A} * \mathcal{D}_{\mathcal{B}}^T \right) \right\|_F^2 - \left\| \mathcal{D}_{\mathcal{A}} * \mathcal{B}^T + \mathcal{A} * \mathcal{D}_{\mathcal{B}}^T \right\|_F^2$$

$$= \Pi, \tag{142}$$

where $\Pi$ has decomposition as

$$\Pi = 2D \left( \Delta_{\mathcal{A}} * (\mathcal{V}^\star)^T, \mathcal{D}_{\mathcal{A}} * \mathcal{D}_{\mathcal{B}}^T \right) + 2D \left( \mathcal{U}^\star * \Delta_{\mathcal{B}}^T, \mathcal{D}_{\mathcal{A}} * \mathcal{D}_{\mathcal{B}}^T \right) + 2D \left( \Delta_{\mathcal{A}} * \Delta_{\mathcal{B}}^T, \mathcal{D}_{\mathcal{A}} * \mathcal{D}_{\mathcal{B}}^T \right)$$

$$+ 2D\left(\mathcal{D}_{\mathcal{A}} * (\mathcal{V}^\star)^T, \mathbf{\Delta}_{\mathcal{A}} * \mathcal{D}_{\mathcal{B}}^T\right) + 2D\left(\mathcal{D}_{\mathcal{A}} * \mathbf{\Delta}_{\mathcal{B}}^T, \mathcal{U}^\star * \mathcal{D}_{\mathcal{B}}^T\right) + 2D\left(\mathcal{D}_{\mathcal{A}} * \mathbf{\Delta}_{\mathcal{B}}^T, \mathbf{\Delta}_{\mathcal{A}} * \mathcal{D}_{\mathcal{B}}^T\right);$$

$$+ D\left(\mathcal{D}_{\mathcal{A}} * (\mathcal{V}^\star)^T, \mathcal{D}_{\mathcal{A}} * (\mathcal{V}^\star)^T\right) + D\left(\mathcal{U}^\star * \mathcal{D}_{\mathcal{B}}^T, \mathcal{U}^\star * \mathcal{D}_{\mathcal{B}}^T\right) + 2D\left(\mathcal{D}_{\mathcal{A}} * (\mathcal{V}^\star)^T, \mathcal{U}^\star * \mathcal{D}_{\mathcal{B}}^T\right);$$

$$+ D\left(\mathcal{D}_{\mathcal{A}} * \mathbf{\Delta}_{\mathcal{B}}^T, \mathcal{D}_{\mathcal{A}} * \mathbf{\Delta}_{\mathcal{B}}^T\right) + D\left(\mathbf{\Delta}_{\mathcal{A}} * \mathcal{D}_{\mathcal{B}}^T, \mathbf{\Delta}_{\mathcal{A}} * \mathcal{D}_{\mathcal{B}}^T\right) + 2D\left(\mathcal{D}_{\mathcal{A}} * (\mathcal{V}^\star)^T, \mathcal{D}_{\mathcal{A}} * \mathbf{\Delta}_{\mathcal{B}}^T\right)$$

$$+ 2D\left(\mathcal{U}^\star * \mathcal{D}_{\mathcal{B}}^T, \mathbf{\Delta}_{\mathcal{A}} * \mathcal{D}_{\mathcal{B}}^T\right). \tag{143}$$

The concentration distance $D\left(\mathcal{A} * \mathcal{C}^T, \mathcal{B} * \mathcal{D}^T\right)$ is defined in Lemma 3. Based on the Lemma 3, we can bound the concentration level $\Pi$ as follows

$$|\Pi| \le m\sqrt{\frac{n_3}{m}} \log(n \vee n_3) \cdot \left(2\|\mathcal{V}^\star\|_F \max_i \|(\mathbf{\Delta}_{\mathcal{A}})_{i::}\| \|\mathcal{D}_{\mathcal{A}}\|_F \|\mathcal{D}_{\mathcal{B}}\|_F + 2\|\mathcal{U}^\star\|_F \max_i \|(\mathbf{\Delta}_{\mathcal{B}})_{i::}\| \|\mathcal{D}_{\mathcal{A}}\|_F \right.$$
$$\cdot \|\mathcal{D}_{\mathcal{B}}\|_F + 2\|\mathbf{\Delta}_{\mathcal{A}}\|_F \max_i \|(\mathbf{\Delta}_{\mathcal{B}})_{i::}\| \|\mathcal{D}_{\mathcal{A}}\|_F \|\mathcal{D}_{\mathcal{B}}\|_F + 2\|\mathcal{D}_{\mathcal{A}}\|_F \max_i \|\mathcal{V}_{i::}^\star\| \|\mathbf{\Delta}_{\mathcal{A}}\|_F \|\mathcal{D}_{\mathcal{B}}\|_F \tag{144}$$

$$+ 2\|\mathcal{D}_{\mathcal{A}}\|_F \max_i \|(\mathbf{\Delta}_{\mathcal{B}})_{i::}\| \|\mathcal{U}^\star\|_F \|\mathcal{D}_{\mathcal{B}}\|_F + 2\|\mathcal{D}_{\mathcal{A}}\|_F \max_i \|(\mathbf{\Delta}_{\mathcal{B}})_{i::}\| \|\mathbf{\Delta}_{\mathcal{A}}\|_F \|\mathcal{D}_{\mathcal{B}}\|_F$$

$$+ \|\mathcal{D}_{\mathcal{A}}\|_F \max_i \|\mathcal{V}_{i::}^\star\| \|\mathcal{V}^\star\|_F \|\mathcal{D}_{\mathcal{A}}\|_F + \max_i \|\mathcal{U}_{i::}^\star\| \|\mathcal{D}_{\mathcal{B}}\|_F \|\mathcal{U}^\star\|_F \|\mathcal{D}_{\mathcal{B}}\|_F$$

$$+ 2\|\mathcal{D}_{\mathcal{A}}\|_F \max_i \|\mathcal{V}_{i::}^\star\| \|\mathcal{U}^\star\|_F \|\mathcal{D}_{\mathcal{B}}\|_F + \|\mathcal{D}_{\mathcal{A}}\|_F \max_i \|(\mathbf{\Delta}_{\mathcal{B}})_{i::}\| \|\mathcal{D}_{\mathcal{A}}\|_F \|\mathbf{\Delta}_{\mathcal{B}}\|_F$$

$$+ \max_i \|(\mathbf{\Delta}_{\mathcal{A}})_{i::}\| \|\mathcal{D}_{\mathcal{B}}\|_F \|\mathbf{\Delta}_{\mathcal{A}}\|_F \|\mathcal{D}_{\mathcal{B}}\|_F + 2\|\mathcal{D}_{\mathcal{A}}\|_F \max_i \|\mathcal{V}_{i::}^\star\| \|\mathcal{D}_{\mathcal{A}}\|_F \|\mathbf{\Delta}_{\mathcal{B}}\|_F$$

$$\left. + 2\max_i \|\mathcal{U}_{i::}^\star\| \|\mathcal{D}_{\mathcal{B}}\|_F \|\mathbf{\Delta}_{\mathcal{A}}\|_F \|\mathcal{D}_{\mathcal{B}}\|_F\right)$$

$$\overset{(i)}{\le} m\sqrt{\frac{n_3}{m}} \log(n \vee n_3) \cdot \left(3\|\mathcal{U}^\star\|_F \max_l \|\mathbf{\Delta}_{l::}\| + 3\max_i \|\mathcal{U}_{i::}^\star\| \|\mathbf{\Delta}\|_F + 3\|\mathbf{\Delta}\|_F \max_l \|\mathbf{\Delta}_{l::}\|\right.$$

$$\left. + 2\max_i \|\mathcal{U}_{i::}^\star\| \|\mathcal{U}^\star\|_F\right) \cdot \left(\|\mathcal{D}_{\mathcal{A}}\|_F^2 + \|\mathcal{D}_{\mathcal{B}}\|_F^2\right)$$

$$\overset{(ii)}{\le} m\sqrt{\frac{n_3}{m}} \log(n \vee n_3) \cdot \left(3\sqrt{n}\sqrt{\frac{\mu r \sigma_{\max}}{n}} \cdot \frac{\sqrt{\sigma_{\max}}}{500\kappa\sqrt{2n}} + 3\sqrt{\frac{\mu r \sigma_{\max}}{n}} \cdot \frac{\sqrt{\sigma_{\max}}}{500\kappa} + 3\frac{\sqrt{\sigma_{\max}}}{500\kappa}\right.$$

$$\left. \cdot \frac{\sqrt{\sigma_{\max}}}{500\kappa\sqrt{2n}} + 2\sqrt{\frac{\mu r \sigma_{\max}}{n}} \cdot \sqrt{n}\sqrt{\frac{\mu r \sigma_{\max}}{n}}\right) \cdot \left(\|\mathcal{D}_{\mathcal{A}}\|_F^2 + \|\mathcal{D}_{\mathcal{B}}\|_F^2\right)$$

$$\le \frac{m\sigma_{\min}}{5}\left(\|\mathcal{D}_{\mathcal{A}}\|_F^2 + \|\mathcal{D}_{\mathcal{B}}\|_F^2\right), \tag{145}$$

where (i) is due to inequality $2\|\mathcal{D}_{\mathcal{A}}\|_F \|\mathcal{D}_{\mathcal{B}}\|_F \le \|\mathcal{D}_{\mathcal{A}}\|_F^2 + \|\mathcal{D}_{\mathcal{B}}\|_F^2$. (ii) is due to the tensor incoherence condition (10), (136), (140) and tensor norm inequality $\|\mathcal{X}\|_F = \sqrt{\sum_{i=1}^n \|\mathcal{X}_{i::}\|_F^2} \le \sqrt{\sum_{i=1}^n \|\mathcal{X}_{i::}\|^2} \le \sqrt{n}\max_i \|\mathcal{X}_{i::}\|$. The last inequality is due to the tube-wise sample complexity in (12). $\qquad\square$

Lemma 8 claims that the distance between two consecutive alignment tensors can be well bounded, which would be useful in proving (22).

**Lemma 8.** *With the same setting as the main Theorem 2, then*

$$\left\|\left(\mathcal{R}^{t,l}\right)^{-1} * \mathcal{R}^{t+1,l} - \mathcal{I}_r\right\| \le 2 \times 1250^2 m\eta C^2 \rho^{2t}\left(\sqrt{\frac{(\mu r)^4 \kappa^8 n_3 \log^2(n \vee n_3)\sigma_{\max}}{nm}}\right)^2 \tag{146}$$

*holds with probability at least $1 - \frac{1}{5(n \vee n_3)^{10}}$.*

*Proof.* This proof follows the proof of Claim 11 in [8] and the proof of Lemma 12 in [32], which makes us use the proof idea for (13) and then utilize the tensor perturbation bound in Lemma 5. Defining the auxiliary iterations as (38)

$$\tilde{\mathcal{A}}^{t+1,l} = \tilde{\mathcal{A}}^{t+1,l} * \mathcal{R}^{t,l} - \eta\mathcal{Z}_{-l,\cdot}^c \mathcal{Z}_{-l,\cdot}\left(\mathbf{\Delta}_{\mathcal{X}}^{t,l}\right) * \mathcal{B}^\star - \eta\mathcal{P}_{l,\cdot}\left(\mathbf{\Delta}_{\mathcal{X}}^{t,l}\right) * \mathcal{B}^\star$$

$$-\frac{m\eta}{2}\mathcal{A}^\star * \left(\mathcal{R}^{t,l}\right)^T * \left(\left(\mathcal{A}^{t,l}\right)^T * \mathcal{A}^{t,l} - \left(\mathcal{B}^{t,l}\right)^T * \mathcal{B}^{t,l}\right) * \mathcal{R}^{t,l}$$

$$\tilde{\mathcal{B}}^{t+1,l} = \tilde{\mathcal{B}}^{t+1,l} * \mathcal{R}^{t,l} - \eta\left(\mathcal{Z}^c_{-l,\cdot}\mathcal{Z}_{-l,\cdot}\left(\Delta^{t,l}_{\mathcal{X}}\right)\right)^T * \mathcal{A}^\star - \eta\left(\mathcal{P}_{l,\cdot}\left(\Delta^{t,l}_{\mathcal{X}}\right)\right)^T * \mathcal{A}^\star$$

$$-\frac{m\eta}{2}\mathcal{B}^\star * \left(\mathcal{R}^{t,l}\right)^T * \left(\left(\mathcal{B}^{t,l}\right)^T * \mathcal{B}^{t,l} - \left(\mathcal{A}^{t,l}\right)^T * \mathcal{A}^{t,l}\right) * \mathcal{R}^{t,l}. \tag{147}$$

Based on definition of $\mathcal{R}^{t,l}$ in (21), we can conclude

$$\left(\mathcal{R}^{t,l}\right)^{-1} * \mathcal{R}^{t+1,l} = \arg\min_{\mathcal{R}\in\mathcal{O}_r} \left\|\begin{bmatrix}\mathcal{A}^{t+1,l} * \mathcal{R}^{t,l} \\ \mathcal{B}^{t+1,l} * \mathcal{R}^{t,l}\end{bmatrix} * \mathcal{R} - \begin{bmatrix}\mathcal{A}^\star \\ \mathcal{B}^\star\end{bmatrix}\right\|_F$$

$$\stackrel{(i)}{=} (\mathcal{C}+\mathcal{E}) * \left((\mathcal{C}+\mathcal{E})^T * (\mathcal{C}+\mathcal{E})\right)^{\frac{1}{2}}$$

$$= \mathrm{sgn}(\mathcal{C}+\mathcal{E}) \tag{148}$$

where the (i) is due to Eq.(134) in [32] in frequency domain and (117) and

$$\mathcal{C} = (\mathcal{A}^\star)^T * \tilde{\mathcal{A}}^{t+1,l} + (\mathcal{B}^\star)^T * \tilde{\mathcal{B}}^{t+1,l} \tag{149}$$

$$\mathcal{E} = \left(\mathcal{A}^{t+1,l} * \mathcal{R}^{t,l} - \tilde{\mathcal{A}}^{t+1,l}\right)^T * \mathcal{A}^\star + \left(\mathcal{B}^{t+1,l} * \mathcal{R}^{t,l} - \tilde{\mathcal{B}}^{t+1,l}\right)^T * \mathcal{B}^\star. \tag{150}$$

To use the Lemma 5, we need to prove

$$\mathcal{I}_r = \arg\min_{\mathcal{R}\in\mathcal{O}_r} \left\|\begin{bmatrix}\tilde{\mathcal{A}}^{t+1,l} \\ \tilde{\mathcal{B}}^{t+1,l}\end{bmatrix} * \mathcal{R} - \begin{bmatrix}\mathcal{A}^\star \\ \mathcal{B}^\star\end{bmatrix}\right\|_F \tag{151}$$

which is equivalent to $\mathrm{sgn}(\mathcal{C}) = \mathcal{I}_r$. Then we can use the Lemma 5 as

$$\left\|\left(\mathcal{R}^{t,l}\right)^{-1} * \mathcal{R}^{t+1,l} - \mathcal{I}_r\right\| = \|\mathrm{sgn}(\mathcal{C}+\mathcal{E}) - \mathrm{sgn}(\mathcal{C})\|$$

$$\leq \frac{\left\|\begin{bmatrix}\mathcal{A}^{t+1,l} * \mathcal{R}^{t,l} - \tilde{\mathcal{A}}^{t+1,l} \\ \mathcal{B}^{t+1,l} * \mathcal{R}^{t,l} - \tilde{\mathcal{B}}^{t+1,l}\end{bmatrix}^T * \begin{bmatrix}\mathcal{A}^\star \\ \mathcal{B}^\star\end{bmatrix}\right\|}{\sigma_{\min}\left(\begin{bmatrix}\tilde{\mathcal{A}}^{t+1,l} \\ \tilde{\mathcal{B}}^{t+1,l}\end{bmatrix}^T * \begin{bmatrix}\mathcal{A}^\star \\ \mathcal{B}^\star\end{bmatrix}\right) - \left\|\begin{bmatrix}\mathcal{A}^{t+1,l} * \mathcal{R}^{t,l} - \tilde{\mathcal{A}}^{t+1,l} \\ \mathcal{B}^{t+1,l} * \mathcal{R}^{t,l} - \tilde{\mathcal{B}}^{t+1,l}\end{bmatrix}^T * \begin{bmatrix}\mathcal{A}^\star \\ \mathcal{B}^\star\end{bmatrix}\right\|}$$

$$\leq \frac{\left\|\begin{bmatrix}\mathcal{A}^{t+1,l} * \mathcal{R}^{t,l} - \tilde{\mathcal{A}}^{t+1,l} \\ \mathcal{B}^{t+1,l} * \mathcal{R}^{t,l} - \tilde{\mathcal{B}}^{t+1,l}\end{bmatrix}\right\| \left\|\begin{bmatrix}\mathcal{A}^\star \\ \mathcal{B}^\star\end{bmatrix}\right\|}{\sigma_{\min}\left(\begin{bmatrix}\tilde{\mathcal{A}}^{t+1,l} \\ \tilde{\mathcal{B}}^{t+1,l}\end{bmatrix}^T * \begin{bmatrix}\mathcal{A}^\star \\ \mathcal{B}^\star\end{bmatrix}\right) - \left\|\begin{bmatrix}\mathcal{A}^{t+1,l} * \mathcal{R}^{t,l} - \tilde{\mathcal{A}}^{t+1,l} \\ \mathcal{B}^{t+1,l} * \mathcal{R}^{t,l} - \tilde{\mathcal{B}}^{t+1,l}\end{bmatrix}^T * \begin{bmatrix}\mathcal{A}^\star \\ \mathcal{B}^\star\end{bmatrix}\right\|}$$

$$\tag{152}$$

To prove the (151), we should prove that $\mathcal{C}$ is the symmetric positive definite tensor and bound $\left\|\begin{bmatrix}\mathcal{A}^{t+1,l} * \mathcal{R}^{t,l} - \tilde{\mathcal{A}}^{t+1,l} \\ \mathcal{B}^{t+1,l} * \mathcal{R}^{t,l} - \tilde{\mathcal{B}}^{t+1,l}\end{bmatrix}\right\|$.

We first bound the $\left\|\begin{bmatrix}\mathcal{A}^{t+1,l} * \mathcal{R}^{t,l} - \tilde{\mathcal{A}}^{t+1,l} \\ \mathcal{B}^{t+1,l} * \mathcal{R}^{t,l} - \tilde{\mathcal{B}}^{t+1,l}\end{bmatrix}\right\|$ as

$$\left\|\begin{bmatrix}\mathcal{A}^{t+1,l} * \mathcal{R}^{t,l} - \tilde{\mathcal{A}}^{t+1,l} \\ \mathcal{B}^{t+1,l} * \mathcal{R}^{t,l} - \tilde{\mathcal{B}}^{t+1,l}\end{bmatrix}\right\| = \eta\left\|\begin{bmatrix}\left(-\mathcal{Z}^c\mathcal{Z} + \mathcal{Z}^c_{l,\cdot}\mathcal{Z}_{l,\cdot} - m\mathcal{P}_{l,\cdot}\right)\left(\Delta^{t,l}_{\mathcal{X}}\right) * \Delta^{t,l}_{\mathcal{A}} \\ \left(\left(-\mathcal{Z}^c\mathcal{Z} + \mathcal{Z}^c_{l,\cdot}\mathcal{Z}_{l,\cdot} - m\mathcal{P}_{l,\cdot}\right)\left(\Delta^{t,l}_{\mathcal{X}}\right)\right)^T * \Delta^{t,l}_{\mathcal{B}}\end{bmatrix}\right\|$$

$$+ \frac{\eta}{2}\left\|\begin{bmatrix}\Delta^{t,l}_{\mathcal{A}} * \left(\mathcal{R}^{t,l}\right)^T * \left(\left(\mathcal{A}^{t,l}\right)^T * \mathcal{A}^{t,l} - \left(\mathcal{B}^{t,l}\right)^T * \mathcal{B}^{t,l}\right) * \mathcal{R}^{t,l} \\ \Delta^{t,l}_{\mathcal{B}} * \left(\mathcal{R}^{t,l}\right)^T * \left(\left(\mathcal{B}^{t,l}\right)^T * \mathcal{B}^{t,l} - \left(\mathcal{A}^{t,l}\right)^T * \mathcal{A}^{t,l}\right) * \mathcal{R}^{t,l}\end{bmatrix}\right\|$$

$$\leq \eta \underbrace{\left\| \left(-\boldsymbol{\mathcal{Z}}^c \boldsymbol{\mathcal{Z}} + \boldsymbol{\mathcal{Z}}_{l,\cdot}^c \boldsymbol{\mathcal{Z}}_{l,\cdot} - m\boldsymbol{\mathcal{P}}_{l,\cdot}\right)\left(\boldsymbol{\Delta}_{\boldsymbol{\mathcal{X}}}^{t,l}\right) \right\| \left\| \boldsymbol{\Delta}^{t,l} \right\|}_{a_1} + \frac{\eta m}{2}$$

$$\cdot \underbrace{\left( \left\| \boldsymbol{\Delta}_{\boldsymbol{\mathcal{A}}}^{t,l} \right\| + \left\| \boldsymbol{\Delta}_{\boldsymbol{\mathcal{B}}}^{t,l} \right\| \right)\left( 2\left\| \boldsymbol{\Delta}_{\boldsymbol{\mathcal{A}}}^{t,l} \right\| \|\boldsymbol{\mathcal{A}}^\star\| + 2\left\| \boldsymbol{\Delta}_{\boldsymbol{\mathcal{B}}}^{t,l} \right\| \|\boldsymbol{\mathcal{B}}^\star\| + \left\| \boldsymbol{\Delta}_{\boldsymbol{\mathcal{A}}}^{t,l} \right\|^2 + \left\| \boldsymbol{\Delta}_{\boldsymbol{\mathcal{B}}}^{t,l} \right\|^2 \right)}_{a_2}$$

$$\leq 2 \times 1250^2 m\eta C^2 \rho^{2t}\sqrt{\sigma_{\max}}\left( \sqrt{\frac{(\mu r)^4 \kappa^7 n_3 \log^2(n \vee n_3)\sigma_{\max}}{nm}} \right)^2$$

$$\tag{153}$$

where the last two inequalities are due to the bounds for $a_1$ and $a_2$ as follows.

For the $a_1$, there is

$$a_1 \leq \left( \left\| \left(\boldsymbol{\mathcal{Z}}^c \boldsymbol{\mathcal{Z}} - m\boldsymbol{\mathcal{I}}_r\right)\left(\boldsymbol{\Delta}_{\boldsymbol{\mathcal{X}}}^{t,l}\right) \right\| + m\left\| \boldsymbol{\mathcal{A}}^{t,l} * (\boldsymbol{\mathcal{B}}^{t,l})^T - \boldsymbol{\mathcal{A}}^\star * (\boldsymbol{\mathcal{B}}^\star)^T \right\| \right.$$
$$\left. + \left\| \left(\boldsymbol{\mathcal{Z}}_{l,\cdot}^c \boldsymbol{\mathcal{Z}}_{l,\cdot} - m\boldsymbol{\mathcal{P}}_{l,\cdot}\right)\left(\boldsymbol{\Delta}_{\boldsymbol{\mathcal{X}}}^{t,l}\right) \right\| \right)\left\| \boldsymbol{\Delta}^{t,l} \right\|$$
$$\overset{(i)}{\leq} \left( 2Cmr\sqrt{\frac{nn_3}{m}}\log(n \vee n_3)\max_{ij}\left\| \left(\boldsymbol{\Delta}_{\boldsymbol{\mathcal{X}}}^{t,l}\right)_{ij:} \right\|_F + 2Cmr\sqrt{\frac{n_3}{m}}\log(n \vee n_3)\max_{j}\left\| \left(\boldsymbol{\Delta}_{\boldsymbol{\mathcal{X}}}^{t,l}\right)_{lj:} \right\|_F \right.$$
$$\left. + m\left\| \boldsymbol{\Delta}^{t,l} \right\|\left( \|\boldsymbol{\mathcal{B}}^\star\| + \|\boldsymbol{\mathcal{A}}^\star\| + \left\| \boldsymbol{\Delta}^{t,l} \right\| \right) \right)\left\| \boldsymbol{\Delta}^{t,l} \right\|$$
$$\overset{(ii)}{\leq} \left( 4Cmr\sqrt{\frac{nn_3}{m}}\log(n \vee n_3) \cdot 1300C\rho^t\sqrt{\frac{(\mu r)^5 \kappa^9 n_3 \log^2(n_3 \vee n)\sigma_{\max}}{n^2 m}} \cdot \sqrt{\frac{\mu r \sigma_{\max}}{n}} \right.$$
$$\left. + 3m \cdot 12\kappa \cdot 52C\rho^t\sqrt{\frac{(\mu r)^4 \kappa^5 n_3 \log^2(n \vee n_3)\sigma_{\max}}{nm}} \cdot \sqrt{\sigma_{\max}} \right)$$
$$\cdot 12\kappa \cdot 52C\rho^t\sqrt{\frac{(\mu r)^4 \kappa^5 n_3 \log^2(n \vee n_3)\sigma_{\max}}{nm}}$$
$$\leq (1248C)^2 m\rho^{2t}\sqrt{\sigma_{\max}}\left( \sqrt{\frac{(\mu r)^4 \kappa^7 n_3 \log^2(n \vee n_3)\sigma_{\max}}{nm}} \right)^2,$$

$$\tag{154}$$

where the (i) is due to concentration Lemma 2 and (ii) are because (64) and (79). The last inequality is due to the tube-wise sample complexity in (12).

For $a_2$, there is

$$a_2 \leq 2\left\| \boldsymbol{\Delta}^{t,l} \right\|\left( 4\left\| \boldsymbol{\Delta}^{t,l} \right\| \|\boldsymbol{\mathcal{A}}^\star\| + \left\| \boldsymbol{\Delta}^{t,l} \right\|^2 \right)$$
$$\overset{(i)}{\leq} 1248C\rho^t\sqrt{\frac{(\mu r)^4 \kappa^5 n_3 \log^2(n \vee n_3)\sigma_{\max}}{nm}}\left( 2496C\rho^t\sqrt{\frac{(\mu r)^4 \kappa^5 n_3 \log^2(n \vee n_3)\sigma_{\max}}{nm}} \cdot \sqrt{\sigma_{\max}} \right.$$
$$\left. + \left( 624C\rho^t\sqrt{\frac{(\mu r)^4 \kappa^5 n_3 \log^2(n \vee n_3)\sigma_{\max}}{nm}} \right)^2 \right)$$
$$\leq 1248 \times 2500C^2 \rho^{2t}\sqrt{\sigma_{\max}}\left( \sqrt{\frac{(\mu r)^4 \kappa^5 n_3 \log^2(n \vee n_3)\sigma_{\max}}{nm}} \right)^2.$$

$$\tag{155}$$

Next, we prove that $\mathcal{C}$ is a symmetric positive definite tensor. The proof idea is similar to that of Section A.2.3. Based on the definition of (147), there is

$$\mathcal{C} = (\mathcal{A}^\star)^T * \tilde{\mathcal{A}}^{t,l} * \mathcal{R}^{t,l} + (\mathcal{B}^\star)^T * \tilde{\mathcal{B}}^{t,l} * \mathcal{R}^{t,l} - \eta (\mathcal{A}^\star)^T * \mathcal{Z}_{l,\cdot}^c \mathcal{Z}_{l,\cdot} \left( \boldsymbol{\Delta}_{\mathcal{X}}^{t,l} \right) * \mathcal{B}^\star$$
$$- \eta (\mathcal{B}^\star)^T * \mathcal{Z}_{l,\cdot}^c \mathcal{Z}_{l,\cdot} \left( \boldsymbol{\Delta}_{\mathcal{X}}^{t,l} \right) * \mathcal{A}^\star - \eta (\mathcal{A}^\star)^T * \mathcal{P}_{l,\cdot} \left( \boldsymbol{\Delta}_{\mathcal{X}}^{t,l} \right) * \mathcal{B}^\star - \eta (\mathcal{B}^\star)^T * \mathcal{P}_{l,\cdot} \left( \boldsymbol{\Delta}_{\mathcal{X}}^{t,l} \right) * \mathcal{A}^\star \tag{156}$$

Based on definition of $\mathcal{R}^{t,l}$ in (21) and Lemma 4, the $(\mathcal{A}^\star)^T * \tilde{\mathcal{A}}^{t,l} * \mathcal{R}^{t,l} + (\mathcal{B}^\star)^T * \tilde{\mathcal{B}}^{t,l} * \mathcal{R}^{t,l}$ is symmetric positive definite. Thus, $\mathcal{C}$ is also symmetric. In addition,

$$\left\| \begin{bmatrix} \mathcal{A}^\star \\ \mathcal{B}^\star \end{bmatrix}^T \begin{bmatrix} \tilde{\mathcal{A}}^{t+1,l} \\ \tilde{\mathcal{B}}^{t+1,l} \end{bmatrix} - \begin{bmatrix} \mathcal{A}^\star \\ \mathcal{B}^\star \end{bmatrix}^T * \begin{bmatrix} \mathcal{A}^\star \\ \mathcal{B}^\star \end{bmatrix} \right\| \leq \left\| \begin{bmatrix} \tilde{\mathcal{A}}^{t+1,l} - \mathcal{A}^\star \\ \tilde{\mathcal{B}}^{t+1,l} - \mathcal{B}^\star \end{bmatrix} \right\| \left\| \begin{bmatrix} \mathcal{A}^\star \\ \mathcal{B}^\star \end{bmatrix} \right\|$$
$$\leq \left( \left\| \begin{bmatrix} \mathbb{E}\left[\tilde{\mathcal{A}}^{t+1,l}\right] - \mathcal{A}^\star \\ \mathbb{E}\left[\tilde{\mathcal{B}}^{t+1,l}\right] - \mathcal{B}^\star \end{bmatrix} \right\| + \left\| \begin{bmatrix} \mathbb{E}\left[\tilde{\mathcal{A}}^{t+1,l}\right] - \tilde{\mathcal{A}}^{t+1,l} \\ \mathbb{E}\left[\tilde{\mathcal{B}}^{t+1,l}\right] - \tilde{\mathcal{B}}^{t+1,l} \end{bmatrix} \right\| \right) \left\| \begin{bmatrix} \mathcal{A}^\star \\ \mathcal{B}^\star \end{bmatrix} \right\|$$
$$\leq 0.5\sigma_{\min}, \tag{157}$$

where the last inequality uses the results in (158) and (160) as follows.

Bounding the $\left\| \begin{bmatrix} \mathbb{E}\left[\tilde{\mathcal{A}}^{t+1,l}\right] - \tilde{\mathcal{A}}^{t+1,l} \\ \mathbb{E}\left[\tilde{\mathcal{B}}^{t+1,l}\right] - \tilde{\mathcal{B}}^{t+1,l} \end{bmatrix} \right\|$ is similar to proof in Section A.2.1 as

$$\left\| \begin{bmatrix} \mathbb{E}\left[\tilde{\mathcal{A}}^{t+1,l}\right] - \tilde{\mathcal{A}}^{t+1,l} \\ \mathbb{E}\left[\tilde{\mathcal{B}}^{t+1,l}\right] - \tilde{\mathcal{B}}^{t+1,l} \end{bmatrix} \right\| \leq \eta \left\| \begin{bmatrix} \mathcal{Z}_{-l,\cdot}^c \mathcal{Z}_{-l,\cdot} \left( \boldsymbol{\Delta}_{\mathcal{X}}^{t,l} \right) * \mathcal{B}^\star - m\mathcal{P}_{-l,\cdot} \left( \boldsymbol{\Delta}_{\mathcal{X}}^{t,l} \right) * \mathcal{B}^\star \\ \left( \mathcal{Z}_{-l,\cdot}^c \mathcal{Z}_{-l,\cdot} \left( \boldsymbol{\Delta}_{\mathcal{X}}^{t,l} \right) \right)^T * \mathcal{A}^\star - m \left( \mathcal{P}_{-l,\cdot} \left( \boldsymbol{\Delta}_{\mathcal{X}}^{t,l} \right) \right)^T * \mathcal{B}^\star \end{bmatrix} \right\|$$
$$\overset{(i)}{\leq} 2\eta \|\mathcal{A}^\star\| \left\| (\mathcal{Z}^c \mathcal{Z} - \mathcal{I}_n) \left( \boldsymbol{\Delta}_{\mathcal{X}}^{t,l} \right) \right\|$$
$$\overset{(ii)}{\leq} 2C\sqrt{\sigma_{\max}} m\eta r \sqrt{\frac{nn_3}{m}} \log(n \vee n_3) \max_{ij} \left\| \left( \boldsymbol{\Delta}_{\mathcal{X}}^{t,l} \right)_{ij:} \right\|_F$$
$$\overset{(iii)}{\leq} 2C\sqrt{\sigma_{\max}} m\eta r \sqrt{\frac{nn_3}{m}} \log(n \vee n_3) \cdot \left( \max_i \left\| (\boldsymbol{\Delta}_{\mathcal{A}}^{t,l})_{i::} \right\| \max_i \|\mathcal{B}_{i::}^\star\| \right.$$
$$\left. + \max_i \left\| (\boldsymbol{\Delta}_{\mathcal{B}}^{t,l})_{i::} \right\| \max_i \|\mathcal{A}_{i::}^\star\| + \max_i \left\| (\boldsymbol{\Delta}_{\mathcal{A}}^{t,l})_{i::} \right\| \max_i \left\| (\boldsymbol{\Delta}_{\mathcal{B}}^{t,l})_{i::} \| \right) \right.$$
$$\overset{(iv)}{\leq} 2C\sqrt{\sigma_{\max}} m\eta r \sqrt{\frac{nn_3}{m}} \log(n \vee n_3) \cdot \sqrt{\frac{\mu r \sigma_{\max}}{n}}$$
$$\cdot 1938\rho^t \sqrt{\frac{(\mu r)^5 \kappa^9 n_3 \log^2(n \vee n_3)\sigma_{\max}}{n^2 m}}$$
$$\leq 0.25 \frac{\sqrt{\sigma_{\max}}}{\kappa}, \tag{158}$$

where the leave-one-out projection operator $\mathcal{P}_{-i,\cdot}$ is defined as

$$[\mathcal{P}_{-i,\cdot}(\mathcal{X})]_{kj:} := \begin{cases} \mathcal{X}_{kj:}, & k \neq i, \forall j \in [n], \\ \mathbf{0}_m, & k = i, \forall j \in [n]. \end{cases} \tag{159}$$

(i) is because the spectral norm of the sub-tensor is not larger than the original tensor. (ii) and (iii) are due to the Lemma 2 and tensor norm inequality (95), respectively. The (iv) is based on the tensor incoherence condition (10), (64), and tube-wise complexity in (12).

For the term $\left\| \begin{bmatrix} \mathbb{E}\left[\tilde{\boldsymbol{\mathcal{A}}}^{t+1,l}\right] - \boldsymbol{\mathcal{A}}^\star \\ \mathbb{E}\left[\tilde{\boldsymbol{\mathcal{B}}}^{t+1,l}\right] - \boldsymbol{\mathcal{B}}^\star \end{bmatrix} \right\|$, we can follow the (45) in Section A.2.2 as

$$
\left\| \begin{bmatrix} \mathbb{E}\left[\tilde{\boldsymbol{\mathcal{A}}}^{t+1,l}\right] - \boldsymbol{\mathcal{A}}^\star \\ \mathbb{E}\left[\tilde{\boldsymbol{\mathcal{B}}}^{t+1,l}\right] - \boldsymbol{\mathcal{B}}^\star \end{bmatrix} \right\|
$$

$$
= \left\| \frac{1}{2}\begin{bmatrix} \boldsymbol{\Delta}_{\boldsymbol{\mathcal{A}}}^{t,l} \\ \boldsymbol{\Delta}_{\boldsymbol{\mathcal{B}}}^{t,l} \end{bmatrix} * \left(\boldsymbol{\mathcal{I}}_r - 2\eta m \left(\boldsymbol{\mathcal{A}}^\star\right)^T * \boldsymbol{\mathcal{A}}^\star \right) + \frac{1}{2}\begin{bmatrix} \left(\boldsymbol{\mathcal{I}}_n - 2\eta m \boldsymbol{\mathcal{A}}^\star \left(\boldsymbol{\mathcal{A}}^\star\right)^T\right) * \boldsymbol{\Delta}_{\boldsymbol{\mathcal{A}}}^{t,l} \\ \left(\boldsymbol{\mathcal{I}}_n - 2\eta m \boldsymbol{\mathcal{B}}^\star \left(\boldsymbol{\mathcal{B}}^\star\right)^T\right) * \boldsymbol{\Delta}_{\boldsymbol{\mathcal{B}}}^{t,l} \end{bmatrix} + \eta m \begin{bmatrix} \boldsymbol{\mathcal{E}}_1^{t,l} \\ \boldsymbol{\mathcal{E}}_2^{t,l} \end{bmatrix} \right\|
$$

$$
\leq \frac{1}{2}\left\| \boldsymbol{\mathcal{I}}_r - 2\eta m \left(\boldsymbol{\mathcal{A}}^\star\right)^T * \boldsymbol{\mathcal{A}}^\star \right\| \left\| \boldsymbol{\Delta}^{t,l} \right\| + \frac{1}{2}\left\| \boldsymbol{\mathcal{I}}_{2n} - 2\eta m \begin{bmatrix} \boldsymbol{\mathcal{A}}^\star * \left(\boldsymbol{\mathcal{A}}^\star\right)^T & \boldsymbol{\mathcal{O}} \\ \boldsymbol{\mathcal{O}} & \boldsymbol{\mathcal{B}}^\star * \left(\boldsymbol{\mathcal{B}}^\star\right)^T \end{bmatrix} \right\| \left\| \boldsymbol{\Delta}^{t,l} \right\|
$$

$$
\leq \left(1 - \eta m \sigma_{\min}\right) \cdot \left\| \boldsymbol{\Delta}^{t,l} \right\| + 4\eta m \left\| \boldsymbol{\Delta}^{t,l} \right\|^2 \left\| \boldsymbol{\mathcal{A}}^\star \right\|
$$

$$
\leq \left(1 - 0.9\eta m \sigma_{\min}\right)\left\| \boldsymbol{\Delta}^{t,l} \right\|
$$

$$
\leq 0.25 \frac{\sqrt{\sigma_{\max}}}{\kappa}, \tag{160}
$$

where the last two inequalities are due to tube-wise sample complexity.

Thus, based on (157) and Wely's inequality, we can also conclude that $\sigma_{\min}\left(\boldsymbol{\mathcal{C}}\right) \geq 1.5\sigma_{\min}$, which means that is positive definite tensor. Moreover, based on (153), we can bound the spectral norm of $\boldsymbol{\mathcal{E}}$ as

$$
\|\boldsymbol{\mathcal{E}}\| \leq 2\sqrt{\sigma_{\max}} \cdot 2 \times 1250^2 m\eta C^2 \rho^{2t} \sqrt{\sigma_{\max}} \left( \sqrt{\frac{(\mu r)^4 \kappa^7 n_3 \log^2(n \vee n_3)\sigma_{\max}}{nm}} \right)^2
$$

$$
\leq 0.5\sigma_{\min}, \tag{161}
$$

where the last inequality is due to the tube-wise sample complexity and step size in Theorem 2. Thus, $\boldsymbol{\mathcal{C}}, \boldsymbol{\mathcal{E}}$ satisfy the condition in Lemma 5, applying Lemma 5 as (152) would have

$$
\left\| \left(\boldsymbol{\mathcal{R}}^{t,l}\right)^{-1} * \boldsymbol{\mathcal{R}}^{t+1,l} - \boldsymbol{\mathcal{I}}_r \right\| \leq \frac{2 \times 1250^2 m\eta C^2 \rho^{2t} \sqrt{\sigma_{\max}} \left( \sqrt{\frac{(\mu r)^4 \kappa^7 n_3 \log^2(n \vee n_3)\sigma_{\max}}{nm}} \right)^2 \cdot \sqrt{\sigma_{\max}}}{1.5\sigma_{\min} - 0.5\sigma_{\min}}
$$

$$
= 2 \times 1250^2 m\eta C^2 \rho^{2t} \left( \sqrt{\frac{(\mu r)^4 \kappa^8 n_3 \log^2(n \vee n_3)\sigma_{\max}}{nm}} \right)^2. \tag{162}
$$

$\square$

# B  Proof for Spectral Initialization

To use the same symmetric dilation technique for the matrix [1, 8, 40] in our setting, we construct the auxiliary tensor

$$
\hat{\boldsymbol{\mathcal{X}}}^\star := \begin{bmatrix} \mathbf{0}_{n \times n \times n_3} & \boldsymbol{\mathcal{X}}^\star \\ \left(\boldsymbol{\mathcal{X}}^\star\right)^T & \mathbf{0}_{n \times n \times n_3} \end{bmatrix} \in \mathbb{R}^{2n \times 2n \times n_3}, \tag{163}
$$

which is the symmetric tensor. This is because

$$
\hat{\boldsymbol{X}}^{\star(1)} = \begin{bmatrix} \mathbf{0}_{n \times n} & \boldsymbol{X}^{\star(1)} \\ \left(\boldsymbol{X}^{\star(1)}\right)^T & \mathbf{0}_{n \times n} \end{bmatrix} \tag{164}
$$

is symmetric. $\forall k = 2, \cdots, n_3$, there is

$$\hat{\boldsymbol{X}}^{\star (k)} = \begin{bmatrix} \boldsymbol{0}_{n \times n} & \boldsymbol{X}^{\star (k)} \\ \left( \boldsymbol{X}^{\star (n_3+2-k)} \right)^T & \boldsymbol{0}_{n \times n} \end{bmatrix} \quad \text{and} \quad \hat{\boldsymbol{X}}^{\star (n_3+2-k)} = \begin{bmatrix} \boldsymbol{0}_{n \times n} & \boldsymbol{X}^{\star (n_3+2-k)} \\ \left( \boldsymbol{X}^{\star (k)} \right)^T & \boldsymbol{0}_{n \times n} \end{bmatrix}.$$
(165)

Thus, there is

$$\left( \hat{\boldsymbol{X}}^{\star (k)} \right)^T = \hat{\boldsymbol{X}}^{\star (n_3+2-k)},$$
(166)

which obeys the definition of the symmetric tensor in Definition 3. Thus, each frontal slice of $\hat{\boldsymbol{\mathcal{X}}}^\star$ is the symmetric matrix and has tensor eigenvalue decomposition as

$$\hat{\boldsymbol{\mathcal{X}}}^\star = \frac{1}{\sqrt{2}} \begin{bmatrix} \boldsymbol{\mathcal{U}}^\star & \boldsymbol{\mathcal{U}}^\star \\ \boldsymbol{\mathcal{V}}^\star & -\boldsymbol{\mathcal{V}}^\star \end{bmatrix} * \begin{bmatrix} \boldsymbol{\mathcal{S}}^\star & \boldsymbol{0}_{n \times n \times n_3} \\ \boldsymbol{0}_{n \times n \times n_3} & -\boldsymbol{\mathcal{S}}^\star \end{bmatrix} * \frac{1}{\sqrt{2}} \begin{bmatrix} \boldsymbol{\mathcal{U}}^\star & \boldsymbol{\mathcal{U}}^\star \\ \boldsymbol{\mathcal{V}}^\star & -\boldsymbol{\mathcal{V}}^\star \end{bmatrix}^T.$$
(167)

Similarly, we can also define

$$\hat{\boldsymbol{\mathcal{X}}}^0 := \begin{bmatrix} \boldsymbol{0}_{n \times n \times n_3} & \boldsymbol{\mathcal{X}}^0 \\ \left( \boldsymbol{\mathcal{X}}^0 \right)^T & \boldsymbol{0}_{n \times n \times n_3} \end{bmatrix} \quad \text{and} \quad \hat{\boldsymbol{\mathcal{X}}}^{0,l} := \begin{bmatrix} \boldsymbol{0}_{n \times n \times n_3} & \boldsymbol{\mathcal{X}}^{0,l} \\ \left( \boldsymbol{\mathcal{X}}^{0,l} \right)^T & \boldsymbol{0}_{n \times n \times n_3} \end{bmatrix}.$$
(168)

Then we have

$$\left\| \hat{\boldsymbol{\mathcal{X}}}^{0,l} - \hat{\boldsymbol{\mathcal{X}}}^\star \right\| = \begin{bmatrix} \boldsymbol{0}_{n \times n \times n_3} & \frac{1}{m} \boldsymbol{\mathcal{Z}}_{-l,\cdot}^c \boldsymbol{\mathcal{Z}}_{-l,\cdot} \left( \boldsymbol{\mathcal{X}}^\star \right) + \boldsymbol{\mathcal{P}}_{l,\cdot} (\boldsymbol{\mathcal{X}}^\star) - \boldsymbol{\mathcal{X}}^\star \\ \left( \frac{1}{m} \boldsymbol{\mathcal{Z}}_{-l,\cdot}^c \boldsymbol{\mathcal{Z}}_{-l,\cdot} \left( \boldsymbol{\mathcal{X}}^\star \right) + \boldsymbol{\mathcal{P}}_{l,\cdot} (\boldsymbol{\mathcal{X}}^\star) - \boldsymbol{\mathcal{X}}^\star \right)^T & \boldsymbol{0}_{n \times n \times n_3} \end{bmatrix},$$
(169)

which is actually the sub-part of the $\hat{\boldsymbol{\mathcal{X}}}^0 - \hat{\boldsymbol{\mathcal{X}}}^\star$. In addition, based on the definition of $\boldsymbol{\mathcal{X}}^0$, there is

$$\begin{aligned}
\left\| \hat{\boldsymbol{\mathcal{X}}}^0 - \hat{\boldsymbol{\mathcal{X}}}^\star \right\| &= \left\| \begin{bmatrix} \boldsymbol{0}_{n \times n \times n_3} & \frac{1}{m} \boldsymbol{\mathcal{Z}}^c \boldsymbol{\mathcal{Z}} (\boldsymbol{\mathcal{X}}^\star) - \boldsymbol{\mathcal{X}}^\star \\ \left( \frac{1}{m} \boldsymbol{\mathcal{Z}}^c \boldsymbol{\mathcal{Z}} (\boldsymbol{\mathcal{X}}^\star) - \boldsymbol{\mathcal{X}}^\star \right)^T & \boldsymbol{0}_{n \times n \times n_3} \end{bmatrix} \right\| \\
&= \left\| \frac{1}{m} \boldsymbol{\mathcal{Z}}^c \boldsymbol{\mathcal{Z}} (\boldsymbol{\mathcal{X}}^\star) - \boldsymbol{\mathcal{X}}^\star \right\| \\
&\leq 2 C r \sqrt{\frac{n n_3}{m}} \log(n \vee n_3) \max_{ij} \left\| \boldsymbol{\mathcal{X}}_{ij:}^\star \right\|_F \\
&\overset{(i)}{\leq} 2 C r \sqrt{\frac{n n_3}{m}} \log(n \vee n_3) \left( \sqrt{\frac{\mu r \sigma_{\max}}{n}} \right)^2 \\
&\leq 2 C \sqrt{\frac{(\mu r)^4 n_3 \log^2 (n \vee n_3) \kappa^2}{mn}} \sigma_{\min} \\
&\leq \frac{1}{40 \kappa^2} \sigma_{\min} \\
&\leq \frac{\sigma_{\min}}{4},
\end{aligned}$$
(170)

where (i) is due to the tensor norm inequality (95) and the tensor incoherence condition in (10). The last inequality is due to the tube-wise sample complexity in (25). Thus, we also have

$$\left\| \hat{\boldsymbol{\mathcal{X}}}^{0,l} - \hat{\boldsymbol{\mathcal{X}}}^\star \right\| \leq \left\| \hat{\boldsymbol{\mathcal{X}}}^0 - \hat{\boldsymbol{\mathcal{X}}}^\star \right\| \leq \frac{\sigma_{\min}}{4}.$$
(171)

Because $\hat{\boldsymbol{\mathcal{X}}}^{0,l}, \hat{\boldsymbol{\mathcal{X}}}^0$ are symmetric tensor, if the top-$r$ skinny of t-SVDs are as

$$\begin{aligned}
\boldsymbol{\mathcal{U}}^0 * \boldsymbol{\mathcal{S}}^0 * \left( \boldsymbol{\mathcal{V}}^0 \right)^T &= \boldsymbol{\mathcal{X}}^0 \\
\boldsymbol{\mathcal{U}}^{0,l} * \boldsymbol{\mathcal{S}}^{0,l} * \left( \boldsymbol{\mathcal{V}}^{0,l} \right)^T &= \boldsymbol{\mathcal{X}}^{0,l}.
\end{aligned}$$
(172)

Then, the top-$r$ T-eigenvalue decompositions [72] for $\hat{\boldsymbol{\mathcal{X}}}^0, \hat{\boldsymbol{\mathcal{X}}}^{0,l}$ are

$$\frac{1}{\sqrt{2}}\begin{bmatrix}\boldsymbol{\mathcal{U}}^0\\\boldsymbol{\mathcal{V}}^0\end{bmatrix} * \boldsymbol{\mathcal{S}}^0 * \frac{1}{\sqrt{2}}\begin{bmatrix}\boldsymbol{\mathcal{U}}^0\\\boldsymbol{\mathcal{V}}^0\end{bmatrix}^T \quad \text{and} \quad \frac{1}{\sqrt{2}}\begin{bmatrix}\boldsymbol{\mathcal{U}}^{0,l}\\\boldsymbol{\mathcal{V}}^{0,l}\end{bmatrix} * \boldsymbol{\mathcal{S}}^{0,l} * \frac{1}{\sqrt{2}}\begin{bmatrix}\boldsymbol{\mathcal{U}}^{0,l}\\\boldsymbol{\mathcal{V}}^{0,l}\end{bmatrix}^T , \tag{173}$$

respectively. Eq.(170) means that in the block diagonal formula, there is

$$\left\|\overline{\hat{\boldsymbol{X}}^{0,l}} - \overline{\hat{\boldsymbol{X}}^\star}\right\| \le \frac{\sigma_{\min}}{4} \quad \text{and} \quad \left\|\overline{\hat{\boldsymbol{X}}^0} - \overline{\hat{\boldsymbol{X}}^\star}\right\| \le \frac{\sigma_{\min}}{4}. \tag{174}$$

Based on Wely's inequality [61] and the above inequalities, there are

$$\frac{3}{4}\sigma_{\min} \le \sigma_{rn_3}(\overline{\boldsymbol{S}^0}) \le \sigma_{\max}(\overline{\boldsymbol{S}^0}) \le \frac{5}{4}\sigma_{\max} \tag{175}$$

$$\frac{3}{4}\sigma_{\min} \le \sigma_{rn_3}(\overline{\boldsymbol{S}^{0,l}}) \le \sigma_{\max}(\overline{\boldsymbol{S}^{0,l}}) \le \frac{5}{4}\sigma_{\max}. \tag{176}$$

Recall the definition of $\boldsymbol{\mathcal{A}}^0 = \boldsymbol{\mathcal{U}}^0 * \left(\boldsymbol{\mathcal{S}}^0\right)^{\frac{1}{2}}, \boldsymbol{\mathcal{B}}^0 = \boldsymbol{\mathcal{V}}^0 * \left(\boldsymbol{\mathcal{S}}^0\right)^{\frac{1}{2}}, \boldsymbol{\mathcal{A}}^{0,l} = \boldsymbol{\mathcal{U}}^{0,l} * \left(\boldsymbol{\mathcal{S}}^{0,l}\right)^{\frac{1}{2}}, \boldsymbol{\mathcal{B}}^0 = \boldsymbol{\mathcal{V}}^{0,l} * \left(\boldsymbol{\mathcal{S}}^{0,l}\right)^{\frac{1}{2}}$. Define the tensor corresponding eigen-spaces

$$\tilde{\boldsymbol{\mathcal{G}}} := \frac{1}{\sqrt{2}}\begin{bmatrix}\boldsymbol{\mathcal{U}}^\star\\\boldsymbol{\mathcal{V}}^\star\end{bmatrix}, \quad \boldsymbol{\mathcal{G}} := \frac{1}{\sqrt{2}}\begin{bmatrix}\boldsymbol{\mathcal{A}}^\star\\\boldsymbol{\mathcal{B}}^\star\end{bmatrix} \tag{177}$$

and

$$\tilde{\boldsymbol{\mathcal{H}}}^0 := \frac{1}{\sqrt{2}}\begin{bmatrix}\boldsymbol{\mathcal{U}}^0\\\boldsymbol{\mathcal{V}}^0\end{bmatrix}, \quad \boldsymbol{\mathcal{H}}^0 := \frac{1}{\sqrt{2}}\begin{bmatrix}\boldsymbol{\mathcal{A}}^0\\\boldsymbol{\mathcal{B}}^0\end{bmatrix}, \quad \tilde{\boldsymbol{\mathcal{H}}}^{0,l} := \frac{1}{\sqrt{2}}\begin{bmatrix}\boldsymbol{\mathcal{U}}^{0,l}\\\boldsymbol{\mathcal{V}}^{0,l}\end{bmatrix}, \quad \boldsymbol{\mathcal{H}}^{0,l} := \frac{1}{\sqrt{2}}\begin{bmatrix}\boldsymbol{\mathcal{A}}^{0,l}\\\boldsymbol{\mathcal{B}}^{0,l}\end{bmatrix}. \tag{178}$$

Further, the alignment tensors between these eigen-spaces are defined as

$$\boldsymbol{\mathcal{Q}}^0 := \arg\min_{\boldsymbol{\mathcal{R}}\in\mathcal{O}_r}\left\|\tilde{\boldsymbol{\mathcal{H}}}^0 * \boldsymbol{\mathcal{R}} - \tilde{\boldsymbol{\mathcal{G}}}\right\|_F \tag{179}$$

$$\boldsymbol{\mathcal{Q}}^{0,l} := \arg\min_{\boldsymbol{\mathcal{R}}\in\mathcal{O}_r}\left\|\tilde{\boldsymbol{\mathcal{H}}}^{0,l} * \boldsymbol{\mathcal{R}} - \tilde{\boldsymbol{\mathcal{G}}}\right\|_F . \tag{180}$$

## B.1 Proof of (26)

$$\begin{aligned}
\left\|\begin{bmatrix}\boldsymbol{\mathcal{A}}^0\\\boldsymbol{\mathcal{B}}^0\end{bmatrix} * \boldsymbol{\mathcal{R}}^0 - \begin{bmatrix}\boldsymbol{\mathcal{A}}^\star\\\boldsymbol{\mathcal{B}}^\star\end{bmatrix}\right\| &= \sqrt{2}\left\|\boldsymbol{\mathcal{H}}^0 * \boldsymbol{\mathcal{R}}^0 - \boldsymbol{\mathcal{G}}\right\| \\
&= \sqrt{2}\left\|\tilde{\boldsymbol{\mathcal{H}}}^0 * (\boldsymbol{\mathcal{S}}^0)^{\frac{1}{2}}\left(\boldsymbol{\mathcal{R}}^0 - \boldsymbol{\mathcal{Q}}^0\right) + \tilde{\boldsymbol{\mathcal{H}}}^0\left((\boldsymbol{\mathcal{S}}^0)^{\frac{1}{2}} * \boldsymbol{\mathcal{Q}}^0 - \boldsymbol{\mathcal{Q}}^0 * (\boldsymbol{\mathcal{S}}^\star)^{\frac{1}{2}}\right)\right. \\
&\quad \left.+ \left(\tilde{\boldsymbol{\mathcal{H}}}^0 * \boldsymbol{\mathcal{Q}}^0 - \tilde{\boldsymbol{\mathcal{G}}}\right) * (\boldsymbol{\mathcal{S}}^\star)^{\frac{1}{2}}\right\| \\
&\le \sqrt{2}\left\|\boldsymbol{\mathcal{S}}^0\right\|^{\frac{1}{2}}\left\|\boldsymbol{\mathcal{R}}^0 - \boldsymbol{\mathcal{Q}}^0\right\| + \sqrt{2}\left\|(\boldsymbol{\mathcal{S}}^0)^{\frac{1}{2}} * \boldsymbol{\mathcal{Q}}^0 - \boldsymbol{\mathcal{Q}}^0 * (\boldsymbol{\mathcal{S}}^\star)^{\frac{1}{2}}\right\| \\
&\quad + \sqrt{2}\|\boldsymbol{\mathcal{S}}^\star\|^{\frac{1}{2}}\left\|\tilde{\boldsymbol{\mathcal{G}}}^0 * \boldsymbol{\mathcal{Q}}^0 - \tilde{\boldsymbol{\mathcal{G}}}\right\| \\
&\overset{(i)}{\le} 15\sqrt{2}\frac{\sigma_{\max}^{\frac{1}{2}}\kappa^{\frac{3}{2}}}{\sigma_{\min}}\left\|\hat{\boldsymbol{\mathcal{X}}}^0 - \hat{\boldsymbol{\mathcal{X}}}^\star\right\| + 15\sqrt{2}\frac{\kappa}{\sigma_{\min}^{\frac{1}{2}}}\left\|\hat{\boldsymbol{\mathcal{X}}}^0 - \hat{\boldsymbol{\mathcal{X}}}^\star\right\| \\
&\quad + \frac{3\sqrt{2}\sigma_{\max}^{\frac{1}{2}}}{\sigma_{\min}}\left\|\hat{\boldsymbol{\mathcal{X}}}^0 - \hat{\boldsymbol{\mathcal{X}}}^\star\right\| \\
&\overset{(ii)}{\le} 2C\sqrt{\frac{(\mu r)^4 n_3 \log^2(n\vee n_3)\kappa\sigma_{\max}}{mn}}\sqrt{\sigma_{\min}} \\
&\quad \cdot \left(22\frac{\sigma_{\max}^{\frac{1}{2}}\kappa^{\frac{3}{2}}}{\sigma_{\min}} + 22\frac{\kappa}{\sigma_{\min}^{\frac{1}{2}}} + 5\frac{\sigma_{\max}^{\frac{1}{2}}}{\sigma_{\min}}\right) \\
&\le 98C\sqrt{\frac{(\mu r)^4\kappa^5 n_3\log^2(n\vee n_3)\sigma_{\max}}{mn}}, \tag{181}
\end{aligned}$$

where we use Lemma 10 and 12 in (i) and (170) in (ii).

## B.2 Proof of (28)

$$
\left\| \left( \begin{bmatrix} \boldsymbol{\mathcal{A}}^{0,l} \\ \boldsymbol{\mathcal{B}}^{0,l} \end{bmatrix} * \boldsymbol{\mathcal{R}}^{0,l} - \begin{bmatrix} \boldsymbol{\mathcal{A}}^\star \\ \boldsymbol{\mathcal{B}}^\star \end{bmatrix} \right)_{l::} \right\| = \sqrt{2} \left\| \left( \boldsymbol{\mathcal{H}}^{0,l} * \boldsymbol{\mathcal{R}}^{0,l} - \boldsymbol{\mathcal{G}} \right)_{l::} \right\|
$$

$$
\leq \sqrt{2} \left\| \left( \boldsymbol{\mathcal{H}}^{0,l} * \boldsymbol{\mathcal{Q}}^{0,l} - \boldsymbol{\mathcal{G}} \right)_{l::} \right\| + \sqrt{2} \left\| \boldsymbol{\mathcal{H}}^{0,l}_{l::} * (\boldsymbol{\mathcal{R}}^{0,l} - \boldsymbol{\mathcal{Q}}^{0,l}) \right\|
$$

$$
= \sqrt{2} \left\| \left( \hat{\boldsymbol{\mathcal{X}}}^{0,l} * \tilde{\boldsymbol{\mathcal{H}}}^{0,l} * (\boldsymbol{\mathcal{S}}^{0,l})^{-\frac{1}{2}} * \boldsymbol{\mathcal{Q}}^{0,l} - \hat{\boldsymbol{\mathcal{X}}}^\star * \tilde{\boldsymbol{\mathcal{G}}} * (\boldsymbol{\mathcal{S}}^\star)^{-\frac{1}{2}} \right)_{l::} \right\|
$$

$$
+ \sqrt{2} \left\| \boldsymbol{\mathcal{H}}^{0,l}_{l::} \right\| \left\| \boldsymbol{\mathcal{R}}^{0,l} - \boldsymbol{\mathcal{Q}}^{0,l} \right\|
$$

$$
= \sqrt{2} \left\| \hat{\boldsymbol{\mathcal{X}}}^\star_{l::} * \left( \tilde{\boldsymbol{\mathcal{H}}}^{0,l} * \left( \left( \boldsymbol{\mathcal{S}}^{0,l} \right)^{-\frac{1}{2}} * \boldsymbol{\mathcal{Q}}^{0,l} - \boldsymbol{\mathcal{Q}}^{0,l} * (\boldsymbol{\mathcal{S}}^\star)^{-\frac{1}{2}} \right) \right. \right.
$$

$$
\left. \left. + \left( \tilde{\boldsymbol{\mathcal{H}}}^{0,l} * \boldsymbol{\mathcal{Q}}^{0,l} - \tilde{\boldsymbol{\mathcal{G}}} \right) * (\boldsymbol{\mathcal{S}}^\star)^{-\frac{1}{2}} \right) \right\| + \sqrt{2} \left\| \boldsymbol{\mathcal{H}}^{0,l}_{l::} \right\| \left\| \boldsymbol{\mathcal{R}}^{0,l} - \boldsymbol{\mathcal{Q}}^{0,l} \right\|
$$

$$
\leq \sqrt{2} \left\| \hat{\boldsymbol{\mathcal{X}}}^\star_{l::} \right\| \left( \left\| \left( \boldsymbol{\mathcal{S}}^{0,l} \right)^{-\frac{1}{2}} * \left( \left( \boldsymbol{\mathcal{S}}^{0,l} \right)^{\frac{1}{2}} * \boldsymbol{\mathcal{Q}}^{0,l} - \boldsymbol{\mathcal{Q}}^{0,l} * (\boldsymbol{\mathcal{S}}^\star)^{\frac{1}{2}} \right) * (\boldsymbol{\mathcal{S}}^\star)^{-\frac{1}{2}} \right\| \right.
$$

$$
\left. + \frac{\left\| \tilde{\boldsymbol{\mathcal{H}}}^{0,l} * \boldsymbol{\mathcal{Q}}^{0,l} - \tilde{\boldsymbol{\mathcal{G}}} \right\|}{\sqrt{\sigma_{\min}}} \right) + \sqrt{2} \left( \max_l \| \boldsymbol{\mathcal{G}}_{l::} \| + \left\| \left( \boldsymbol{\mathcal{H}}^{0,l} * \boldsymbol{\mathcal{Q}}^{0,l} - \boldsymbol{\mathcal{G}} \right)_{l::} \right\| \right) \left\| \boldsymbol{\mathcal{R}}^{0,l} - \boldsymbol{\mathcal{Q}}^{0,l} \right\|
$$

$$
\overset{(i)}{\leq} \sqrt{2} \max_i \| \boldsymbol{\mathcal{A}}^\star_{i::} \| \cdot \sqrt{\sigma_{\max}} \left( \cdot \sqrt{\frac{4}{3\sigma_{\min}}} \cdot 15 \frac{\sigma_{\max}}{\sigma_{\min}^{\frac{3}{2}}} \left\| \hat{\boldsymbol{\mathcal{X}}}^{0,l} - \hat{\boldsymbol{\mathcal{X}}}^\star \right\| \cdot \frac{1}{\sqrt{\sigma_{\min}}} \right.
$$

$$
\left. + \frac{3}{\sigma_{\min}^{\frac{3}{2}}} \left\| \hat{\boldsymbol{\mathcal{X}}}^{0,l} - \hat{\boldsymbol{\mathcal{X}}}^\star \right\| \right) + \sqrt{2} \left( \sqrt{\frac{\mu r \sigma_{\max}}{2n}} + \left\| \left( \boldsymbol{\mathcal{H}}^{0,l} * \boldsymbol{\mathcal{Q}}^{0,l} - \boldsymbol{\mathcal{G}} \right)_{l::} \right\| \right)
$$

$$
\cdot 15 \frac{\sigma_{\max}^{\frac{3}{2}}}{\sigma_{\min}^{\frac{5}{2}}} \left\| \hat{\boldsymbol{\mathcal{X}}}^{0,l} - \hat{\boldsymbol{\mathcal{X}}}^\star \right\|
$$

$$
\overset{(ii)}{\leq} \frac{33\kappa\sigma_{\max}}{\sigma_{\min}^{\frac{3}{2}}} \cdot 2C \sqrt{\frac{(\mu r)^4 n_3 \log^2(n \vee n_3)\kappa^2}{mn^2}} \sigma_{\min} + \sqrt{2} \left( \sqrt{\frac{\mu r \sigma_{\max}}{2n}} \right.
$$

$$
\left. + \left\| \left( \boldsymbol{\mathcal{H}}^{0,l} * \boldsymbol{\mathcal{Q}}^{0,l} - \boldsymbol{\mathcal{G}} \right)_{l::} \right\| \right) \cdot 15 \frac{\sigma_{\max}^{\frac{3}{2}}}{\sigma_{\min}^{\frac{5}{2}}} \cdot 2C \sqrt{\frac{(\mu r)^4 n_3 \log^2(n \vee n_3)\kappa^2}{mn}} \sigma_{\min}
$$

$$
\leq 66C \sqrt{\frac{(\mu r)^4 n_3 \log^2(n \vee n_3)\kappa^5 \sigma_{\max}}{mn^2}} + \sqrt{2} \left( 66C \sqrt{\frac{(\mu r)^4 n_3 \log^2(n \vee n_3)\kappa^5 \sigma_{\max}}{mn^2}} \right.
$$

$$
\left. + \sqrt{\frac{\mu r \sigma_{\max}}{2n}} \right) \cdot 15 \frac{\sigma_{\max}^{\frac{3}{2}}}{\sigma_{\min}^{\frac{5}{2}}} \cdot 2C \sqrt{\frac{(\mu r)^4 n_3 \log^2(n \vee n_3)\kappa^2}{mn}} \sigma_{\min}
$$

$$
\overset{(iii)}{\leq} 66C \sqrt{\frac{(\mu r)^4 n_3 \log^2(n \vee n_3)\kappa^5 \sigma_{\max}}{mn^2}} + 60C \sqrt{\frac{(\mu r)^5 n_3 \log^2(n \vee n_3)\kappa^5 \sigma_{\max}}{mn^2}}
$$

$$
\leq 126C \sqrt{\frac{(\mu r)^5 n_3 \log^2(n \vee n_3)\kappa^5 \sigma_{\max}}{mn^2}}, \tag{182}
$$

where (i) uses the Lemma 12, (ii) is due to (171), and (iii) is due to the sample complexity bound in (25).

## B.3 Proof of (29)

Without loss of generality, we consider the case $l \in [n]$, the denote the alignment tensor

$$\boldsymbol{\mathcal{J}}^{0,l} := \arg\min_{\boldsymbol{\mathcal{R}} \in \mathcal{O}_r} \left\| \tilde{\boldsymbol{\mathcal{H}}}^{0,l} * \boldsymbol{\mathcal{R}} - \tilde{\boldsymbol{\mathcal{H}}}^0 \right\|_F. \tag{183}$$

Then based on definition of $\boldsymbol{\mathcal{T}}^{0,l}$, there is

$$
\begin{aligned}
\left\| \begin{bmatrix} \boldsymbol{\mathcal{A}}^{0,l} \\ \boldsymbol{\mathcal{B}}^{0,l} \end{bmatrix} * \boldsymbol{\mathcal{T}}^{0,l} - \begin{bmatrix} \boldsymbol{\mathcal{A}}^0 \\ \boldsymbol{\mathcal{B}}^0 \end{bmatrix} * \boldsymbol{\mathcal{R}}^0 \right\| &= \sqrt{2} \left\| \boldsymbol{\mathcal{H}}^{0,l} * \boldsymbol{\mathcal{T}}^{0,l} - \boldsymbol{\mathcal{H}}^0 * \boldsymbol{\mathcal{R}}^0 \right\| \\
&\leq \sqrt{2} \left\| \boldsymbol{\mathcal{H}}^{0,l} * \boldsymbol{\mathcal{J}}^{0,l} - \boldsymbol{\mathcal{H}}^0 * \boldsymbol{\mathcal{R}}^0 \right\| \\
&\leq \sqrt{2} \left\| \tilde{\boldsymbol{\mathcal{H}}}^{0,l} \left( \left(\boldsymbol{\mathcal{S}}^{0,l}\right)^{\frac{1}{2}} * \boldsymbol{\mathcal{J}}^{0,l} - \boldsymbol{\mathcal{J}}^{0,l} * \left(\boldsymbol{\mathcal{S}}^0\right)^{\frac{1}{2}} \right) \right\| \\
&\quad + \sqrt{2} \left\| \left( \tilde{\boldsymbol{\mathcal{H}}}^{0,l} * \boldsymbol{\mathcal{J}}^{0,l} - \tilde{\boldsymbol{\mathcal{H}}}^0 \right) * \left(\boldsymbol{\mathcal{S}}^0\right)^{\frac{1}{2}} \right\| \\
&\overset{(i)}{\leq} 15\sqrt{2} \frac{\sigma_{\max}}{\sigma_{\min}^{\frac{3}{2}}} \left\| \hat{\boldsymbol{\mathcal{X}}}^{0,l} - \hat{\boldsymbol{\mathcal{X}}}^0 \right\| + \sqrt{2} \cdot \sqrt{\frac{5\sigma_{\max}}{4}} \cdot \left\| \tilde{\boldsymbol{\mathcal{H}}}^{0,l} * \boldsymbol{\mathcal{J}}^{0,l} - \tilde{\boldsymbol{\mathcal{H}}}^0 \right\| \\
&\overset{(ii)}{\leq} 15\sqrt{2} \frac{\sigma_{\max}}{\sigma_{\min}^{\frac{3}{2}}} \left\| \hat{\boldsymbol{\mathcal{X}}}^{0,l} - \hat{\boldsymbol{\mathcal{X}}}^0 \right\| + \sqrt{\frac{5\sigma_{\max}}{2}} \cdot \frac{2\sqrt{2}}{\sigma_{\min}} \left\| \left( \hat{\boldsymbol{\mathcal{X}}}^{0,l} - \hat{\boldsymbol{\mathcal{X}}}^0 \right) * \tilde{\boldsymbol{\mathcal{H}}}^{0,l} \right\| \\
&\leq 26 \frac{\kappa}{\sqrt{\sigma_{\min}}} \left\| \hat{\boldsymbol{\mathcal{X}}}^{0,l} - \hat{\boldsymbol{\mathcal{X}}}^0 \right\| \\
&\overset{(iii)}{=} 26 \frac{\kappa}{\sqrt{\sigma_{\min}}} \cdot \left\| \boldsymbol{\mathcal{P}}_{l,\cdot}(\boldsymbol{\mathcal{X}}^\star) - \frac{1}{m} \boldsymbol{\mathcal{Z}}_{l,\cdot}^c \boldsymbol{\mathcal{Z}}_{l,\cdot}(\boldsymbol{\mathcal{X}}^\star) \right\| \\
&\overset{(iv)}{\leq} 26 \frac{\kappa}{\sqrt{\sigma_{\min}}} \cdot 2C \sqrt{\frac{n_3}{m}} r \log(n \vee n_3) \max_{ij} \left\| \boldsymbol{\mathcal{X}}_{ij:}^\star \right\|_F \\
&\overset{(v)}{\leq} 52C \sqrt{\frac{n_3 r^2 \log^2(n \vee n_3) \kappa}{m \sigma_{\min}}} \cdot \left( \sqrt{\frac{\mu r \sigma_{\max}}{n}} \right)^2 \\
&\leq 52C \sqrt{\frac{(\mu r)^4 \kappa^2 n_3 \log^2(n \vee n_3) \sigma_{\max}}{n^2 m}}, \tag{184}
\end{aligned}
$$

where (i) is due to Lemma 12 and (175). (ii) is based on matrix version of Davis-Kahan SinΘ theorem [9] and (iii) is based on definition of $\hat{\boldsymbol{\mathcal{X}}}^{0,l}$, $\hat{\boldsymbol{\mathcal{X}}}^0$ in (168). (iv) and (v) are due to our Lemma 2 and tensor incoherence condition in (10), respectively.

## B.4 Proof of (27)

*Proof.* Its proof is the same as proving (14) that $\forall l \in [2n]$, there is

$$
\begin{aligned}
\left\| \left( \begin{bmatrix} \boldsymbol{\mathcal{A}}^0 \\ \boldsymbol{\mathcal{B}}^0 \end{bmatrix} * \boldsymbol{\mathcal{R}}^0 - \begin{bmatrix} \boldsymbol{\mathcal{A}}^\star \\ \boldsymbol{\mathcal{B}}^\star \end{bmatrix} \right)_{l::} \right\| &\leq \left\| \begin{bmatrix} \boldsymbol{\mathcal{A}}^0 \\ \boldsymbol{\mathcal{B}}^0 \end{bmatrix} * \boldsymbol{\mathcal{R}}^0 - \begin{bmatrix} \boldsymbol{\mathcal{A}}^{0,l} \\ \boldsymbol{\mathcal{B}}^{0,l} \end{bmatrix} * \boldsymbol{\mathcal{R}}^{0,l} \right\| + \left\| \left( \begin{bmatrix} \boldsymbol{\mathcal{A}}^{0,l} \\ \boldsymbol{\mathcal{B}}^{0,l} \end{bmatrix} * \boldsymbol{\mathcal{R}}^{0,l} - \begin{bmatrix} \boldsymbol{\mathcal{A}}^\star \\ \boldsymbol{\mathcal{B}}^\star \end{bmatrix} \right)_{l::} \right\| \\
&\overset{(i)}{\leq} 9\kappa \left\| \begin{bmatrix} \boldsymbol{\mathcal{A}}^0 \\ \boldsymbol{\mathcal{B}}^0 \end{bmatrix} * \boldsymbol{\mathcal{R}}^0 - \begin{bmatrix} \boldsymbol{\mathcal{A}}^{0,l} \\ \boldsymbol{\mathcal{B}}^{0,l} \end{bmatrix} * \boldsymbol{\mathcal{T}}^{0,l} \right\| + \left\| \left( \begin{bmatrix} \boldsymbol{\mathcal{A}}^{0,l} \\ \boldsymbol{\mathcal{B}}^{0,l} \end{bmatrix} * \boldsymbol{\mathcal{R}}^{0,l} - \begin{bmatrix} \boldsymbol{\mathcal{A}}^\star \\ \boldsymbol{\mathcal{B}}^\star \end{bmatrix} \right)_{l::} \right\| \\
&\overset{(ii)}{\leq} 9\kappa \cdot 52C \sqrt{\frac{(\mu r)^4 \kappa^2 n_3 \log^2(n \vee n_3) \sigma_{\max}}{n^2 m}} \\
&\quad + 126C \sqrt{\frac{(\mu r)^5 n_3 \log^2(n \vee n_3) \kappa^5 \sigma_{\max}}{mn^2}} \\
&\leq 594C \sqrt{\frac{(\mu r)^5 \kappa^5 n_3 \log^2(n \vee n_3) \sigma_{\max}}{mn^2}}. \tag{185}
\end{aligned}
$$

We apply the Lemma 6 in (i) as

$$\mathcal{X}_0 = \begin{bmatrix} \mathcal{A}^\star \\ \mathcal{B}^\star \end{bmatrix}, \quad \mathcal{X}_1 = \begin{bmatrix} \mathcal{A}^0 \\ \mathcal{B}^0 \end{bmatrix} * \mathcal{R}^0, \quad \mathcal{X}_2 = \begin{bmatrix} \mathcal{A}^{0,l} \\ \mathcal{B}^{0,l} \end{bmatrix} * \mathcal{T}^{0,l}. \tag{186}$$

The conditions $\|\mathcal{X}_1 - \mathcal{X}_0\| \|\mathcal{X}_0\| \leq \frac{\sigma_{\min}}{2}$ and $\|\mathcal{X}_1 - \mathcal{X}_2\| \|\mathcal{X}_0\| \leq \frac{\sigma_{\min}}{4}$ can be satisfied based on (29),(26) and tube-wise sample complexity in (25). (ii) is due to the result in (28) and (29). $\square$

## B.5 Technical lemmas

This section provides the necessary lemmas used in proving that spectral initialization satisfied the results in Theorem 4. These can obtained based on results of matrix setting in [32, 8].

**Lemma 9.** *(T-eigenvalue decomposition [72]) Any symmetric tensor $\mathcal{X} \in \mathbb{R}^{n \times n \times n_3}$ (i.e., $\mathcal{X}^T = \mathcal{X}$) could be decomposed into $\mathcal{X} = \mathcal{U} * \mathcal{S} * \mathcal{U}^T$, where $\mathcal{U} \in \mathbb{R}^{n \times n \times n_3}$ is orthogonal tensor and $\mathcal{S} \in \mathbb{R}^{n \times n \times n_3}$ is a f-diagonal tensor that diagonal entries of $\overline{S}$ are T-eigenvalues of $\mathcal{X}$.*

**Lemma 10.** *Let the $\mathcal{X} \in \mathbb{R}^{n \times n \times n_3}$ be a symmetric tensor with top-r T-eigenvalue decomposition as $\mathcal{U} * \mathcal{S} * \mathcal{U}^T$ and $\|\mathcal{X} - \mathcal{X}^\star\| \leq \frac{\sigma_{\min}}{2}$ and denote*

$$\hat{\mathcal{Q}} := \arg \min_{\mathcal{R} \in \mathcal{O}_r} \|\mathcal{U} * \mathcal{R} - \mathcal{U}^\star\|_F, \tag{187}$$

*where $\mathcal{X}^\star$ has T-eigenvalue decomposition as $\mathcal{U}^\star * \mathcal{S}^\star * (\mathcal{U}^\star)^T$. Then there is*

$$\left\| \mathcal{U} * \hat{\mathcal{Q}} - \mathcal{U}^\star \right\| \leq \frac{3}{\sigma_{\min}} \|\mathcal{X} - \mathcal{X}^\star\|. \tag{188}$$

*Proof.* The proof applies the result under the matrix case for each frontal slice of the tensor in the frequency domain. Based on (117), $\forall k \in [n_3]$

$$\overline{\hat{Q}}^{(k)} := \arg \min_{R \in C_r} \left\| \overline{U}^{(k)} * R - \overline{U^\star}^{(k)} \right\|_F, \tag{189}$$

where $\overline{U}^{(k)}, \overline{U^\star}^{(k)} \in \mathbb{C}^{n \times r}$ are orthogonal matrices. Since $\left\| \overline{X}^{(k)} - \overline{X^\star}^{(k)} \right\| \leq \frac{\sigma_{\min}}{2} \leq \frac{\sigma_{\min}\left(\overline{X^\star}^{(k)}\right)}{2}$, based on Lemma 32 in [32], there is $\forall k \in [n_3]$

$$\left\| \overline{U}^{(k)} * \overline{\hat{Q}}^{(k)} - \overline{U^\star}^{(k)} \right\| \leq \frac{3}{\sigma_{\min}(\overline{X^\star}^{(k)})} \left\| \overline{X}^{(k)} - \overline{X^\star}^{(k)} \right\|, \tag{190}$$

which induces

$$\max_{k \in [n_3]} \left\| \overline{U}^{(k)} * \overline{\hat{Q}}^{(k)} - \overline{U^\star}^{(k)} \right\| \leq \frac{3}{\min_{k \in [n_3]} \sigma_{\min}(\overline{X^\star}^{(k)})} \max_{k \in [n_3]} \left\| \overline{X}^{(k)} - \overline{X^\star}^{(k)} \right\|, \tag{191}$$

which is the conclusion of the lemma. $\square$

**Lemma 11.** *Let $\mathcal{X}_1, \mathcal{X}_2 \in \mathbb{R}^{n \times n \times n_3}$ be symmetric tensors with top-r T-eigenvalue decomposition $\mathcal{U}_1 * \mathcal{S}_1 * \mathcal{U}_1^T, \mathcal{U}_2 * \mathcal{S}_2 * \mathcal{U}_2^T$ correspondingly. Assume for each frontal slice $\lambda_1(\overline{X}_1^{(k)}) = -\lambda_n(\overline{X}_1^{(k)}), \lambda_r(\overline{X}_1^{(k)}) > 0, \lambda_{r+1}(\overline{X}_1^{(k)}) = 0$ and*

$$\|\mathcal{X}_1 - \mathcal{X}_2\| \leq \frac{1}{40} \frac{\lambda_{\min}^{\frac{5}{2}}(\mathcal{X}_1)}{\lambda_{\max}^{\frac{3}{2}}(\mathcal{X}_1)}. \tag{192}$$

*Denote $\mathcal{A}_1 = \mathcal{U}_1 * \mathcal{S}_1^{\frac{1}{2}}, \mathcal{A}_2 = \mathcal{U}_2 * \mathcal{S}_2^{\frac{1}{2}}$ and define*

$$\hat{\mathcal{Q}} := \arg \min_{\mathcal{R} \in \mathcal{O}_r} \|\mathcal{U}_2 * \mathcal{R} - \mathcal{U}_1\|_F, \tag{193}$$

$$\mathcal{Q} := \arg \min_{\mathcal{R} \in \mathcal{O}_r} \|\mathcal{A}_2 * \mathcal{R} - \mathcal{A}_1\|_F. \tag{194}$$

*Then there is*

$$\left\| \hat{\mathcal{Q}} - \mathcal{Q} \right\| \leq 15 \frac{\lambda_{\max}^{\frac{3}{2}}(\mathcal{X}_1)}{\lambda_{\min}^{\frac{5}{2}}(\mathcal{X}_1)} \|\mathcal{X}_1 - \mathcal{X}_2\|. \tag{195}$$

*Proof.* The proof is similar to the proof in Lemma 10. Based on condition in (192), $\forall k \in [n_3]$, there is

$$\left\|\overline{\boldsymbol{X}}_1^{(k)} - \overline{\boldsymbol{X}}_2^{(k)}\right\| \leq \frac{1}{40} \frac{\lambda_{\min}^{\frac{5}{2}}(\boldsymbol{\mathcal{X}}_1)}{\lambda_{\max}^{\frac{3}{2}}(\boldsymbol{\mathcal{X}}_1)} \leq \frac{1}{40} \frac{\lambda_{\min}^{\frac{5}{2}}(\overline{\boldsymbol{X}}_1^{(k)})}{\lambda_{\max}^{\frac{3}{2}}(\overline{\boldsymbol{X}}_1^{(k)})}. \tag{196}$$

Combined with result in Lemma 34 in [32], we have

$$\left\|\overline{\widehat{\boldsymbol{Q}}}^{(k)} - \overline{\boldsymbol{Q}}^{(k)}\right\| \leq 15 \frac{\lambda_{\max}^{\frac{3}{2}}\left(\overline{\boldsymbol{X}}_1^{(k)}\right)}{\lambda_{\min}^{\frac{5}{2}}(\overline{\boldsymbol{X}}_1^{(k)})} \left\|\overline{\boldsymbol{X}}_1^{(k)} - \overline{\boldsymbol{X}}_2^{(k)}\right\|, \tag{197}$$

which induce

$$\max_{k \in [n_3]} \left\|\overline{\widehat{\boldsymbol{Q}}}^{(k)} - \overline{\boldsymbol{Q}}^{(k)}\right\| \leq 15 \frac{\lambda_{\max}^{\frac{3}{2}}(\boldsymbol{\mathcal{X}}_1)}{\lambda_{\min}^{\frac{5}{2}}(\boldsymbol{\mathcal{X}}_1)} \max_{k \in [n_3]} \left\|\overline{\boldsymbol{X}}_1^{(k)} - \overline{\boldsymbol{X}}_2^{(k)}\right\|, \tag{198}$$

which is exactly the result of the lemma. $\square$

**Lemma 12.** *Let $\boldsymbol{\mathcal{X}}_1, \boldsymbol{\mathcal{X}}_2, \boldsymbol{\mathcal{X}}_3 \in \mathbb{R}^{n \times n \times n_3}$ be symmetric tensors with top-$r$ T-eigenvalue decomposition $\boldsymbol{\mathcal{U}}_1 * \boldsymbol{\mathcal{S}}_1 * \boldsymbol{\mathcal{U}}_1^T, \boldsymbol{\mathcal{U}}_2 * \boldsymbol{\mathcal{S}}_2 * \boldsymbol{\mathcal{U}}_2^T$ and $\boldsymbol{\mathcal{U}}_3 * \boldsymbol{\mathcal{S}}_3 * \boldsymbol{\mathcal{U}}_3^T$ correspondingly. Assume for each frontal slice $\lambda_1(\overline{\boldsymbol{X}}_1^{(k)}) = -\lambda_n(\overline{\boldsymbol{X}}_1^{(k)}), \lambda_r(\overline{\boldsymbol{X}}_1^{(k)}) > 0, \lambda_{r+1}(\overline{\boldsymbol{X}}_1^{(k)}) = 0$ and $\|\boldsymbol{\mathcal{X}}_1 - \boldsymbol{\mathcal{X}}_2\| \leq \frac{\lambda_{\min}(\boldsymbol{\mathcal{X}}_1)}{4}, \|\boldsymbol{\mathcal{X}}_1 - \boldsymbol{\mathcal{X}}_3\| \leq \frac{\lambda_{\min}(\boldsymbol{\mathcal{X}}_1)}{4}$. Denote*

$$\widehat{\boldsymbol{\mathcal{Q}}} := \arg \min_{\boldsymbol{\mathcal{R}} \in \boldsymbol{\mathcal{O}}_r} \|\boldsymbol{\mathcal{U}}_2 * \boldsymbol{\mathcal{R}} - \boldsymbol{\mathcal{U}}_3\|_F. \tag{199}$$

*Then*

$$\left\|\boldsymbol{\mathcal{S}}_2^{\frac{1}{2}} * \widehat{\boldsymbol{\mathcal{Q}}} - \widehat{\boldsymbol{\mathcal{Q}}} * \boldsymbol{\mathcal{S}}_3^{\frac{1}{2}}\right\| \leq 15 \frac{\lambda_{\max}(\boldsymbol{\mathcal{X}}_1)}{\lambda_{\min}^{\frac{3}{2}}(\boldsymbol{\mathcal{X}}_1)} \|\boldsymbol{\mathcal{X}}_2 - \boldsymbol{\mathcal{X}}_3\|. \tag{200}$$

*Proof.* The proof follows the idea in Lemma 10 and Lemma 11 and applies the Lemma 33 in [32] to each frontal slice of $\overline{\widehat{\boldsymbol{Q}}}^{(k)}, \overline{\boldsymbol{S}}_2^{(k)}$ and $\overline{\boldsymbol{S}}_3^{(k)}$ to obtain the final result based on definition of $\lambda_{\max}(\boldsymbol{\mathcal{X}}_1), \lambda_{\min}(\boldsymbol{\mathcal{X}}_1)$ for third-order tensor. $\square$

# C Experiments on real world data

All experiments were conducted using MATLAB R2022a on a Windows system equipped with a 12th Gen Intel(R) Core(TM) i7-12700 CPU at 2.10 GHz and 16.0 GB of RAM.

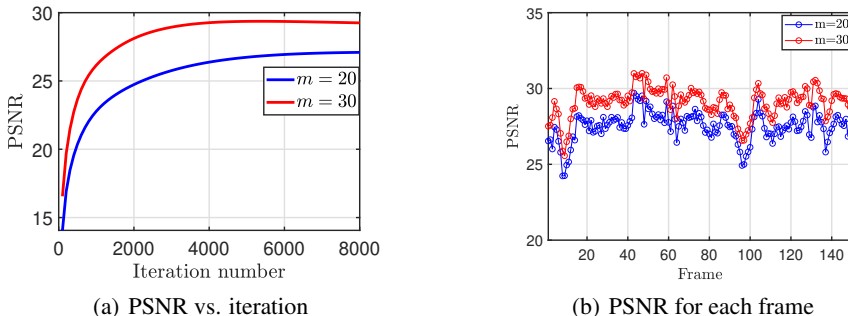

(a) PSNR vs. iteration          (b) PSNR for each frame

Figure 3: Convergence and PSNR values for video sequence recovery.

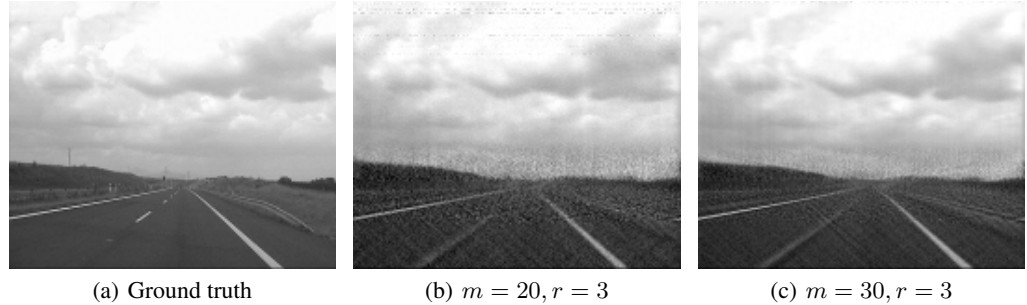

(a) Ground truth      (b) $m = 20, r = 3$      (c) $m = 30, r = 3$

Figure 4: Visual comparison at 17th frame .

We test the proposed local TCS model on the "highway" video sequence, which is reshaped into third-order tensor with size $144 \times 150 \times 176$. 144, 150 denote the spatial dimension, and 176 denotes the tube size. When we compare with FGD [26] and Alt-PGD-Min [64] methods, both methods require large sample sizes that exceed our machine's memory capacity, making them infeasible to run in our setting. Because the video has a full tubal rank and only approximately has a low tubal rank, we have no prior setting of the parameter $r$ in our model. Thus, we set $r = 3$ in our algorithm. For this $r$, we select the two tube-wise sample sizes as $m = 20, 30$, which are much smaller than the tube size 176. For setting the step size $\eta$, the video tensor has a large condition number $\kappa = 1.1408 \times 10^6$, which means the ground truth is an ill-conditioned tensor. We have to select small step sizes (0.00005) to guarantee the convergence, which leads to slow convergence rates. We use the PSNR metric to evaluate the recovery performance and plot the PSNR vs. iteration number in Fig. 3(a). We also plot the PSNR values across video frames in Fig. 3(b). The visual recovery results are shown in Fig. 4.

We can observe that although our recovered tensor only has $r = 3$, it has a very small reconstruction error for the ground-truth video sequence. This means that the original video sequence approximately has a low tubal rank. Besides the video sequence, MRI medical images and hyperspectral images often exhibit a low tubal rank structure because spectral (wavelength), temporal (frame), or slice axes are highly redundant. Low tubal rank-based tensor methods are often effective for these tasks, as shown in existing works in [23, 42, 57, 71]. Thus, based on these experimental results on real-world data, our low tubal rank Assumption 1 is realistic in practical applications.

We can observe that the GD converges slowly due to the very large condition number of the video tensor. This motivates us to use the preconditioning technique as [51, 52, 64] to get rid of dependence on $\kappa$ to accelerate the convergence rate. Despite the slow convergence and the fact that the ground truth video tensor is not strictly low-tubal-rank, GD still achieves satisfactory recovery performance with small sample sizes. Moreover, as the sample size increases, the recovery quality improves accordingly.

