# OpenReview forum: "Non-Convex Tensor Recovery from Tube-Wise Sensing"
_NeurIPS.cc/2025/Conference — NeurIPS 2025 poster_

### Official Review · Reviewer_TA7y · 2025-06-05

**Clarity:** 4
**Significance:** 3
**Originality:** 3
**Rating:** 5
**Confidence:** 3

**Summary:**

In conventional Tensor Compressed Sensing (TCS), the sensing operator $A_k$ acts on the entire tensor $X$, which has several drawbacks: (1) it is not scalable for large-scale data, (2) it is not suitable when the data size is variable, and (3) block-wise $Ak$ is preferable for privacy-preserving applications. To address these issues, slice-wise TCS approaches have been proposed. Inspired by existing slice-wise methods, the authors develop a tube-wise TCS method.

**Questions:**

### Suggestions:

I hope the following suggestions help improve the clarity and readability of the paper:

- [**S1**] Please explicitly state the dimension of the vector y in Equation (6). While I understood that y_ij corresponds to y(i,j,:) as defined in Definition 1, it took time for me to understand it.

- [**S2**] Consider describing the tensor size of R in Equation (11). Where is the set O_R defined?

- [**S3**] The notation for truncated SVD is inconsistent: it appears as “t-SVD” (Line 137) and “T-SVD” (Algorithm 2). Please unify this notation throughout the paper.

- [**S4**]  Please explain the meaning of the check symbol and the variable “r” (tubal-rank) in the caption of Table 1.

- [**S5**] At first glance, Equations (6) and (7) give the impression that the gradient is independent of y, which feels a bit odd. Of course, I understand that X itself depends on y, but I think it would be helpful to explicitly mention this dependency in the paragraph for clarity.

**Ethical Concerns:**

["NO or VERY MINOR ethics concerns only"]

**Final Justification:**

I did not identify any major weaknesses in this submission. In their rebuttal, the authors promised to improve the clarity of the paper, particularly by explicitly stating the assumptions of the main theorem, which I consider important. Given that my original score was already positive, I will keep it unchanged. I remain supportive of accepting this paper.

**Limitations:**

yes (but please fill in the justification in the checklist)

**Quality:**

4

**Strengths And Weaknesses:**

### Strengths:

- [**S1**] Despite the non-convex nature of the problem, the authors provide a theoretical guarantee that their proposed method can recover an exact low-rank tensor $X$ at a linear rate under the following conditions:
  - A sufficient number of measurements are available.
  - Each frontal slice of X has a very small rank.
  - X satisfies an incoherence condition.
  - The algorithm is initialized by spectral initialization.

- [**S2**] The sample complexity is shown to be proportional to 1/n, which is a favorable result.

- [**S3**] The introduction clearly motivates the problem by highlighting limitations in existing methods and proposing a justified new approach. The authors clearly explain the differences from prior work and the novelty of their own method, positioning their research well within the existing literature. The writing is easy to follow.

Overall, the paper is well-written, and these strengths suggest the paper makes a strong contribution to the community.

### Weaknesses:

The following are minor concerns, and I do not consider them grounds for rejection, provided the authors make slight revisions.

-  [**W1**] Although the method assumes a small tubal rank and the sample complexity is proportional to r^5, real-world datasets in the ML community often do not guarantee such low ranks. The authors assume rank-5 tensors in their experiments. It would be helpful to illustrate practical use cases where such an assumption is realistic and beneficial.

- [**W2**] The paper focuses only on third-order tensors. While I am not sure if this may be standard in signal processing, tensors of arbitrary order are frequently discussed in the ML community, such as tensor networks. I recommend that the authors either justify their focus on third-order tensors or discuss how their approach could be generalized for arbitrary order as future work.

- [**W3**] To avoid misunderstanding, Assumptions 1 and 2 should be more clearly explained in the introduction. The authors claim: “We prove that despite being nonconvex, with a sufficient amount of measurements, applying gradient descent with proper spectral initialization finds the ground truth X at a linear rate.” However, this result holds under the additional assumptions that each frontal slice has a sufficiently small rank (assumption 1) and the tensor satisfies the incoherence condition (assumption 2) . These assumptions are currently not clearly described in the abstract or introduction.

- [**W4**] In the checklist, the authors do not include a justification for their claims. While this is likely acceptable given the theoretical nature of the paper (with clear assumptions, proofs, and also source code), please ensure this section is properly filled in the camera-ready version.

---

> ### Author Rebuttal · Authors · 2025-07-29
>
> Thank you for your detailed review and professional comments. We have addressed your concerns and respond to your questions below point by point.
> # [W1]:
> *Although the method assumes a small tubal rank and the sample complexity is proportional to r^5, real-world datasets in the ML community often do not guarantee such low ranks. The authors assume rank-5 tensors in their experiments. It would be helpful to illustrate practical use cases where such an assumption is realistic and beneficial.*
> # Response:
> Thank you for your useful question. We have provided a real-world video recovery experiment in  our **Appendix C.2**, where we use $r=3$ to obtain the high recovery accuracy as shown in Fig. (3) and Fig. (4) of Appendix C.2. Although our recovered tensor only has $r=3$, it has a very small reconstruction error for the ground-truth video sequence. This means that the original video sequence approximately has a low tubal rank. Besides the video sequence,  MRI medical images and hyperspectral images often exhibit a low tubal rank structure because spectral (wavelength), temporal (frame), or slice axes are highly redundant.
> Low tubal rank-based tensor methods are often effective for these tasks, as shown in existing works in [1-8]. Thus, this assumption is realistic in practical applications.
> # [W2]:
> *The paper focuses only on third-order tensors. While I am not sure if this may be standard in signal processing, tensors of arbitrary order are frequently discussed in the ML community, such as tensor networks. I recommend that the authors either justify their focus on third-order tensors or discuss how their approach could be generalized for arbitrary order as future work.*
> # Response:
> Thank you for insight suggestion. We agree that higher‑order tensors play a vital role in many ML applications. Here we focus on third‑order tensors because 3‑way data are prevalent in many applications—e.g., video (height $\times $ width $\times$ time), hyperspectral imagery (row $\times $ column $\times $ wavelength), and dynamic MRI (x$\times $y $\times $ time). The mathematical properties of t‑product/t‑SVD and tubal‑rank are relatively more mature for third‑order arrays than higher order ones, which allows us to  build our work on these recent results in [52].
>
> A potential direction to extend our local TCS model to arbitrary‑order tensors could be leveraging the recently developed high‑order tensor algebra framework in [9-11]. This will allow us to generalize our low‑rank, incoherence assumptions and the leave‑one‑out proof techniques to develop a general local TCS model for tensors of any order.
> # [W3]:
> *To avoid misunderstanding, Assumptions 1 and 2 should be more clearly explained in the introduction. The authors claim: “We prove that despite being nonconvex, with a sufficient amount of measurements, applying gradient descent with proper spectral initialization finds the ground truth X at a linear rate.” However, this result holds under the additional assumptions that each frontal slice has a sufficiently small rank (assumption 1) and the tensor satisfies the incoherence condition (assumption 2) . These assumptions are currently not clearly described in the abstract or introduction.*
> # Response:
> Thank you for the suggestion. We will explicitly mention these conditions in the abstract and introduction in the revision.
> # [W4]:
> *In the checklist, the authors do not include a justification for their claims. While this is likely acceptable given the theoretical nature of the paper (with clear assumptions, proofs, and also source code), please ensure this section is properly filled in the camera-ready version.*
> #  Response:
>  Thank you for the notification. We will fix this in the revision.
> # [S1]:
> *Please explicitly state the dimension of the vector y in Equation (6). While I understood that y_ij corresponds to y(i,j,:) as defined in Definition 1, it took time for me to understand it.*
> # Response:
> Thank you for pointing this out.  In the revision, we will clarify that $y \in \mathbb R^m$, where $m$ is the number of measurements.
> # [S2]:
> *Consider describing the tensor size of R in Equation (11). Where is the set O_R defined?*
> # Response:
> Thank you for noting this omission. The set ${\mathcal O}_r$ includes all orthogonal tensors with size $r\times r \times n_3$. Thus. the tensor size of $\mathcal R$ in Eq. (11) is $r\times r \times n_3$.
> # [S3]:
> *The notation for truncated SVD is inconsistent: it appears as “t-SVD” (Line 137) and “T-SVD” (Algorithm 2). Please unify this notation throughout the paper.*
> # Response:
> Thank you for highlighting this inconsistency. We will standardize the notation to “T‑SVD” throughout the manuscript.
> # [S4]:
> *Please explain the meaning of the check symbol and the variable “r” (tubal-rank) in the caption of Table 1.*
> # Response:
> Thank you for your careful observation. We will add the explanations in the revision.
> # [S5]:
> *At first glance, Equations (6) and (7) give the impression that the gradient is independent of y, which feels a bit odd. Of course, I understand that X itself depends on y, but I think it would be helpful to explicitly mention this dependency in the paragraph for clarity.*
> # Response:
> Thank you for this helpful suggestion. We have substituted the expression of measurement $\mathcal Y$ defined in Eq. (2) into the loss function Eq. (6), which gives the gradient in Eq. (7) and Eq. (8). We will add this explanation to avoid confusion.
>
> # Reference
> [1] Chunyan Liu, Sui Li, Dianlin Hu, Jianjun Wang, Wenjin Qin, Chen Liu, and Peng Zhang. Nonlocal tensor decomposition with joint low rankness and smoothness for spectral ct image reconstruction. **IEEE Transactions on Computational Imaging**, 10:613–627, 2024.263.\
> [2] Wei Pu, Junjie Wu, Yulin Huang, and Jianyu Yang. Ortp: A video sar imaging algorithm based on low-tubal-rank tensor recovery. **IEEE Journal of Selected Topics in Applied Earth Observations and Remote Sensing**, 15:1293–1308, 2021.\
> [3] Andong Wang, Chao Li, Mingyuan Bai, Zhong Jin, Guoxu Zhou, and Qibin Zhao. Transformed low-rank parameterization can help robust generalization for tensor neural networks. **Advances in Neural Information Processing Systems**, 36:3032–3082, 2023.\
> [4] Andong Wang, Yuning Qiu, Mingyuan Bai, Zhong Jin, Guoxu Zhou, and Qibin Zhao. Generalized tensor decomposition for understanding multi-output regression under combinatorial shifts. **Advances in Neural Information Processing Systems**, 37:47559–47635, 2024.\
> [5] Andong Wang, Yuning Qiu, Zhong Jin, Guoxu Zhou, and Qibin Zhao. Low-rank tensor transitions (lort) for transferable tensor regression. In **Forty-second International Conference on Machine Learning**.292\
> [6] Zemin Zhang, Gregory Ely, Shuchin Aeron, Ning Hao, and Misha Kilmer. Novel methods for multilinear data completion and de-noising based on tensor-svd. In Proceedings of the **IEEE conference on computer vision and pattern recognition**, pages 3842–3849, 2014\
> [7] Xi-Le Zhao, Hao Zhang, Tai-Xiang Jiang, Michael K Ng, and Xiong-Jun Zhang. Fast algorithm with theoretical guarantees for constrained low-tubal-rank tensor recovery in hyperspectral images denoising. **Neurocomputing**, 413:397–409, 2020.\
> [8] Salman Ahmadi‑Asl et al. Efficient Algorithms for Low‑Tubal‑Rank Tensor Approximation with Applications to Image Compression, Super‑Resolution, and Deep Learning, arXiv:2412.02598 (2024).\
> [9] Sheng Liu, Xi-Le Zhao, Jinsong Leng, Ben-Zheng Li, Jing-Hua Yang, and Xinyu Chen. Revisiting high-order tensor singular value decomposition from basic element perspective. **IEEE Transactions on Signal Processing**, 2024.266.\
> [10] Wenjin Qin, Hailin Wang, Feng Zhang, Jianjun Wang, Xin Luo, and Tingwen Huang. Low-rank high-order tensor completion with applications in visual data. **IEEE Transactions on Image Processing**, 31:2433–2448, 2022.\
> [11] Dan Garber et al. Efficiency of First-Order Methods for Low-Rank Tensor Recovery with the Tensor Nuclear Norm Under Strict Complementarity. arXiv:2308.01677 (2023).

---

> ### Comment · Reviewer_TA7y · 2025-08-01
>
> Thank you for your very informative response. I am fully satisfied with your clarifications. The authors have promised to address several points based on my suggestions, which I believe will make the paper more readable. Therefore, I will increase my clarity score from 3 to 4.
>
> I would just like to add two minor points:
>
> > The set ${\mathcal O}_r$ includes all orthogonal tensors with size $r\times r \times n_3$.
>
> Please consider including this statement in the final version for clarity.
>
> > We have provided a real-world video recovery experiment in our Appendix C.2, where we use $r=3$ to obtain the high recovery accuracy as shown in Fig. (3) and Fig. (4) of Appendix C.2. Although our recovered tensor only has $r=3$, it has a very small reconstruction error for the ground-truth video sequence. This means that the original video sequence approximately has a low tubal rank. Besides the video sequence, MRI medical images and hyperspectral images often exhibit a low tubal rank structure because spectral (wavelength), temporal (frame), or slice axes are highly redundant. Low tubal rank-based tensor methods are often effective for these tasks, as shown in existing works in [1-8]. Thus, this assumption is realistic in practical applications.
>
> I found this explanation very convincing. Thank you for the detailed clarification. I suggest adding the rank used in the caption of Figure 4. For example, replacing "m = 20" with "m = 20, r = 3".

---

> > ### Author Response · Authors · 2025-08-01
> >
> > Thank you for the positive feedback and increasing our clarity score. We promise to incorporate all of your suggestions in revised version, including the two new points you provided.
> >
> > 1. We will explicitly state “$\mathcal O_r$ is the set of all orthogonal tensors of size $r\times r \times n_3$'' when it first appears.
> >
> > 2. We will change “$m=20$'' with “$m=20, r=3$” in caption of Figure 4 to make the chosen tubal rank clear.
> >
> > We appreciate your careful reading and helpful suggestions.

---

### Official Review · Reviewer_sxUs · 2025-07-01

**Clarity:** 3
**Significance:** 3
**Originality:** 2
**Rating:** 4
**Confidence:** 4

**Summary:**

This manuscript propose a new tensor compressed sensing (TCS) model with local tube-wise strategy. The method works by minimizing a nonconvex objective via Burer-Monteiro factorization. Besides, the authors prove that this method achieves exact recovery with a linear convergence rate.

**Questions:**

1. The slice-wise method, i.e., Alt-PGD-Min, and the proposed method are both local TCS models. What are the advantages of the tube-wise method compared with the slice-wise approach?

2. This paper provides tube-wise sample complexity. However, the experiments do not involve a comparison of the sample complexity with other methods. How can the effectiveness of the proposed tube-wise sample complexity be sufficiently verified?

3. There was no comparison with other benchmark methods in the experiments. Moreover, the effectiveness of the proposed method in practical application scenarios remains to be verified.

4. The author claims that tube-wise is more flexible. How is this flexibility demonstrated and supported in the paper?

5. Some typos exist, e.g., “GD” in abstract and several definitions lack references.

**Ethical Concerns:**

["NO or VERY MINOR ethics concerns only"]

**Final Justification:**

After reading the rebuttal, I am satisfied with the new experiments, so I decide to raise my score.

**Limitations:**

Yes

**Quality:**

3

**Strengths And Weaknesses:**

This paper introduces a local tube-wise method under the tensor product framework. The corresponding detailed definitions and theory are provided. Additionally, experiments analyze about convergence, initialization, and sample complexity.

However, this paper lacks sufficient experiments to demonstrate the contributions of this work. More comparative methods need to be added. Furthermore, the motivation from slice-wise to tube-wise requires further description.

---

> ### Author Rebuttal · Authors · 2025-07-29
>
> Thank you for your professional comments. We appreciate the opportunity to address your concerns here point by point.
> # Weakness 1:
> *However, this paper lacks sufficient experiments to demonstrate the contributions of this work. More comparative methods need to be added.*
> # Response:
> Due to page limits, we have placed a part of  experimental results in Appendix C (please refer to **pp. 54–55 in Appendix.pdf**), which may be easy to miss. Comparison  with the global TCS method FGD [23] and the latest slice
> wise TCS method Alt-PGD-Min [52] on both synthetic and real-world data have been provided therein.
>
> In addition, during the rebuttal period, we ran additional experiments that include new comparisons with two global TCS methods: the non‑convex **Schatten‑$2/3$ TNN** [17] and the convex **TNN** [23]. We report results for $n=20$ and $n=50$ only, as both  of them encounter out-of-memory issues for larger $n$ due to the large sample size. The comparison results are summarized in the following table.
>
> # Table 3:
>  Comparison of runtime and the required number of measurements $m$ to achieve the same level of relative recovery error in tensor compressive sensing (TCS). (**m** is the sample size in tensor unit, **n₃**  is the tube size, **n** is the spatial dimension, **r** is the tubal rank ).
>
> | **n₃(r)**  | **n** | **FGD  (m / time)** | **TNN (m / time)** | **Schatten-2/3 TNN (m / time)** | **Alt-PGD-Min  (m / time)** | **Ours (m / time)** |
> |:---------------:|:-----:|:------------------:|:------------------:|:-------------------------------:|:--------------------------:|:-------------------:|
> | **10 (r=2)**    | 20    | 840 / 7.1          | 1450 / 1.6        | 1020 / 5.1                    | 43 / 13.8                  | 3 / 1.1             |
> | **10 (r=2)**    | 50    | 2200 / 175.7       | 4600 / 109.4       | 3100 / 181.7                    | 45 / 15.6                  | 2 / 3.7             |
> | **20 (r=3)**    | 20    | 2400 / 86.9        | 3950 / 7.7         | 2750 / 20.7                    | 130 / 13.8                 | 7 / 4.5             |
> | **20 (r=3)**    | 50    | 6800 / 442.4       | 13000 / 717.5      | 8500 / 988.2                    | 135 / 80.4                 | 3 / 11.6            |
> | **30 (r=4)**    | 20    | 4900 / 181.2       | 7100 / 20.9       | 5250 / 90.0                     | 250 / 47.3                 | 12 / 16.1           |
> | **30 (r=4)**    | 50    | 13500 / 1626.1     | 24000 / 2395.6     | 16500 / 2344.9                  | 268 / 155.9                | 6 / 18.9            |
>
> It can be observed that our method attains the same relative recovery error $\frac{|| \hat{\boldsymbol{\mathcal X}^\star}- \boldsymbol{\mathcal X}^\star||_F}{|| \boldsymbol{\mathcal X}^\star ||_F}$ with the **minimum** sample complexity  and runtime.
>
> # Weakness 2:
> *Furthermore, the motivation from slice-wise to tube-wise requires further description.*
> # Response:
> Thanks for your helpful suggestions. In line 25-51 of the main paper, we have provided a short paragraph motivating why local sensing is of interest.  For a more detailed motivation of using local measurement model from an application perspective, please see our response to first Reviewer 58Bp's first question. In what follows, we provide a detailed elaboration why we move from slice-wise to tube-wise sensing within the local TCS framework:\
> **(1) Practical Applications.** There are applications where tube‑wise access is natural or mandatory. This is because some real‑world systems provide access to one tube of the tensor at a time, rather than an entire slice. For example, pushbroom (line scan) hyperspectral imagers used on aircraft, satellites or in conveyor belt inspections acquire one spatial line per exposure but capture the full spectrum of that line. Until the scan completes, one never have a full spectral slice (all pixels at a single wavelength) in memory; however, each pixel’s spectrum (a tube) is instantly available. Any algorithm requiring complete slices cannot run online, whereas a tube-wise method can. Similarly, rolling‑readout video sensors (e.g. CMOS cameras) stream raster lines one after another. In low‑latency or high‑speed settings, you see each pixel’s temporal tube well before you have a clean, full frame. A tube wise approach lets one process data as it arrives, supporting true streaming recovery with only small buffers.\
> **(2) Motivated from Existing Work in [52].** Latest work [52] proposed slice-wise local TCS model that provably recover the ground truth tensor  of low tubal rank with a lower sample complexity than the global TCS model. We are interested in whether an even finer‑grained local TCS model—tube‑wise sensing—can further reduce the sample complexity. This  motivates our shift from slice‑wise to tube‑wise setting.\
> **(3) Sample and Computational Efficiency.**  Theoretically, we show that tube‑wise sensing achieves  lower sample complexity than existing tensor compressed sensing (TCS) models, including slice‑wise sensing. Specifically, in Table 1 of the main paper, to exactly recover an $n\times n\times n_{3}$ tensor of low tubal rank $r$, the slice‑wise model of [52] requires $\mathcal O(r^2n_3)$ samples, whereas our tube‑wise model needs only $\mathcal O(r^5n_3/n)$.
> Empirically, the results in the above **Table 3** in our response to your Weakness 1 validate that the tube‑wise model outperforms the slice‑wise approach in both sample complexity and runtime.
>
> # Question 1:
> *The slice-wise method, i.e., Alt-PGD-Min, and the proposed method are both local TCS models. What are the advantages of the tube-wise method compared with the slice-wise approach?*
> # Response:
> Thank you for the question. As stated in point (3) of our reply to your Weakness 2 , tube-wise is more efficient than slice-wise sensing in both sample complexity and computational cost, which is validated by both theory and experiments in **Table 3** above. We also note that the two approaches address different scenarios, with the tube-wise and slice-wise sensing applying, respectively, to applications where measurements are taken with respect to a tube and a slice of the ground truth tensor.
> # Question 2:
> *This paper provides tube-wise sample complexity. However, the experiments do not involve a comparison of the sample complexity with other methods. How can the effectiveness of the proposed tube-wise sample complexity be sufficiently verified?*
> # Response:
> Thank you for insight comments. The  theoretical comparison of sample complexities has been provided in **Table 1 of main paper**. Experimental validation of our method’s sample efficiency has been provided in **Table 2 (Appendix C.pdf)** and the **Table 3** in our reply to your Weakness 1. In all settings, our tube‑wise model achieves exact recovery with the **minimum** sample complexity and runtime.
> # Question 3:
> *There was no comparison with other benchmark methods in the experiments. Moreover, the effectiveness of the proposed method in practical application scenarios remains to be verified.*
> # Response:
> Some of our experimental results are provided in the **Appendix.pdf** due to the page limit. Specifically,\
> (1) Comparison with FGD and Alt‑PGD‑Min has been provided in **Appendix C.1**. To make the evaluation more comprehensive,  we have added  two extra benchmarks: nonconvex Schatten‑$2/3$ TNN [2] and convex TNN [7]. The result is shown in the above **Table 3**.\
> (2) To show its effectiveness on practical applications, we have tested it on a video‑recovery problem in **Appendix C.2**. Here, we have not compared with the benchmark methods because they require a large number of measurements for recovery, which leads to memory and computational costs exceeding our computational resources at hand.
> # Question 4:
> *The author claims that tube-wise is more flexible. How is this flexibility demonstrated and supported in the paper?*
> # Response:
> Thank you for your insightful question. For applications where one can design the sensing tensor, tube‑wise method is more flexible because:\
> **(1)** Tube‑wise sensing only requires access to individual tubes (mode‑3 fiber vectors) of the tensor, whereas slice‑wise and global sensing methods must measure the entire frontal slices (full matrices) and the whole tensor, respectively. \
> **(2)** Given samples obtained via the tube-wise TCS model in Definition 1, we can  turn it into an equivalent slice-wise or global TCS model, but not the other way around. We take the slice-wise model as an example to elaborate. Let $\mathcal X \in \mathbb R^{n\times n \times n_3}$ be the tensor to be measured and $\mathcal X(:,j,:)$ be its $j$-th lateral slice composed of $n$ tubes, namely, $\mathcal X(i,j,:)$ for $i=1,\cdots,n$. In tube-wise TCS, the $m$-th measurement of tube $\mathcal X(i,j,:)$ is $\mathcal Y(i,j,m)=\mathcal Z^{ij}(m,:)\mathcal X (i,j,:)$, where $\mathcal Z^{ij}(m,:)$ is the $m$-th sensing vector for tube $\mathcal X(i,j,:)$.
> Summing up these measurements over $i$ (all the tubes belonging to the $j$-th slice)  as $ \hat{\mathcal Y}(j,m): = \sum_{i=1}^n \mathcal Y(i,j,m)$ gives the $m$-th measurement for $j$-th slice in the slice-wise TCS model: $\hat{\mathcal Y}(j,m) = \langle\hat {\mathcal Z}^{j,m},\mathcal X(:,j,:)\rangle$, where the sensing matrix $\hat{\mathcal{Z}}^{j,m}$ is formed by taking each $\mathcal{Z}^{ij}(m,:)$ as its $i$‑th column. On the other hand, a slice‑wise model cannot be converted into a tube‑wise one because its measurements mix information across the tubes. \
> **(3)** In terms of implementation cost, tube‑wise TCS needs fewer measurements and incurs less memory cost than slice‑wise methods. Therefore, the method can be more flexibly applied to real-world applications under resource constrains such as computing power.
> # Question 5:
> *Some typos exist, e.g., “GD” in abstract and several definitions lack references.*
> # Response:
> Thank you for catching these. "GD" is short for "gradient descent" and we will fixed these typos in the revision.

---

> ### Comment · Reviewer_sxUs · 2025-08-04
> **raise the score**
>
> After reading the rebuttal, I am satisfied with the new experiments, so I decide to raise my score.

---

> > ### Author Response · Authors · 2025-08-04
> >
> > We sincerely appreciate you taking the time to read our response and raising your score. We’re glad that the additional experiments and clarifications addressed your concerns. We’ll incorporate all your helpful suggestions into the revised version. Thank you again for your constructive and valuable feedback.

---

### Official Review · Reviewer_UaeD · 2025-07-02

**Clarity:** 3
**Significance:** 3
**Originality:** 3
**Rating:** 5
**Confidence:** 3

**Summary:**

The paper consider a tensor compressive sensing problem, with tube-wise linear measurements as defined in Definition 1, and under a low tubal rank assumption. They show that using gradient descent with a Burer-Monteiro factorization of the matrix is able to recover the original tensor and provide several theoretical guarantees.

**Questions:**

- Line 48: Tubal rank is yet to be defined. Here the authors should point to the definition of tubal rank (Definition 8).
- Line 129: The definition of $\mathcal{I}$ is inconsistent. The authors alternate between using $\mathcal{I}$ and $\mathcal{I}_n$. I think dropping the index and using $\mathcal{I}$ is fine as it is clear its definition in the context.
- Line 146: It should be made more clear at this time what the second term does in the loss does. In particular I believe it should be mentioned that the minima are the balanced factors defined in 162, up to an orthogonal transformation (which is referred in equation (11)).
- The norm in (10) is for a matrix, yet the authors refer to a tensor norm.
- In section 4, let us suppose that another random model for $\mathcal{Z}$ also possesses an RIP condition. Would that imply the results obtained also hold for that random model?
- The authors consider the $t$-rank of a tensor, which is the $M$-rank of a tensor when $M$ is the DFT matrix. Would the results obtained hold for other $M$-products?

**Ethical Concerns:**

["NO or VERY MINOR ethics concerns only"]

**Final Justification:**

The authors addressed all my questions and concerns. I recommend acceptance.

**Limitations:**

The authors address the limitations adequately.

**Quality:**

3

**Strengths And Weaknesses:**

The topic of the paper is quite interesting. Although the applications of the paper are limited, as the model considered by the authors seldom arises in applications, the mathematical analysis behind the results makes up for that.  They provide a very complete analysis of the algorithm, under random settings, and obtain results in line with other lines of work. Moreover, they obtain an interesting dependence between sample complexity and the dimensions of the tensor $(O(r^5 n_3/\min(n_1, n_2))$, where the tensor is $n_1\times n_2 \times n_3$.

---

> ### Author Rebuttal · Authors · 2025-07-29
>
> Thank you for your detailed review and comments. We have addressed your concerns and respond to your questions below point by point.
> # Question 1:
> *Line 48: Tubal rank is yet to be defined. Here the authors should point to the definition of tubal rank (Definition 8).*
> # Response:
> Thank you for pointing this out. We will revise accordingly.
> # Question 2:
> *Line 129: The definition of $\mathcal I$ is inconsistent. The authors alternate between using $\mathcal I$
>  and $\mathcal I_n$. I think dropping the index and using
>  it is fine as it is clear its definition in the context.*
> # Response:
> Thank you for pointing out this notational inconsistency. We will drop the index and unify the notation by using $\mathcal I$ throughout the paper.
> # Question 3:
> *Line 146: It should be made more clear at this time what the second term does in the loss does. In particular I believe it should be mentioned that the minima are the balanced factors defined in 162, up to an orthogonal transformation (which is referred in equation (11)).*
> # Response:
> Thank you for your professional comment. The second term in line 146 is to balance the norms of factor $\mathcal A$ and factor $\mathcal B$. To see this, if there is no second term, then the loss in line 146 becomes $\mathcal L\left( \mathcal A, \mathcal B \right):= \sum_{i=1}^{n_1}\sum_{j=1}^{n_2}  ||  Z_{ij} \left(\mathcal A(i,:,:) * \left(  \mathcal B(j,:,:) \right)^T \right) -  y_{ij} ||^2_F$, which has identifiability issue since for any invertible $\mathcal R \in \mathbb R^{r\times r \times n_3}$, factors $(\mathcal A,\mathcal B)$ and  $(\mathcal A*\mathcal R^{-1},\mathcal B*\mathcal R^T)$ has the same loss value. Such ambiguity may result in the norms of the two factors being highly unbalanced, leading to divergence in  the algorithm. The second term is added to penalize such an imbalance.
>
> Indeed, we agree with your comment and will add this discussion in the revision.
> # Question 4:
> *The norm in (10) is for a matrix, yet the authors refer to a tensor norm.*
> # Response:
> Thank you for pointing this out. In Eq. (10), $\mathcal U^\star(i,:,:) \in \mathbb R^{1\times r \times n_3}$ is a degenerate third‐order tensor. Although it can be viewed as an $r \times n_3$  matrix, the norm here indeed stands for the tensor spectral norm. We will clarify this in the revision.
> # Question 5:
> *In section 4, let us suppose that another random model also possesses an RIP condition. Would that imply the results obtained also hold for that random model?*
> # Response:
> Thank you for the insightful question. If an alternative random model also satisfies the RIP condition in Eq. (17), we anticipate that our analysis can be generalized. However, we would like to note that the major technical challenge actually lies in proving the  RIP condition. One of the difficulties is to find the structural assumption on $\mathcal{X}$ so as to obtain a tight sample complexity.   For tube-wise TCS, our RIP is proved to hold over the set of all low rank tensors satisfying an incoherence condition as stated in Assumption 2. Such a set often depends on the sensing model and needs a case-by-case study.
> For example, if one  aims to prove the  RIP that is used in global TCS analysis: the sensing map $\boldsymbol{\mathcal A}: \mathbb R^{n\times n\times n_3} \rightarrow \mathbb R^m$ in global TCS model satisfies T-RIP condition with parameter $\delta$, if it has $ (1-\delta)||\mathcal X ||_F^2 \leq ||\boldsymbol{\mathcal A}({\mathcal X})||_F^2\leq (1+\delta) ||{\mathcal X} ||_F^2$ for $\forall {\mathcal X} \in \mathbb R^{n\times n\times n_3}$ with tubal-rank of at
> most $r$ (cf. Definition 2.12 in [22] ), then higher sample complexity would be needed with samples taken by our local TCS model.
>  Moreover, our main  result presented in Theorem 2 shows that the GD algorithm provably recovers $\mathcal{X}^\star$. To this end, we must prove that all the iterates satisfy the incoherence condition, and thus RIP holds along the trajectory of the algorithm. This is another main technical challenge and cannot be assumed a priori.
>
> In summary, RIP alone is insufficient. To inherit our results, a new model must (a) achieve the same‑order sample complexity for the desired RIP and (b) provide a mechanism  ensuring the iterates stay within the region where that RIP holds. Our high-level of proof may be generalized, but verifying these two ingredients requires a dedicated analysis for specific sensing model.
> # Question 6:
> *The authors consider the t-rank of a tensor, which is the M-rank of a tensor when M is the DFT matrix. Would the results obtained hold for other M-products?*
> # Response:
> Thank you for the question. Our analysis extends to any transform matrix satisfying $M^TM = MM^T = n_3I$, a property enjoyed by the DFT matrix. This condition is standard in the transformed t‑SVD framework [1-3]. Thus, our results still hold for such $M$-products.
>
> # Reference
> [1] Lin Chen, Li Ge, Xue Jiang, Hongbin Li, and Martin Haardt. Slim is better: Transform-based tensor robust principal component analysis. **IEEE Transactions on Signal Processing**, 2025.\
> [2] Canyi Lu. Transforms based tensor robust pca: Corrupted low-rank tensors recovery via convex optimization. In Proceedings of the **IEEE/CVF International Conference on Computer Vision**,270 pages 1145–1152, 2021.\
> [3] Canyi Lu, Peng Xi, and Wei Yunchao. Low-rank tensor completion with a new tensor nuclear norm induced by invertible linear transforms. In **IEEE International Conference on Computer Vision and Pattern Recognition (CVPR)**, 2019.

---

> > ### Comment · Reviewer_UaeD · 2025-08-05
> >
> > Thank you for addressing my comments. Regarding question 6, I think the authors should add this information in the manuscript

---

> > > ### Author Response · Authors · 2025-08-05
> > >
> > > Thank you for your feedback. We will incorporate all of your suggestions, including our response in question 6, into the revised manuscript. Thank you very much again for your valuable and constructive comments.

---

### Official Review · Reviewer_58Bp · 2025-07-03

**Clarity:** 3
**Significance:** 3
**Originality:** 3
**Rating:** 4
**Confidence:** 3

**Summary:**

This paper studies spectral-initialized factorized GD for tensor sensing problem under the so-called local measurement model, where each one-dimensional slice along the third axis $\\mathcal{X}(i, j, :)$ is measured by i.i.d. Gaussian matrices. Under a somewhat stronger coherence condition than the usual one, it is proved that factorized GD converges linearly when the number of measurements $\\gtrsim r^5  n_3 n$, where it is assumed $n=n_1=n_2$.

**Questions:**

Please refer to the Weaknesses part.

**Ethical Concerns:**

["NO or VERY MINOR ethics concerns only"]

**Final Justification:**

The author has addressed my concerns, and I will keep my positive recommendation.

**Limitations:**

yes

**Quality:**

3

**Strengths And Weaknesses:**

## Strengths
The local measurement model is relatively new (built upon the idea of [58]). The overall approach appears rigorous, though I have not checked all the details in the proof.

## Weaknesses
- It is not clear whether the local measurement model is of significant practical relevance. It would be beneficial to discuss potential real-world applications.
- It is not adequately addressed in the paper whether the local measurement model is simpler or harder to deal with than the more common models. Given the similarities of the proof to well-known proofs in the literature, the main technical contribution of the paper should be highlighted better.

---

> ### Author Rebuttal · Authors · 2025-07-29
>
> Thank you for your detailed review and comments. We have addressed your concerns and respond to your questions below point by point.
> # Weakness 1:
> *It is not clear whether the local measurement model is of significant practical relevance. It would be beneficial to discuss potential real-world applications.*
> # Response:
> The local measurement model is useful when each sensing operator can only  be applied to a small slice/tube/fiber of the ground truth tensor (e.g., one spatial line, one spectral vector, or one 2‑D slice). This is exactly how many real-world  systems are engineered, either because of hardware constraints (e.g., detectors read row by row, line by line, or slice by slice) or because the scene/object is physically scanned  piece by piece. For example, local sensing can be potentially used in the following two applications:\
> **(1) Hyperspectral remote-sensing cameras (pushbroom / line-scan).** Pushbroom sensors on aircraft and satellites acquire one spatial line at a time while recording the full spectrum for that line, producing exactly tube-wise (line × wavelength) measurements. Industrial line-scan cameras similarly read a single row each clock cycle for high-speed inspection on conveyor belts.\
> **(2)  Medical and biological volumetric imaging (slice/plane scanning).**  MRI / fMRI: 3‑D volumes are routinely built from multiple 2‑D slices acquired sequentially (multi-slice EPI, etc.), so each acquisition operator is confined to a slice. Practical CS-MRI implementations often undersample $k$-space slice by slice or frame by frame, exploiting local sampling patterns rather than a single global operator—exactly the scenario our theory targets.
>
> In summary, due to hardware designs, many sensors cannot implement dense, global random projections; they naturally produce localized measurements (lines, slices, small patches). Additionally, for throughput and memory considerations, local readout enables devices to stream data with small buffers and parallel electronics, which are critical for high-resolution or high-speed systems.
>
> # Weakness 2:
> *It is not adequately addressed in the paper whether the local measurement model is simpler or harder to deal with than the more common models. Given the similarities of the proof to well-known proofs in the literature, the main technical contribution of the paper should be highlighted better.*
> # Response:
> The local (tube‑wise) model is harder than standard global models, because we must show that with lower sample complexity it still admits exact recovery at a linear convergence rate—something we validate empirically in Figure 1 of main paper and Table 2 of Appendix.
>
> We are agree with you that there are several well-known proofs in common TCS models such as convex nuclear norm, projected GD, alternating minimization, and nonconvex factor GD (FGD). Our method falls under the last category. However,
> straightforwardly generalizing their techniques without exploiting the local tube-wise sensing model structure will not lead to the sample complexity provided in our paper. Detailed elaborations are given as follows:\
> **(1)** For the global TCS model, existing proofs used a Bernstein inequality that is uniformly applicable to all rank $r$ matrices or tensors to prove the   Restricted Isometry Property (RIP) condition. If we use the same inequality, the sample size required to establish the RIP condition given in Proposition 1 would be much larger than our current result due to ignoring the structures of the tube-wise sensing model.\
> **(2)** Prior works leverage global RIP (please refer to our response to Question 5 in second Reviewer UaeD ) to show the loss function satisfies the restricted strong convexity (RSC) and smoothness (RSM) property on a neighborhood of optimum and then prove linear convergence of GD using a spectral initializer. Our RIP, on the other hand, only holds for incoherent rank‑$r$ tensors, but not uniformly over all rank‑$r$ tensors. As such, to prove GD recovers $\mathcal X^\star$, we need to show all the iterates stay in a region that is smaller than existing works. To establish this result, we derived new tubal/t‑product norm inequalities (Appendix, Lemma 1), tube‑wise concentration bounds in Lemmas 2, 3, 7, and a proof for Eq. (23) in the Appendix; and combined them with a leave‑one‑out argument that decouples iterate–measurement dependence along tubes. To our knowledge, all these are new in TCS literature.

---

> > ### Comment · Reviewer_58Bp · 2025-08-05
> >
> > Thank you for your detailed response. My comments are addressed properly, and I have no further questions.

---

> > > ### Author Response · Authors · 2025-08-05
> > >
> > > Thank you very much for your comments and follow-up message. We are pleased that our clarifications addressed your concerns.

---

### Decision · Program_Chairs · 2025-09-17

**Decision:**

Accept (poster)

**Comment:**

The paper develops gradient descent for nonconvex tensor recovery from tube-wise sensing. The analysis built upon existing GD analyses of tensor/matrix completion with leave-one-out arguments. Although the analysis does not bring new insights, the cleanliness of the results and potential applicability in applications makes the paper well-appreciated by the reviewers. The reviewers are all positive about the paper and I recommend acceptance.